

# LPJmL4 - a dynamic global vegetation model with managed land: Part I – Model description

Sibyll Schaphoff[1], Werner von Bloh[1], Anja Rammig[2], Kirsten Thonicke[1],
Hester Biemans[3], Matthias Forkel[4], Dieter Gerten[1,5], Jens Heinke[1],
Jonas Jägermeyr[1], Jürgen Knauer[6], Fanny Langerwisch[1], Wolfgang Lucht[1,5],
Christoph Müller[1], Susanne Rolinski[1], and Katharina Waha[1,7]

[1]Potsdam Institute for Climate Impact Research, PO Box 60 12 03, 14412 Potsdam, Germany
[2]Technical University of Munich, School of Life Sciences Weihenstephan, 85354 Freising, Germany
[3]Alterra, Wageningen University & Research, PO Box 47, 6700 AA Wageningen, the Netherlands
[4]TU Wien, Climate and Environmental Remote Sensing Group, Department of Geodesy and Geoinformation, Gusshausstraße 25-29, 1040 Wien, Austria
[5]Humboldt Universität zu Berlin, Department of Geography, Unter den Linden 6, 10099 Berlin, Germany
[6]Max Planck Institute for Biogeochemistry, Hans-Knöll-Str. 10, 07745 Jena, Germany
[7]CSIRO Agriculture & Food, 306 Carmody Rd, St Lucia QLD 4067, Australia

*Correspondence to:* Sibyll.Schaphoff@pik-potsdam.de

**Abstract.**

This paper provides a comprehensive description of the newest version of the Dynamic Global Vegetation Model with managed Land, LPJmL4. This model simulates - internally consistently - the growth and productivity of both natural and agricultural vegetation in direct coupling with water

and carbon fluxes. These features render LPJmL4 suitable for assessing a broad range of feedbacks within, and impacts upon, the terrestrial biosphere as increasingly shaped by human activities such as climate change and land-use change. Here we describe the core model structure including recently developed modules now unified in LPJmL4. Thereby we also summarize LPJmL model developments and evaluations (based on 34 earlier publications focused e.g. on improved representations

of crop types, human and ecological water demand, and permafrost) and model applications (82 papers, e.g. on historical and future climate change impacts) since its first description in 2007. To demonstrate the main features of the LPJmL4 model, we display reference simulation results for key processes such as the current global distribution of natural and managed ecosystems, their productivities, and associated water fluxes. A thorough evaluation of the model is provided in a companion

paper. By making the model source code freely available at a Gitlab server, we hope to stimulate the application and further development of LPJmL4 across scientific communities, not least in support of major activities such as the IPCC and SDG process.





## 1 Introduction

The terrestrial biosphere, a highly dynamic key component of the Earth system, is undergoing sig-
nificant and widespread transformations induced by human activities such as climate and land-use
change. Humans have by now transformed about 40% of the terrestrial ice-free land surface into land
used for agriculture and urban settlements (Ellis et al., 2010), thus pushing the planetary dynamics
beyond boundaries that have been characteristic for the past ca. 12,000 years (Rockström et al.,
2009). These interventions put at risk important functions of the biosphere such as the provisioning
of floral and faunal biodiversity, the terrestrial carbon sink (Le Quéré et al., 2015) and the provision-
ing of accessible freshwater. Understanding and modelling the current and potential future dynamics
of the Earth system thus renders it necessary to consider human activities as an integral part while
representing the major dynamics of the biosphere in a spatio-temporally explicit and process-based
manner, accounting for the feedbacks between vegetation, global carbon and water cycling, and the
atmosphere. This would also allow numerical evaluation of potential implementation pathways for
the United Nation's Sustainable Development Goals (SDG – https://sustainabledevelopment.un.org)
and their impacts on the terrestrial environment, complementing the important role that dynamic bio-
sphere models have played in the United Nation's scientific assessment reports on climate change
published by the United Nation's Intergovernmental Panel on Climate Change (noa, 2014).

By combining core features of global biogeographical and biogeochemical models developed in
the 1990s, Dynamic Global Vegetation Models (DGVMs) emerged as the main tool to simulate pro-
cesses underlying the dynamics of natural vegetation types (growth, mortality, resource competition,
disturbances such as wildfires) and the associated carbon and water fluxes (Cramer et al., 2001;
Prentice et al., 2007; Sitch et al., 2008; Friend et al., 2014). In light of the strengthening human
interferencces, DGVMs were further developed to integrate additional processes that are relevant
to the original research quest of studying biogeography and biogeochemical cycles under climate
change (Canadell et al., 2007). This includes the incorporation of human land-use and the simu-
lation of agricultural production systems (Bondeau et al., 2007; Lindeskog et al., 2013), nutrient
limitation (Zaehle et al., 2010; Smith et al., 2014), as well as hydrological modules and river routing
schemes (Gerten et al., 2004; Rost et al., 2008). Knowledge derived from models that are designed
to cover aspects of the earth system other than terrestrial vegetation and the carbon cycle, such as
models of the global water balance, could evidentially improve the DGVMs' skills (Bondeau et al.,
2007; Smith et al., 2014) and the ability to also evaluate model performance for processes (e.g. river
discharge) that are closely connected to the simulated vegetation and carbon cycle dynamics. The
development towards more comprehensive models of Earth's land surface offers new possibilities
for cross-disciplinary research.

DGVMs as land components of Earth system models still show large uncertainties about the ter-
restrial carbon (C) balance under future climate change (Friedlingstein et al., 2013). This uncertainty
partly results from differences in the simulation of soil and vegetation C residence times (Carval-



hais et al., 2014; Friend et al., 2014). The time that C resides in an ecosystem is thereby strongly affected by the simulated processes of vegetation dynamics (Ahlström et al., 2015). These examples highlight the need to continuously improve process representations in DGVMs in order to reduce the uncertainty in projected ecosystem functioning and services under future climate change. This requires however, that model developments in specific fields or improvements for certain processes are synthesized and integrated into a unified, internally consistent model version.

Recent developments focused on an improved energy balance model able to estimate permafrost dynamics based on a vertical soil carbon distribution scheme and a new soil hydrological scheme (Schaphoff et al., 2013). Also, a new process-based fire module (SPITFIRE) was implemented that allows for detailed simulation of fire ignition, spread and effects to estimate fire impacts and emissions (Thonicke et al., 2010). An updated phenology scheme was developed, which now takes phenology limitations arising from low temperatures, limited light and drought into account (Forkel et al., 2014). Further model developments encompass the paralellization of the model to efficiently simulate river routing (Von Bloh et al., 2010) and the implementation of irrigation scheme (Rost et al., 2008), recently updated with a mechanistic representation of the three major irrigation systems (Jägermeyr et al., 2015). Biemans et al. (2011) implemented reservoir operations and irrigation extraction and evaluated the impact on river discharge. Other developments focused on a newly formulated implementation of different cropping systems in sub-Saharan Africa (Waha et al., 2013), Mediterranean agricultural plant types (Fader et al., 2015) and bioenergy crops such as sugarcane (Lapola et al., 2009), fast-growing grasses and bioenergy trees (Beringer et al., 2011). With these implementations, the potential of bioenergy production under future land-use, population and climate development could be extensively investigated (Haberl et al., 2011; Popp et al., 2011; Humpenöder et al., 2014). All developments, the core model structure and recently developed modules of DGVM LPJmL version 4.0 (in the following referred to as LPJmL4) will be described in section 2 in more detail. We show that the model in its present form allows for consistent and joint quantification of climate and land-use change impacts on the terrestrial biosphere, the water cycle, the carbon cycle, and on agricultural production (a systematic evaluation can be found in Part II of this paper). To give an overview of recent developments and applications of LPJmL4, we present:

1. A comprehensive description of the full model with all contributing developments since its original publication by Sitch et al. (2003); Bondeau et al. (2007). We aim at consistently uniting all developments, including undocumented and already published developments, thus providing a comprehensive description of the full LPJmL4 model.

2. An overview over published LPJmL applications to review the improvement of process understanding.

3. A discussion of here presented standard LPJmL4 results that give an overview of simulated biogeochemical, hydrological and agricultural patterns at global scale.




## 2 Model description

The original Lund-Potsdam-Jena (LPJ) DGVM was described in detail by Sitch et al. (2003). This description and the associated model evaluation focused on the modelling of growth and geographical distribution of natural "plant functional types" (PFTs) and associated biogeochemical processes (mainly carbon cycling). Building on the improved representation of the water balance (Gerten et al., 2004). Bondeau et al. (2007) introduced the representation of "crop functional types" (CFTs) and evaluated the role of agriculture for the terrestrial carbon balance in particular. This model is since referred to as LPJmL (Lund-Potsdam-Jena managed Land) and provided the foundation for explicitly simulating agricultural production in a changing climate and for quantifying impacts of agricultural activities in assessments of the terrestrial carbon and water cycle.

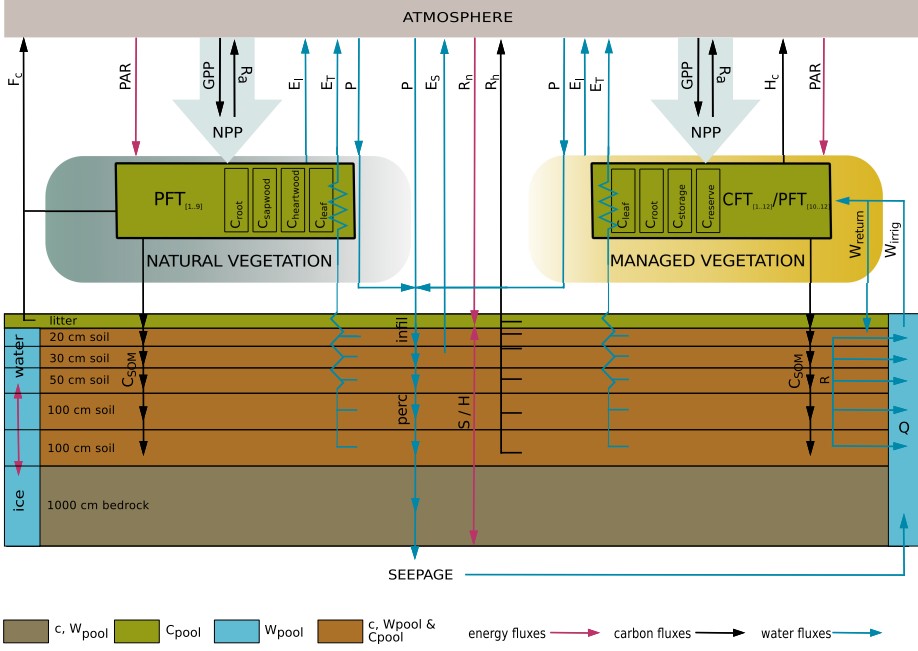

**Figure 1.** LPJmL4-scheme for Carbon, Water and Energy fluxes.

Since, a number of further specific model developments and applications have been published, but a comprehensive model description of all developments and amendments is missing. The parts of LPJmL4 building on (Bondeau et al., 2007) not only allow for quantifying changes in vegetation composition, the water cycle, the carbon cycle, and agricultural production, but also for explicitly simulating the dynamics and constraints within and among the modules, thereby providing a consistent and comprehensive representation of Earth's land surface processes. To demonstrate and make transparent the interplay of all these model features in the new LPJmL4 version, the present





paper documents the core model structure including equations and parameters from Sitch et al. (2003); Bondeau et al. (2007) and all more recent code developments. SI-Fig. 1 provides a schematic overview of the model structure and Fig. 1 of the simulated carbon, water and energy fluxes. The following sections describe the model components: energy balance model and permafrost (2.1), plant physiology (2.2), plant functional (2.3) and crop functional types (2.4), soil litter and carbon pools (2.5), water balance (2.6) and land use (2.7).

### 2.1 Energy balance model and permafrost

The energy balance model includes the calculation of photosynthetic active radiation, daylength and potential evapotranspiration (2.1.1) and albedo (2.1.2). The permafrost module is based on a new calculation of the soil energy balance (2.1.3). The energy balance model includes the calculation of photosynthetic active radiation, daylength and potential evapotranspiration (2.1.1) and albedo (2.1.2). The permafrost module is based on a new calculation of the soil energy balance (2.1.3).

#### 2.1.1 Photosynthetic active radiation, daylength and potential evapotranspiration

Photosynthetic active radiation (PAR) is the primary input to photosynthesis (2.2.1) and, thus, to the whole carbon cycle. Total daily PAR in mol m$^{-2}$ day$^{-1}$ is calculated as:

$$\text{PAR} = 0.5 \cdot c_q \cdot R_{s_{\text{day}}}, \tag{1}$$

where $c_q = 4.6 \times 10^{-6}$ is the conversion factor from J to mol for solar radiation at 550 nm. Half of the daily incoming solar irradiance $R_{s_{\text{day}}}$ is assumed to be PAR and atmospheric absorption to be the same for PAR and $R_{s_{\text{day}}}$ (Prentice et al., 1993; Haxeltine and Prentice, 1996).

Similar to the role of PAR for the carbon cycle, potential evapotranspiration (PET) is the primary driver of the water cycle. The calculation of both PAR and PET follows the approach of Prentice et al. (1993), where the calculation of potential evapotranspiration is based on the theory of equilibrium evapotranspiration $E_{\text{eq}}$ (Jarvis and McNaughton, 1986), given by:

$$E_{\text{eq}} = \frac{s}{s + \gamma} \cdot \frac{R_{n_{\text{day}}}}{\lambda}, \tag{2}$$

where $R_{n_{\text{day}}}$ is daily surface net radiation in J m$^{-2}$ day$^{-1}$ and $\lambda$ is the latent heat of vaporization in J kg$^{-1}$ with a weak dependence on air temperature (in °C):

$$\lambda = 2.495 \times 10^6 + 2380 \cdot T_{\text{air}} \tag{3}$$

$s$ is the slope of the saturation vapour pressure curve in Pa K$^{-1}$, given by

$$s = 2.502 \times 10^6 \cdot \frac{\exp[17.269 \cdot T_{\text{air}}/(237.3 + T_{\text{air}})]}{(237.3 + T_{\text{air}})^2} \tag{4}$$

and $\gamma$ is the psychrometric constant in Pa K$^{-1}$, given by

$$\gamma = 65.05 + 0.064 \cdot T_{\text{air}} \tag{5}$$





Following Priestley and Taylor (1972), PET is subsequently calculated from $E_{\mathrm{eq}}$ as:

$$\mathrm{PET} = \mathrm{pt} \cdot E_{\mathrm{eq}}, \tag{6}$$

where pt is the empirically derived Priestley-Taylor coefficient ($\mathrm{pt} = 1.32$).

The terrestrial radiation balance is written as

$$R_n = (1 - \beta) \cdot R_s + R_l, \tag{7}$$

where $R_n$ is net surface radiation; $R_s$ is incoming solar irradiance (downward) at the surface and $R_l$ the outgoing (upward positive) net long-wave radiation flux at the surface (both in W m$^{-2}$); $\beta$ is the short-wave reflection coefficient of the surface (albedo). The calculation of albedo depending on land surface conditions is described in section 2.1.2.

If not supplied directly as input variables to the model, the radiation terms $R_s$ and $R_l$ can be computed for any day and latitude at given cloudiness levels (input), following Prentice et al. (1993). $R_l$ can be approximated by a linear function of temperature and clear sky fraction:

$$R_l = (b + (1 - b) \cdot \mathrm{ni}) \cdot (A - T_{\mathrm{air}}), \tag{8}$$

where $b = 0.2$ and $A = 107$ are empirical constants. $T_{\mathrm{air}}$ is the mean daily air temperature in Celsius degrees, i.e. any effects of diurnal temperature variations are ignored. The net outgoing daytime long-wave flux $R_{l_{\mathrm{n_{day}}}}$ is obtained by multiplying with the length of the day in seconds:

$$R_{l_{\mathrm{n_{day}}}} = R_l \cdot \mathrm{daylength} \cdot 3600 \tag{9}$$

Instantaneous solar irradiance at the surface is computed from the solar constant, accounting for the proportion of bright sky ($\mathrm{ni} = 1 - \mathrm{cloudiness}$) and the angular distance between the sun's rays and the local vertical ($z$):

$$R_s = (c + d \cdot \mathrm{ni}) \cdot Q_0 \cdot \cos(z) \tag{10}$$

where $c = 0.25$ and $d = 0.5$ are empirical constants that together represent the clear-sky transmittivity (0.75). $Q_0$ is the solar constant, which is corrected for daily solar angle for day ($i$) as

$$Q_0 = Q_{00} \cdot (1 + 2 \cdot 0.01675 \cdot \cos(2 \cdot \pi \cdot i/365)) \tag{11}$$

and the solar zenith angle ($z$) correction, computed from the solar declination ($\delta$), i.e. the angle between the orbital plane and the Earth's equatorial plane), which varies between $+23.4°$ in northern hemisphere midsummer and $-23.4°$ in northern hemisphere midwinter, the latitude (lat, in radians) and the hour angle $h$, i.e. the fraction of $2 \cdot \pi$ (in radians) which the earth has turned since the local solar noon, $Q_{00} = 1360$ W m$^{-2}$.

$$\cos(z) = \sin(\mathrm{lat}) \cdot \sin(\delta) + \cos(\mathrm{lat}) \cdot \cos(\delta) \cdot \cos(h) \tag{12}$$





with

$$\delta = -23.4 \cdot \pi/180 \cdot \cos(2 \cdot \pi \cdot (i+10)/365) \tag{13}$$

To obtain the $R_{s_{\text{day}}}$, eq. (10) needs to be integrated from sunrise to sunset, i.e. from $-h_{1/2}$ to $h_{1/2}$, where $h_{1/2}$ is the half-day length in angular units, computed as:

$$h_{1/2} = \arccos\left(-\frac{\sin(\text{lat}) \cdot \sin(\delta)}{\cos(\text{lat}) \cdot \cos(\delta)}\right) \tag{14}$$

thus

$$R_{s_{\text{day}}} = (c + d \cdot \text{ni}) \cdot Q_0 \cdot (\sin(\text{lat}) \cdot \sin(\delta) \cdot h_{1/2}$$
$$+ \cos(\text{lat}) \cdot \cos(\delta) \cdot h_{1/2}) \tag{15}$$

The duration of sunshine of a single day (daylength in hours) is computed as:

$$\text{daylength} = 24 \cdot \frac{h_{1/2}}{\pi} \tag{16}$$

### 2.1.2 Albedo

Albedo ($\beta$), the average reflectivity of the grid cell, was first implemented by Strengers et al. (2010) and later improved by considering several drivers of phenology as in Forkel et al. (2014).

$$\beta = \sum_{\text{PFT}=1}^{n_{\text{PFT}}} \beta_{\text{PFT}} \cdot \text{FPC} + F_{\text{bare}} \cdot (F_{\text{snow}} \cdot \beta_{\text{snow}} + (1 - F_{\text{snow}}) \cdot \beta_{\text{soil}}) \tag{17}$$

$\beta$ depends on land surface condition and is based on a combination of defined albedo values for bare soil ($\beta_{\text{soil}} = 0.3$), snow ($\beta_{\text{snow}} = 0.7$ average value taken from Liang et al. (2005); Malik et al. (2012)) and plant compartments specific albedo values, where vegetation albedo ($\beta_{\text{PFT}}$) is simulated as the albedo of each existing PFT $\beta_{\text{PFT}}$. $\text{FPC}_{\text{PFT}}$ is the foliage projective cover of the respective PFT (see eq. (55)) Parameters ($\beta_{\text{leaf,PFT}}$) were taken as suggested by Strugnell et al. (2001) (see SI-Table 3). Parameters $\beta_{\text{stem,PFT}}$ and $\beta_{\text{litter,PFT}}$ were obtained from Forkel et al. (2014) who optimized these parameters by using MODIS albedo time series. $F_{\text{snow}}$ and $F_{\text{bare}}$ are the snow coverage and the fraction of bare soil, respectively (Strengers et al., 2010).

### 2.1.3 Soil energy balance

The newly implemented calculation of the soil energy balance as described in Schaphoff et al. (2013) marks a new development and differs markedly from previous implementations of permafrost modules in LPJ (Beer et al., 2007). Soil temperatures ($T_{\text{soil}}$) for each layer are computed with an energy balance model, including one-dimensional heat conduction and convection of latent heat. Freezing and thawing has been added to better account for soil ice dynamics. Soil water dynamics are computed daily (see section 2.6). The soil column is divided into five hydrological active layers of 0.2,



0.3, 0.5, 1 and 1 m depth ($\Delta z$) (see section 2.6.1). For a thermal buffer we assume an additional layer of 10 m thickness, which is only thermally active. Soil parameters for thermal diffusivity ($\mathrm{m^2\,s^{-1}}$)

at wilting point, at 15% of water holding capacity, and at field capacity and for thermal conductivity ($\mathrm{W\,m^{-1}\,K^{-1}}$) at wilting point, and at saturation (for water and ice) are derived for each grid cell using soil texture from the Harmonized World Soil Database (HWSD) version (Nachtergaele et al., 2008). Relationships between texture and thermal properties are taken from Lawrence and Slater (2008). The one-dimensional heat conduction equation is:

$$\frac{\partial T_{\mathrm{soil}}}{\partial t} = \alpha \cdot \frac{\partial^2 T_{\mathrm{soil}}}{\partial z^2}, \tag{18}$$

where $\alpha = \lambda/c$ is thermal diffusivity, $\lambda$ thermal conductivity, $c$ heat capacity (in $\mathrm{J\,m^{-3}\,K^{-1}}$). $T_{\mathrm{soil}}$ at position $z$ and time $t$ is solved in its finite difference form following Bayazıtoğlu, Yıldız and Özişik, M. Necati. (1988):

$$\frac{T_{\mathrm{soil}(t+1,l)} - T_{\mathrm{soil}(t,l)}}{\Delta t} = \alpha \cdot \frac{T_{\mathrm{soil}(t,l-1)} + T_{\mathrm{soil}(t,l+1)} - 2T_{\mathrm{soil}(t,l)}}{(\Delta z)^2} \tag{19}$$

for soil layers $l$, including a snow layer, and time step $t$ with the following boundary conditions:

$$T_{\mathrm{soil}(t=1,l=1)} = T_{\mathrm{air}}, \tag{20}$$

$$T_{\mathrm{soil}(t,l=n_{\mathrm{soil}}+1)} = T_{\mathrm{soil}(t,l=n_{\mathrm{soil}})}, \tag{21}$$

where $n_{\mathrm{soil}} = 6$ is the number of soil layers. We assume a heatflux of zero below the lowest soil layer, i.e. below 13 m depth. The largest possible, numerically still stable time step $\Delta t$ is calculated

depending on $\Delta z$ and soil thermal diffusivity $\alpha$ (Bayazıtoğlu, Yıldız and Özişik, M. Necati., 1988), which gives the stability criterion ($r$) for the finite-difference solution:

$$r = \frac{\alpha \Delta t}{(\Delta z)^2}. \tag{22}$$

For numerical stability $(1 - 2r)$ needs to be $> 0$, so that $r \leq 0.5$ as $\Delta z$ is given from soil depth and $\alpha$ can be calculated from soil properties. The maximum stable $\Delta t$ can be calculated:

$$\Delta t \leq \frac{(\Delta z)^2}{2 \cdot \alpha} \tag{23}$$

and therefore eq. (19) becomes:

$$T_{\mathrm{soil}(t+1,l)} = r \cdot \left( T_{\mathrm{soil}(t,l-1)} + T_{\mathrm{soil}(t,l+1)} + (1-2r)T_{\mathrm{soil}(t,l)} \right) \tag{24}$$

For the diurnal temperature range after Parton and Logan (1981), at least 4 time steps per day are calculated and the maximum time steps is set to 40 per day. Heat capacity ($c$) of the soil is calculated

as the sum of the volumetric-specific heat capacities (in $\mathrm{J\,m^{-3}\,K^{-1}}$)) of soil minerals ($c_{\mathrm{min}}$), soil water content ($c_{\mathrm{water}}$) and soil ice content ($c_{\mathrm{ice}}$) and their corresponding shares ($m$, in $\mathrm{m^3}$) of the soil bucket:

$$c = c_{\mathrm{min}} \cdot m_{\mathrm{min}} + c_{\mathrm{water}} \cdot m_{\mathrm{water}} + c_{\mathrm{ice}} \cdot m_{\mathrm{ice}} \tag{25}$$





The heat capacity of air is neglected because of its comparatively low contribution to overall heat
capacity. Thermal conductivity ($\lambda$) is calculated following Johansen (1977). Sensible and latent heat
fluxes are calculated explicitly for the snow layer by assuming a constant snow density of $0.3 \, \text{t m}^{-3}$
and the resulting thermal diffusivity of $3.17 \times 10^{-7} \, \text{m}^2 \, \text{s}^{-1}$. Sublimation is assumed to be $0.1$ mm
$\text{day}^{-1}$.

The active layer thickness represents the depth of maximum thawing of the year. Freezing depth
is calculated by assuming that the fraction of frozen water is congruent with the frozen soil bucket.
The $0°\text{C}$–isotherm within a layer is estimated by assuming a linear temperature gradient within the
layer and this fraction of heat is assumed to be used for the thawing respectively freezing process.
Temperature represents the amount of thermal energy available, whereas heat transport represents the
movement of thermal energy into the soil by rain and melt water. Precipitation and percolation energy
and the amount of energy which arises from the temperature difference between the temperature of
the above layer (or the air temperature for the upper layer) and the temperature of the below layer,
are assumed to be used for converting latent heat fluxes first. The residual energy is used to increase
soil temperature. $T_{\text{soil}}$ is initialized at the beginning of the spinup simulation by the mean annual air
temperature.

## 2.2 Plant physiology

### 2.2.1 Photosynthesis

The LPJmL4 photosynthesis model is a 'big leaf' model, the leaf biochemical model of photosynthe-
sis developed by Farquhar et al. (1980) and Farquhar and von Caemmerer (1982). These assumptions
have been generalized by Collatz et al. (1991, 1992) for global modelling applications and for the
stomatal response. Most details are as in Sitch et al. (2003) but a summary is provided in the fol-
lowing. The 'strong optimality' hypothesis (Haxeltine and Prentice, 1996; Prentice et al., 2000) is
applied by assuming that Rubisco activity and the nitrogen content of leaves vary with canopy po-
sition and seasonally such as to maximize net assimilation at the leaf level. Most details are as in
(Sitch et al., 2003) but a summary is provided in the following.

In LPJmL4, photosynthesis is simulated as a function of absorbed photosynthetically active radi-
ation (APAR), temperature, daylength, and canopy conductance, for each PFT or CFT present in a
grid cell and at daily time steps. APAR is calculated as the fraction of incoming net photosyntheti-
cally active radiation (PAR, see eq. (1)) that is absorbed by green vegetation (FAPAR):

$$\text{APAR}_{\text{PFT}} = \text{PAR} \cdot \text{FAPAR}_{\text{PFT}} \cdot \alpha_{a_{\text{PFT}}} \quad (26)$$





$\alpha_{a_{PFT}}$ is a scaling factor to scale leaf-level photosynthesis in LPJmL4 to biome level respectively agricultural stand level. The PFT-specific FAPAR$_{PFT}$ is calculated as follows:

$$\text{FAPAR}_{PFT} = \text{FPC}_{PFT} \quad \cdot \quad \Big( (\text{phen}_{PFT} - F_{SnowGC}) \cdot (1 - \beta_{\text{leaf,PFT}}) \tag{27}$$
$$- (1 - \text{phen}_{PFT}) \cdot c_{\text{fstem}} \cdot \beta_{\text{stem,PFT}} \Big),$$

where phen$_{PFT}$ is the daily phenological status (ranging between 0 and 1) representing the fraction of full leaf coverage currently attained by the PFT, reduced by the green-leaf albedo $\beta_{\text{leaf,PFT}}$, the stem albedo $\beta_{\text{stem,PFT}}$ (for trees), and $F_{SnowGC}$ is the fraction of snow in the green canopy. $c_{\text{fstem}} = 0.7$ is the masking of the ground by stems and branches without leaves (Strengers et al., 2010).

Based on this, gross photosynthesis rate $A_{gd}$ is computed as the minimum of two functions (details in Haxeltine and Prentice (1996)):

1. The light-limited photosynthesis rate $J_E$ (mol C m$^{-2}$ hour$^{-1}$)

$$J_E = C_1 \cdot \frac{\text{APAR}}{\text{daylength}}, \tag{28}$$

where for C3-Photosynthesis

$$C_1 = \alpha_{C3} \cdot T_{\text{stress}} \cdot \left( \frac{p_i - \Gamma_*}{p_i + \Gamma_*} \right) \tag{29}$$

and for C4-Photosynthesis

$$C_1 = \alpha_{C4} \cdot T_{\text{stress}} \cdot \left( \frac{\lambda}{\lambda_{\max_{C4}}} \right). \tag{30}$$

$p_i$ is the leaf internal partial pressure of CO$_2$ given by $p_i = \lambda \cdot p_a$, where $\lambda$ gives the relation of internal and ambient pressure ($p_a$), which reflects soil-plant water interaction (see eq. (39)). $T_{\text{stress}}$ is the PFT-specific temperature inhibition function, which limits photosynthesis at high and low temperatures. $\alpha_{C3}$ and $\alpha_{C4}$ are the intrinsic quantum efficiencies for CO$_2$ uptake in C3 and C4 plants respectively and $\Gamma_*$ is the photorespiratory CO$_2$ compensation point.

$$\Gamma_* = \frac{[\text{O}_2]}{2 \cdot \tau}, \tag{31}$$

where $\tau = \tau_{25} \cdot q_{10_\tau}^{(T_{\text{air}} - 25) \cdot 0.1}$ is the specificity factor, it reflects the ability of Rubisco to discriminate between CO$_2$ and O$_2$. [O$_2$] is the partial pressure of O$_2$ and $\tau_{25}$ is the $\tau$ value at 25°C and $q_{10_\tau}$ is the temperature sensitivity parameter.

2. The Rubisco-limited photosynthesis rate $J_C$ (mol C m$^{-2}$ hour$^{-1}$).

$$J_C = C_2 \cdot V_m, \tag{32}$$

where $V_m$ is the maximum Rubisco capacity (see eq. (35)) and

$$C_2 = \frac{p_i - \Gamma_*}{p_i + K_C \left( 1 + \frac{[\text{O}_2]}{K_O} \right)} \tag{33}$$





$K_C$ and $K_O$ representing the Michaelis-Menten constants. Daily gross photosynthesis $A_{\mathrm{gd}}$ is then given by:

$$A_{\mathrm{gd}} = \left( J_E + J_C - \sqrt{(J_E + J_C)^2 - 4 \cdot \theta \cdot J_E \cdot J_C} \right)$$
$$/(2 \cdot \theta) \cdot \mathrm{daylength} \tag{34}$$

The shape parameter $\theta$ describes the co-limitation of light and Rubisco activity (Haxeltine and Prentice, 1996). Subtracting leaf respiration ($R_{\mathrm{leaf}}$, given in eq. (45)), gives the daily net photosynthesis ($A_{\mathrm{nd}}$), so that $V_m$ is included in $J_C$ and $R_{\mathrm{leaf}}$. To calculate optimal $A_{\mathrm{nd}}$, the zero point of the first derivative is calculated (i.e. $\partial A_{\mathrm{nd}}/\partial V_m \equiv 0$). The thus derived maximum Rubisco capacity $V_m$ is:

$$V_m = \frac{1}{b} \cdot \frac{C_1}{C_2} \left( (2 \cdot \theta - 1) \cdot s - (2 \cdot \theta \cdot s - C_2) \cdot \sigma \right) \cdot \mathrm{APAR} \tag{35}$$

with

$$\sigma = \sqrt{1 - \frac{C_2 - 2}{C_2 - \theta s}} \text{ and } s = 24/\mathrm{daylength} \cdot b \tag{36}$$

The daily net daytime photosynthesis is given by subtracting dark respiration $R_d = (1 - \mathrm{daylength}/24) \cdot R_{\mathrm{leaf}}$ (see eq. (45)):

$$A_{\mathrm{dt}} = A_{\mathrm{nd}} - R_d \tag{37}$$

The photosynthesis rate can be related to canopy conductance through the $CO_2$ diffusion gradient between the intercellular air spaces and the atmosphere:

$$g_c = \frac{1.6 A_{\mathrm{dt}}}{p_a \cdot (1 - \lambda)} + g_{\mathrm{min}}, \tag{38}$$

where $g_{\mathrm{min}}$ is a PFT-specific minimum canopy conductance scaled by FPC that occurs due to processes other than photosynthesis, $\lambda$ is a parameter describing the ratio of the intercellular to the ambient $CO_2$ concentration and $p_a$ is the ambient partial pressure of $CO_2$. Combining both methods determining $A_{\mathrm{dt}}$ gives:

$$0 = A_{\mathrm{dt}} - A_{\mathrm{dt}} = A_{\mathrm{nd}} + (1 - \mathrm{daylength}/24) \cdot R_{\mathrm{leaf}}$$
$$- p_a \cdot (g_c - g_{\mathrm{min}}) \cdot (1 - \lambda)/1.6 \tag{39}$$

This equation has to be solved for $\lambda$ which is not possible analytically because of the occurrence of $\lambda$ in $A_{\mathrm{nd}}$ and in the second term of the equation. Therefore a numerical bisection algorithm is used to obtain $\lambda$ solving the equation. The actual canopy conductance is calculated as a function of water stress depending on the soil moisture status (section 2.6.2) and thus the photosynthesis rate is related to actual canopy conductance. All parameter values are given in SI-Table 4.





### 2.2.2 Phenology

The phenology module of tree and grass PFTs is based on the growing season index (GSI) approach (Jolly et al., 2005). Thereby the continuous development of canopy greenness is modelled based on empirical relations to temperature, day length and drought conditions. The GSI approach was modified for its use in LPJmL (Forkel et al., 2014) so that it accounts for the limiting effects of cold temperature, light, water availability and heat stress on the daily phenology status $\text{phen}_{\text{PFT}}$:

$$\text{phen}_{\text{PFT}} = f_{\text{cold}} \cdot f_{\text{light}} \cdot f_{\text{water}} \cdot f_{\text{heat}} \tag{40}$$

Each limiting function can range between 0 (full limitation of leaf development) and 1 (no limitation of leaf development). The limiting functions are defined as logistic functions and depend also on the previous day's value:

$$f(x)_t = f(x)_{t-1} + (1/(1 + \exp(\text{sl}_x \cdot (x - b_x)) - f(x)_{t-1}) \cdot \tau_x, \tag{41}$$

where $x$ is daily air temperature for the cold and heat stress-limiting functions $f_{\text{cold}}$ and $f_{\text{heat}}$, respectively, and stands for, short-wave downward radiation in the light-limiting function $f_{\text{light}}$, and water availability for the water-limiting function $f_{\text{water}}$. The parameters $b_x$ and $\text{sl}_x$ are the inflection point and slope of the respective logistic function; $\tau_x$ is a change rate parameter that introduces a time-lagged response of the canopy development to the daily meteorological conditions. The em-

pirical parameters were estimated by optimizing LPJmL simulations of FAPAR against 30 years of satellite-derived time series of FAPAR (Forkel et al., 2014).

### 2.2.3 Productivity

#### Autotrophic respiration

Autotrophic respiration is separated into carbon costs for maintenance and growth and is calculated

as in Sitch et al. (2003). Maintenance respiration depends on tissue-specific C:N ratios (for above-$\text{CN}_{\text{sapwood}}$ and below-ground tissues $\text{CN}_{\text{root}}$). It further depends on temperature ($T$), either air temperature ($T_{\text{air}}$) for above- and soil temperatures ($T_{\text{soil}}$) for below-ground tissues, on tissue biomass ($C_{\text{sapwood}}$ resp. $C_{\text{root}}$) and phenology ($\text{phen}_{\text{PFT}}$, see eq. (40)) and is calculated at a daily timestep as follows:

$$R_{\text{sapwood}} = r_{\text{PFT}} \cdot k \cdot \frac{C_{\text{sapwood}}}{\text{CN}_{\text{sapwood}}} \cdot g(T_{\text{air}}) \tag{42}$$

$$R_{\text{root}} = r_{\text{PFT}} \cdot k \cdot \frac{C_{\text{root}}}{\text{CN}_{\text{root}}} \cdot g(T_{\text{soil}}) \cdot \text{phen}_{\text{PFT}} \tag{43}$$

The respiration rate ($r_{\text{PFT}}$, in $\text{gC gN}^{-1}\text{day}^{-1}$) is a PFT-specific parameter on a 10°C base to represent acclimation of respiration rates to average conditions (Ryan, 1991): $k$ refers to the value proposed by (Sprugel et al., 1995).



The temperature function $g(T)$, describing the influence of temperature on maintenance respiration, is defined as:

$$g(T) = \exp\left[308.56 \cdot \left(\frac{1}{56.02} - \frac{1}{(T+46.02)}\right)\right] \tag{44}$$

Eq. (44) is a modified Arrhenius equation (Lloyd and Taylor, 1994), where $T$ is either air or soil temperature (°C). This relationship is described by Tjoelker et al. (1999) for a consistent decline of autotrophic respiration with temperature.

While leaf respiration ($R_{\text{leaf}}$) depends on $V_m$ (see eq. (35)) with a static parameter $b$ depending on photosynthetic pathway:

$$R_{\text{leaf}} = V_m \cdot b \tag{45}$$

Gross primary production (GPP, calculated by eq. (34) and converted to $\text{gC}\,\text{m}^{-2}\,\text{day}^{-1}$) is reduced by maintenance respiration. Growth respiration, the carbon costs for producing new tissue, is assumed to be 25% of the remainder. The residual is the annual net primary production (NPP):

$$\text{NPP} = (1 - r_{\text{gr}}) \cdot (\text{GPP} - R_{\text{leaf}} - R_{\text{sapwood}} - R_{\text{root}}), \tag{46}$$

where $r_{\text{gr}} = 0.25$ is the share of growth respiration.

**Reproduction cost**

As in Sitch et al. (2003), a fixed fraction of 10% of annual NPP is assumed to be carbon costs for producing reproductive organs and propagules in LPJmL4. Since only a very small part of the carbon allocated to reproduction finally enters the next generation, the reproductive carbon allocation is added to the above-ground litter pool to preserve a closed carbon balance in the model.

**Tissue turnover**

As in Sitch et al. (2003), a PFT-specific tissue turnover rate is assigned to the living tissue pools (SI-Table 6 and Fig. 1). Leaves and fine roots are transferred to litter and living sapwood to heartwood. Root turnover rates are calculated on a monthly basis, and the conversion of sapwood to heartwood annually. Leaf turnover rates depend on the phenology of the PFT: it is calculated at leaf fall for deciduous and daily for evergreen PFTs.

## 2.3 Plant functional types

Vegetation composition is determined by the fractional coverage of populations of different plant functional types (PFTs). PFTs are defined to account for the variety of structure and function among plants (Smith et al., 1993). In LPJmL4 11 PFTs are defined, of which eight are woody (two tropical, three temperate, three boreal) and three are herbaceous (Tables 1). PFTs are simulated in LPJmL4



as average individuals. Woody PFTs are characterised by the population density and the state variables: crown area (CA) and the size of four tissue compartments: leaf mass ($C_{leaf}$), fine root mass ($C_{root}$), sapwood mass ($C_{sapwood}$), and heartwood mass ($C_{heartwood}$). The size of all state variables is averaged across the modelled area. The state variables of grasses are represented only by the leaf and root compartments. The physiological attributes and bioclimatic limits controlling the dynamics

of the PFT (see SI-Table 2). PFTs are located in one stand per grid cell and as such compete for light and soil water. That means their crown area and leaf area index determines their capacity to absorb photosynthetic active radiation for photosynthesis (see section 2.2.1) and their rooting profiles determine the access to soil water influencing their productivity (see section 2.6.2). In the following, we describe, how carbon is allocated to the different tissue compartments of a PFT (2.3.1) and veg-

etation dynamics (2.3.2), i.e. how the different PFTs interact. The vegetation dynamics component of LPJmL4 includes the simulation of establishment and different mortality processes.

### 2.3.1 Allocation

The allocation of carbon is simulated as described in Sitch et al. (2003) and all parameter values are given in SI-Table 4. The assimilated amount of carbon (the remaining NPP) constitutes the annual

woody carbon increment which is allocated to leaves, fine roots and sapwood such that four basis allometric relationships (eq. (47) - (50)) are satisfied. The pipe model from Shinozaki et al. (1964); Waring et al. (1982) prescribes that each unit of leaf area must be accompanied by a corresponding area of transport tissue (described by the parameter $k_{la:sa}$) and, the sapwood cross-sectional area (SA):

$$LA = k_{la:sa} \cdot SA, \tag{47}$$

where LA is the average individual leaf area.

A functional balance exists between investment in fine root biomass and investment in leaf biomass, respectively. Carbon allocation to $C_{leaf}$ is determined by the maximum leaf-to-root mass ratio $lr_{max}$ (SI-Table 6), which is a constant and by $\omega$ a water stress index (Sitch et al., 2003), which stands for

that under water-limited conditions, plants are modelled to allocate relatively more carbon to fine root biomass.

$$C_{leaf} = lr_{max} \cdot \omega \cdot C_{root} \tag{48}$$

The relation between tree height ($H$) and stem diameter ($D$) is given as in Huang et al. (1992).

$$H = k_{allom2} \cdot D^{k_{allom3}} \tag{49}$$

The crown area (CA) to stem diameter ($D$) relation is based on inverting Reinecke's rule (Zeide, 1993) $k_{rp}$ as the Reineke parameter:

$$CA = \min(k_{allom1} \cdot D^{k_{rp}}, CA_{max}), \tag{50}$$





which relates tree density to stem diameter under self-thinning conditions. $CA_{max}$ is the maximum crown area allowed. The reversal used in LPJmL4 gives the expected relation between stem diameter and crown area. The assumption here is a closed canopy, but no crown overlap.

By combining the allometric relations of eq. (47) - (50) it follows that the relative contribution of sapwood respiration increases with height, which restricts the possible height of trees.

Assuming cylindrical stems and constant wood density (WD), $H$ can be computed and is related to SA (see eq. (47)). From this follows:

$$H = \frac{C_{\text{sapwood}} \cdot k_{\text{la:sa}}}{\text{WD} \cdot C_{\text{leaf}} \cdot \text{SLA}} \tag{51}$$

Stem diameter can then be calculated by inverting eq. (49). Leaf area is related to leaf biomass $C_{\text{leaf}}$ by PFT-specific SLA, thus, the individual leaf area index ($LAI_{\text{ind}}$) is given by:

$$\text{LAI}_{\text{ind}} = \frac{C_{\text{leaf}} \cdot \text{SLA}}{\text{CA}} \tag{52}$$

SLA is related to leaf longevity ($\alpha_{\text{leaf}}$) in month (see SI-Table 6), which determines whether deciduous or evergreen phenology suits a given climate suggested by Reich et al. (1997). The equation is based on the form suggested by Smith et al. (2014) for needleleaved and broadleaved PFTs as follows:

$$\text{SLA} = \frac{2 \times 10^{-4}}{\text{DM}_C} \cdot 10^{\beta_0 - \beta_1 \cdot \log(\alpha_{\text{leaf}})/\log(10)} \tag{53}$$

The parameter $\beta_0$ is adapted for broadleaved ($\beta_0 = 2.2$) and needleleaved trees ($\beta_0 = 2.08$) and for grass ($\beta_0 = 2.25$) and $\beta_1$ is set to $0.4$. Both parameters were derived from data given in Kattge et al. (2011). The dry matter carbon content of leaves $\text{DM}_C$ is set to $0.4763$ obtained from Kattge et al. (2011). $LAI_{\text{ind}}$ can be converted into foliar projective cover ($FPC_{\text{ind}}$, which is the proportion of ground area covered by leaves) using the canopy light-absorption model (Lambert-Beer law, Monsi (1953)):

$$\text{FPC}_{\text{ind}} = 1 - \exp(-k \cdot \text{LAI}_{\text{ind}}), \tag{54}$$

where $k$ is the PFT specific light extinction coefficient (see SI-Table 3). The overall FPC of a PFT in a grid cell is obtained by the product $LAI_{\text{ind}}$, mean individual CA, and mean number of individuals per unit area ($P$), which is determined by the vegetation dynamics (see section 2.3.2).

$$\text{FPC} = \text{CA} \cdot P \cdot \text{FPC}_{\text{ind}} \tag{55}$$

FPC directly measures the ability of the canopy to intercept radiation (Haxeltine and Prentice, 1996).





### 2.3.2 Vegetation dynamics

**Establishment**

For PFTs within their bioclimatic limits ($T_{c,min}$, see SI-Table 2), each year, new woody PFT in-
dividuals and herbaceous PFTs can establish depending on available space. Woody PFTs have a
maximum establishment rate $k_{est}$ of $0.12$ saplings m$^{-2}$, which is a medium value of tree density for
all biomes (Luyssaert et al., 2007). New saplings can establish on bare ground in the grid cell that is
not occupied by woody PFTs. The number of new saplings per unit area is proportional to $k_{est}$ and
to the FPC of each PFT present in the grid cell. It declines in proportion to canopy light attenuation
when the sum of woody FPCs exceeds $0.95$, thus simulating a decline in establishment success with
canopy closure (Prentice et al., 1993). Establishment increases the number of individuals per unit
area ($P$), the population density.

**Background mortality**

Mortality is modelled by a fractional reduction of $P$. Mortality always leads to a reduction in biomass
per unit area. Similar as in Sitch et al. 2003, a background mortality rate (mort$_{greff}$), the inverse of
mean PFT longevity, is applied from the yearly growth efficiency (greff) (Waring, 1983) expressed
as the ratio of net biomass increment to leaf area:

$$\text{mort}_{greff} = \frac{k_{mort1}}{1 + k_{mort2} \cdot greff}, \tag{56}$$

where $k_{mort1}$ is an asymptotic maximum mortality rate, and $k_{mort2}$ is a parameter governing the
slope of the relationship between growth efficiency and mortality (SI-Table 4).

**Stress mortality**

Mortality from competition occurs when tree growth leads to too high tree densities (FPC of all trees
exceeds $> 95\%$). In this case, all tree PFTs are reduced proportionally to their expansion. Herbaceous
PFTs are outcompeted by expanding trees until these reach their maximum FPC of $95\%$ or by light
competition between herbaceous PFTs. Dead biomass is transferred to the litter pools.

Boreal trees can die from heat stress mort$_{heat}$ occurs when a temperature threshold of $23°C$ is
exceeded, but only for boreal trees. Temperatures above this threshold are accumulated over the year
(gdd$_{tw}$) and this is related to a parameter value of the heat damage function (tw$_{PFT}$), which is set to
$400$:

$$\text{mort}_{heat} = \min\left(\frac{gdd_{tw}}{tw_{PFT}}, 1\right) \tag{57}$$

**Fire disturbance and mortality**

Two different fire modules can be applied in the LPJmL4 model: the simple Glob-FIRM model
(Thonicke et al., 2001) and the process-based SPITFIRE model (Thonicke et al., 2010). In Glob-





FIRM, fire disturbances are calculated as an exponential probability function dependent on soil

moisture in the top 50 cm and a fuel load threshold. The sum of the daily probability determines
the length of the fire season. Burnt area is assumed to increase nonlinearly with increasing length of
fire season. The fraction of trees killed within the burnt area depends on a PFT-specific fire resistance
parameter for woody plants, while all litter and live grasses are consumed by fire. Glob-FIRM does
not specify fire ignition sources and assumes a constant relationship between fire season length and

resulting burnt area. The PFT-specific fire resistance parameter implies that fire severity is always
the same, an approach suitable for model applications to multi-century time scales or paleo-climate
conditions.

In SPITFIRE, fire disturbances are simulated as the fire processes risk, ignition, spread and effects
separately. The climatic fire danger is based on the Nesterov index $\mathrm{NI}(N_d)$ [°C], which describes

atmospheric conditions critical to fire risk for day $N_d$:

$$\mathrm{NI}(N_d) = \sum_{\substack{N_d \\ \text{if } \mathrm{Pr}(d) \leq 3\ \mathrm{mm}}} T_{\max}(d) \cdot \big(T_{\max}(d) - T_{\mathrm{dew}}(d)\big), \tag{58}$$

where $T_{\max}$ and $T_{\mathrm{dew}}$ are the daily maximum and dew-point temperature, and $d$ is a positive temperature day with a precipitation of less than 3 mm. The probability of fire spread $P_{\mathrm{spread}}$ decreases linearly as litter moisture $\omega_0$ increases towards its moisture of extinction $m_e$:

$$P_{\mathrm{spread}} = \begin{cases} 1 - \omega_0/m_e, & \omega_0 \leq m_e \\ 0, & \omega_0 > m_e \end{cases} \tag{59}$$

Combining NI and $P_{\mathrm{spread}}$ we can calculate the fire danger index FDI:

$$\mathrm{FDI} = \max\left\{0, 1 - \frac{1}{m_e} \cdot \exp\left(-\mathrm{NI} \cdot \sum_{p=1}^{n} \frac{\alpha_p}{n}\right)\right\} \tag{60}$$

to interpret the qualitative fire risk in quantitative terms. The value of $\alpha_p$ defines the slope of the probability risk function given as the average PFT parameter (see SI-Table 7) for all existing PFTs (n).

SPITFIRE considers human-caused and lightning-caused fires as sources for fire ignition. Lightning-caused ignition rates are prescribed from the OTD/LIS satellite product (Christian et al., 2003). Since it quantifies total flash rate, we assume that 20% of these are cloud-to-ground flashes (Latham and Williams, 2001) and that, under favourable burning conditions, their effectiveness to start fires is $0.04$ (Latham and Williams, 2001; Latham and Schlieter, 1989). Human-caused ignitions are modelled

as a function of human population density assuming that ignition rates are higher in remote regions
and declines with increasing level of urbanisation and associated effects of landscape fragmentation,
infrastructure and improved fire monitoring. The function is:

$$n_{h,\mathrm{ig}} = P_D \cdot k(P_D) \cdot a(N_D)/100, \tag{61}$$

where

$$k(P_D) = 30.0 \cdot \exp(-0.5 \cdot \sqrt{P_D}). \tag{62}$$





$P_D$ is the population density [individuals $\mathrm{km}^{-2}$], and $a(N_D)$ [ignitions individual$^{-1}$ day$^{-1}$] is a parameter describing the inclination of humans to use fire and cause fire ignitions. In absence of further information $a(N_D)$ can be calculated from fire statistics using the following approach

$$a(N_D) = \frac{N_{h,\mathrm{obs}}}{t_{\mathrm{obs}} \cdot \mathrm{LFS} \cdot \overline{P_D}},$$ (63)

where $N_{h,\mathrm{obs}}$ is the average number of human-caused fires observed during the observation years $t_{\mathrm{obs}}$ in a region with the average length of fire season (LFS) and the mean human population density. Assuming that all fires ignited in one day have the same burning conditions in a $0.5°$ grid cell with the grid cell size $A$, we combine fire danger, potential ignitions and the mean fire area $A_f$ to obtain daily total burnt area with:

$$A_b = \min(E(n_{\mathrm{ig}}) \cdot \mathrm{FDI} \cdot A_f, A)$$ (64)

We calculate $E(n_{\mathrm{ig}})$ with the sum of independent estimates of numbers of lightning ($n_{l,\mathrm{ig}}$) and human-caused ignition events ($n_{h,\mathrm{ig}}$), disregarding stochastic variations. $A_f$ is calculated from forward and backward rate of spread which depends on the dead fuel characteristics, fuel load in the respective dead fuel classes and wind speed. Dead plant material entering the litter pool is subdivided

into 1-, 10-, 100- and 1000 hour fuel classes, describing the amount of time to dry a fuel particle of a specific size (1-hour fuel refers to leaves and twigs and 1000 hour fuel to tree boles). As described by Thonicke et al. (2010): "The forward rate of spread ROS$_{f,\mathrm{surface}}$ [m min$^{-1}$] is given by:

$$\mathrm{ROS}_{f,\mathrm{surface}} = \frac{I_R \cdot \zeta \cdot (1 + \Phi_w)}{\rho_b \cdot \epsilon \cdot Q_{\mathrm{ig}}},$$ (65)

where $I_R$ is the reaction intensity, i.e. the energy release rate per unit area of fire front [kJ m$^{-2}$

min$^{-1}$]; $\zeta$ is the propagating flux ratio, i.e. the proportion of $I_R$ that heats adjacent fuel particles to ignition; $\Phi_w$ is a multiplier that accounts for the effect of wind in increasing the effective value of $\zeta$; $\rho_b$ is the fuel bulk density [kg m$^{-3}$], assigned by PFT and weighted over the 1-, 10- and 100-hour dead fuel classes; $\epsilon$ is the effective heating number, i.e. the proportion of a fuel particle that is heated to ignition temperature at the time flaming combustion starts; and $Q_{\mathrm{ig}}$ is the heat of pre-ignition,

i.e. the amount of heat required to ignite a given mass of fuel [kJ kg$^{-1}$]. With fuel bulk density $\rho_b$ defined as a PFT parameter, surface-area-to-volume ratios change with fuel load." Assuming that fires burn longer under high fire danger, we define fire duration ($t_{\mathrm{fire}}$) as

$$t_{\mathrm{fire}} = \frac{241}{1 + 240 \cdot \exp(-11.06 \cdot \mathrm{FDI})}$$ (66)

In the absence of topographic influence and changing wind directions during one fire event or

discontinuities of the fuel bed, fires burn an elliptical shape. Thus, the mean fire area [in ha] is defined as follows:

$$\overline{A_f} = \frac{\pi}{4 \cdot L_B} \cdot D_T^2 \cdot 10^{-4}$$ (67)





with $L_B$ is length to breadth ratio of elliptical fire, and $D_T$ is the length of major axis with:

$$D_T = \mathrm{ROS}_{f,\mathrm{surface}} \cdot t_{\mathrm{fire}} + \mathrm{ROS}_{b,\mathrm{surface}} \cdot t_{\mathrm{fire}} \qquad (68)$$

with $\mathrm{ROS}_{b,\mathrm{surface}}$, surface as the backward rate of spread. $L_B$ for grass and trees, respectively, is weighted depending on the foliage projective cover of grasses relative to woody PFTs in each grid cell.

SPITFIRE differentiates fire effects depending on burning conditions (intra- and interannual). If fires have developed insufficient surface fire intensity ($< 50\,\mathrm{kW\,m^{-1}}$), ignitions are extinguished (and

not counted in the model output). If the surface fire intensity has supported high enough scorch height of the flames, resulting scorching of the crown is simulated. Here, the tree architecture through the crown length, height of the tree determines fire effects and describes an important feedback between vegetation and fire in the model. PFT-specific parameters describe the trees sensitivity to or influence on scorch height and crown scorch. Surface fires consume dead fuel and live grass as a function of

their fuel moisture content. The amount of biomass burnt results from crown scorch and surface fuel consumption.

Post-fire mortality is modelled as a result of two fire mortality causes: crown and cambial damage. The latter occurs when insufficient bark thickness allows the heat of the fire to damage the cambium. It is defined as the ratio of the residence time of the fire to the critical time for cambial damage. The

probability of mortality due to crown damage (CK) is:

$$P_m(\mathrm{CK}) = r_{\mathrm{CK}} \cdot \mathrm{CK}^p, \qquad (69)$$

where $r_{\mathrm{CK}}$ is a resistance factor between 0 and 1, and $p$ is in the range of 3 to 4 (see SI-Table 7). The biomass of trees which die from either mortality cause is added to the respective dead fuel classes.

In summary, the PFT composition and productivity strongly influences fire risk through the mois-

ture of extinction, fire spread through composition of fuel classes (fine vs. coarse fuel), openness of the canopy and fuel moisture, fire effects through stem diameter, crown length and bark thickness of the average tree individual. The higher the proportion of grasses in a grid cell the faster fires can spread, the smaller the trees and/or the thinner their bark the higher the proportion of the crown scorched and the higher their mortality.

## 2.4 Crop functional types

In LPJmL4, twelve different annual crop functional types (CFTs) are simulated (SI-Table 8), similar to Bondeau et al. (2007) with the addition of sugarcane. The basic idea of CFTs is that these are parameterized as one specific representative crop (e.g. wheat, *Triticum aestivum* L.) to represent a broader group of similar crops (e.g. temperate cereals). In addition to the crops represented by the

twelve CFTs, other annual and perennial crops (*other crops*) are typically represented as managed grassland. Bioenergy crops are simulated to account for woody (willow trees in temperate regions,





eucalyptus for tropical regions) and herbaceous types (Miscanthus) (Beringer et al., 2011)). The physical cropping area (i.e. proportion per grid cell) of each CFT, the group of *other crops*, managed grasslands and bioenergy crops can be prescribed for each year and grid cell by using gridded land use data described in Fader et al. (2010) and Jägermeyr et al. (2015), see section 2.7. In principle, any land use dataset (including future scenarios) can be implemented in LPJmL4 at any resolution.

**Crop varieties and phenology**

Phenological development of crops in LPJmL4 is driven by temperature by way of accumulation of growing degree days, modified by vernalization requirements and sensitivity to daylength (photoperiod) for some CFTs and some varieties. Phenology is represented as a single phase from planting to physiological maturity. Different varieties of a single crop species are represented by different phenological heat unit requirements to reach maturity (phu), but also different harvest indices ($hi_{opt}$), i.e. the fraction of the above-ground biomass that is harvested, is typically CFT-specific, but can be specified to represent specific varieties (Asseng et al., 2013; Bassu et al., 2014; Kollas et al., 2015; Fader et al., 2010).

Heat units ($hu_t$, growing degree days) are accumulated ($hu_{sum}$) daily (see eq. (70)). The daily heat unit increment ($hu_t$) is the difference between the daily mean temperature of day $t$ and the CFT-specific base temperature (see SI-Table 8). The increment $hu_t$ cannot be less than zero at any given day. The phenological stage of the crop development (fphu) is expressed as the ratio of accumulated ($hu_{sum}$) and required phenological heat units (phu) (see eq. (71)). Physiological maturity is reached as soon as the sufficient growing degree days have been accumulated (fphu= 1.0). Both unfulfilled vernalization requirements as well as unsuitable photoperiod affect the phenological development of the CFTs (see eq. (75) and eq. (76)). Therefore, the daily increment $hu_t$ at day $t$ is scaled by reduction factors $v_{rf}$ for vernalization and $p_{rf}$ for photoperiod:

$$hu_{sum} = \sum_{t'=sdate}^{t} hu_{t'} \cdot v_{rf} \cdot p_{rf} \tag{70}$$

and

$$fphu = hu_{sum}/phu. \tag{71}$$

Wheat and rapeseed are implemented as spring and winter varieties. The model endogenously determines which variety to grow based on the average climate of past decades. If internally computed sowing dates for winter varieties (see below Section 2.7.1) indicate that the winter is too long to allow for growing winter varities, which is prior to day 258 (90) for wheat and 241 (61) for rapeseed on the northern (southern hemisphere), spring varieties are grown instead. These are computed on the basis of the sowing dates (sdate) as an indication for the length of the cropping season, constrained



by crop-specific limits. For winter varieties of wheat and rapeseed, phu is computed as:

$$\text{phu} = -0.1081 \cdot (\text{sdate} - \text{keyday})^2 + 3.1633$$

$$\cdot (\text{sdate} - \text{keyday}) + \text{phu}_{w_{\text{high}}}, \quad \text{phu} \leq \text{phu}_{w_{\text{low}}}, \tag{72}$$

where $\text{phu}_{w_{\text{low}}}$ and $\text{phu}_{w_{\text{high}}}$ are minimum and maximum phu requirements for winter varieties, respectively. The sowing date sdate can either be internally computed (see section 2.7.1) or prescribed for a crop and pixel. The parameter keyday is day 365 on the northern and day 181 on the southern hemisphere. For spring varieties of wheat and rapeseed, as well as for all other crops, $phu$ is computed as:

$$\text{phu} = \max(T_{\text{base}_{\text{low}}}, \text{atemp}_{20}) \cdot \text{pf}_{\text{CFT}},$$

$$\text{phu}_{s_{\text{high}}} \geq \text{phu} \geq \text{phu}_{s_{\text{low}}}, \tag{73}$$

where $\text{phu}_{s_{\text{low}}}$ and $\text{phu}_{s_{\text{high}}}$ are minimum and maximum phu requirements for spring varieties, respectively, $T_{\text{base}_{\text{low}}}$ is the minimum base temperature for the accumulation of heat units, $\text{atemp}_{20}$ is the 20-year moving average annual temperature and $\text{pf}_{\text{CFT}}$ is a CFT-specific scaling factor.

Vernalization requirements pvd are zero for spring varieties and are computed for winter varieties:

$$\text{pvd} = \text{vern}_{\text{date20}} - \text{sdate} - \text{ppvd}_{\text{CFT}}, \quad 0 \leq \text{pvd} \leq 60 \tag{74}$$

with $\text{ppvd}_{\text{CFT}}$ as a CFT-specific vernalization factor, sdate as the Julian day of the year of sowing and $\text{vern}_{date20}$ as the multi-annual average of the first day of the year when temperatures rise above a CFT-specific vernalization threshold ($T_{\text{vern}}$, see SI-Table 8). The effectiveness of vernalization is dependent on the daily mean temperature, being ineffective below -4°C and above 17°C, being fully effective between 3°C and 10°C and the effectiveness scales linear between -4 and 3 and between 10 and 17°C. The effective number of vernalizing days $\text{vd}_{\text{sum}}$ is accumulated until the requirements (pvd) as computed in eq. (74) are met or until phenology has progressed over 20% of its phenological development (i.e. fphu $\geq$ 0.2). Crop varieties can be parameterized as sensitive to photoperiod (i.e. daylength), but typically are assumed to be insensitive. Parameter settings can be adjusted for specific applications, such as in model intercomparisons (Asseng et al., 2013; Bassu et al., 2014; Kollas et al., 2015). Photoperiod restrictions are active until the crop reaches senescence.

The reduction factors are computed as:

$$v_{\text{rf}} = (\text{vd}_{\text{sum}} - 10.0)/(\text{pvd} - 10.0), \tag{75}$$

forcing $v_{\text{rf}}$ to be between 0 and 1, and

$$p_{\text{rf}} = (1 - p_{\text{sens}}) \cdot \min\big(1,$$

$$\max(0, (\text{daylength} - p_b)/(p_s - p_b))\big) + p_{\text{sens}}, \tag{76}$$



where $p_{\text{sens}}$ is the parametrized sensitivity to photoperiod $(0\ldots1)$, daylength is the duration of daylight (sunrise to sunset) in hours (see section 2.1.1), $p_b$ is the base photoperiod in hours and $p_s$ is the saturation photoperiod in hours.

**Crop growth and allocation**

Photosynthesis and autotrophic respiration of crops are computed as for the herbaceous natural PFTs (see section 2.2.1 and 2.2.3). Light absorption for photosynthesis is computed based on the Lamber-Beer law (Monsi, 1953), except for maize. For maize, LPJmL4 employs a linear LAI-FPAR model (Zhou et al., 2002) and a maximum leaf area index ($\text{LAI}_{\text{max}}$) of 5 instead of 7 as for all other CFTs (Fader et al., 2010). Daily NPP accumulates to total biomass and is allocated daily to crop organs in a

hierarchical order: roots, leaves, storage organ, mobile reserves/stem (pool). The fraction of biomass that is allocated to each compartment depends on the phenological development stage (fphu). The fraction of total biomass that is allocated to the roots ($f_{\text{root}}$) ranges between 40% at planting and 10% at maturity, modified by water stress:

$$f_{\text{root}} = \frac{0.4 - (0.3 \cdot \text{fphu}) \cdot \text{wdf}}{\text{wdf} + \exp(6.13 - 0.0883 \cdot \text{wdf})}, \tag{77}$$

where wdf is the ratio between accumulated daily transpiration and accumulated daily water demand since planting, representing a measure of the average water stress. After allocation to the roots, biomass is allocated to the leaves. Leaf area development follows a CFT-specific shape that is controlled by phenological development (fphu), the onset of senescence (ssn) and the shape of green LAI decline after onset of senescence. The ideal CFT-specific development of the canopy (equation

78) is thus described as a function of the maximum LAI ($\text{lai}_{\text{max}}$) and the phenological development (fphu) with two turning points in the phenological development ($\text{fphu}_c$ and $\text{fphu}_k$) and the corresponding fraction of the maximum green LAI reached at these stages ($\text{flai}_{\text{max}_c}$ and $\text{flai}_{\text{max}_k}$):

$$\text{flai}_{\text{max}} = \frac{\text{fphu}}{\text{fphu} + c \cdot (c/k)^{\frac{\text{fphu}_c - \text{fphu}}{\text{fphu}_k - \text{fphu}_c}}} \tag{78}$$

with

$$c = \frac{\text{fphu}_c}{\text{flai}_{\text{max}_c} - \text{fphu}_c} \tag{79}$$

$$k = \frac{\text{fphu}_k}{\text{flai}_{\text{max}_k} - \text{fphu}_k} \tag{80}$$

The onset of senescence is defined as a point in the phenological development $\text{fphu}_{\text{sen}}$. After the onset of senescence, i.e. $\text{fphu} \leq fphu_{\text{sen}}$, no more biomass is allocated to the leaves and the maximum green LAI is computed as:

$$\text{flai}_{\text{max}} = \left(\frac{1 - \text{fphu}}{1 - \text{fphu}_{\text{sen}}}\right)^{\text{ssn}} \cdot (1 - \text{flai}_{\text{max}_h}) + \text{flai}_{\text{max}_h} \tag{81}$$

with $\text{flai}_{\text{max}_h}$ as the green LAI fraction at which harvest occurs. This optimal development of LAI is modified by acute water stress. For this, the daily increment $\text{lai}_{inc}$, which is optimal for day $t$ is





computed as:

$$\text{lai}_{\text{inc}} = (\text{flai}_{\text{max}_t} - \text{flai}_{\text{max}_{t-1}}) \cdot \text{lai}_{\text{max}} \tag{82}$$

with $\text{flai}_{\text{max}_t}$ as the maximum green LAI of day $t$ and $\text{flai}_{\text{max}_{t-1}}$ as the maximum green LAI of the previous day. The daily increment $\text{lai}_{\text{inc}}$ is additionally scaled with the daily water stress ($\omega$), which is calculated as the ratio of actual transpiration and demand (see section 2.6.2) on that day. The calculation of $\text{lai}_{\text{inc}}$ applies to daily LAI increments which are independent of each other. The LAI on day $t$ is accumulated from daily LAI increments,

$$\text{LAI}_t = \sum_{t'=\text{sdate}}^{t} \text{lai}_{\text{inc}_{t'}} \cdot \omega \tag{83}$$

and implies that the LAI development cannot recover from water-limitation induced reductions in LAI. Until the onset of senescence, the daily LAI determines the biomass allocated to the leaves by dividing LAI by specific leaf area (SLA). SLA is computed as in eq. (53) using the $\beta_0$ value for grasses (2.25) and CFT-specific $\alpha_{\text{leaf}}$ values (SI-Table 9). Its calculation was adjusted for SLA

values given in Xu et al. (2010). Biomass in the storage organ is computed by phenological stage and the harvest index (HI), which describes the fraction of the above ground biomass that is allocated to the storage organ:

$$\text{HI} = \begin{cases} \text{fhi}_{\text{opt}} \cdot \text{hi}_{\text{opt}}, & \text{if } \text{hi}_{\text{opt}} \geq 1 \\ \text{fhi}_{\text{opt}} \cdot (\text{hi}_{\text{opt}} - 1) + 1, & \text{otherwise} \end{cases} \tag{84}$$

with

$$\text{fhi}_{\text{opt}} = 100 \cdot \text{fphu}/(100 \cdot \text{fphu} + \exp(11.1 - 10.0 \cdot \text{fphu})) \tag{85}$$

As the harvest index HI is defined relative to above-ground biomass, roots and tubers have HI values larger than 1.0 which needs to be accounted for in the allocation of biomass to the storage organ (see eq. (84)). If biomass is limiting (low NPP), biomass is allocated in hierarchical order, starting with roots (which can always be satisfied, as it is 40% of total biomass maximum), followed

by leaves ($C_{\text{leaf}}$) (where eventually the LAI is temporarily reduced, impacting APAR and thus NPP) and the storage organ ($C_{\text{so}}$). If biomass is not limiting, the allocation to the storage organ, this is computed from the harvest index (HI) and total above ground biomass:

$$C_{\text{so}} = \text{HI} \cdot (C_{\text{leaf}} + C_{\text{so}} + C_{\text{pool}}) \tag{86}$$

Excess biomass after allocating to roots, leaves and storage organ is allocated to a pool ($C_{\text{pool}}$)

that represents mobile reserves and the stem. At harvest, storage organs are collected from the field and crop residues can be left on the field or removed (for scenario setting, see Bondeau et al. (2007, e.g.)). If removed, a fraction of 10% of the above-ground biomass (leaves and pool) is assumed to remain on the field as stubbles. Stubbles and root biomass enter the litter pools after harvest.





### 2.5 Soil and litter carbon pools

Important for the global carbon balance are the biogeochemical processes in soil and litter. The LPJmL4 litter pool consists of CFT- resp. PFT-dependent pools for leaf, root and wood. The soil consists of a fast and a slow organic matter pool. Decomposition fluxes transferring litter carbon into soil carbon and losses for heterotrophic respiration ($R_h$) are described in the following section.

### 2.5.1 Decomposition

Decomposition of organic matter pools is represented by first-order kinetics (Sitch et al., 2003)

$$\frac{dC_{(l)}}{dt} = -k_{(l)} \cdot C_{(l)}, \tag{87}$$

where $C_{(l)}$ is the carbon pool size of soil or litter and $k_{(l)}$ is the annual decomposition rate per layer ($l$) in day$^{-1}$. Integrating for a time interval $\Delta t$ (here 1 day) yields:

$$C_{(t+1,l)} = C_{(t,l)} \cdot \exp(-k_{(l)} \cdot \Delta t), \tag{88}$$

where $C_{(t,l)}$ and $C_{(t+1,l)}$ are the carbon pool sizes at the beginning and the end of the day. The amount of carbon decomposed per layer is:

$$C_{(t,l)} \cdot (1 - \exp(-k_{(l)} \cdot \Delta t)) \tag{89}$$

at which 70% of decomposed litter goes directly into the atmosphere $R_{h,\text{litter}}$, the remaining is transferred the the soil carbon pools, 98.5% to the fast soil carbon pool and 1.5% to the slow carbon

pool.

$$R_h = R_{h,\text{litter}} + R_{h,\text{fastSoil}} + R_{h,\text{slowSoil}} \tag{90}$$

The decomposition rates for root litter and soil ($k_{(l,p)}$) is a function of soil temperature and soil moisture:

$$k_{(l,p)} = \frac{1}{\tau_{10_{(p)}}} \cdot g(T_{\text{soil}_{(l)}}) \cdot f(\theta_{(l)}), \tag{91}$$

which is reciprocal to the mean residence time ($\tau_{10_{(p)}}$). Root litter decomposition is defined for all PFTs (0.3 yr$^{-1}$) and for fast and slow soil carbon (0.03 and 0.001 yr$^{-1}$ resp.) as in Sitch et al. (2003), $p$ represents the different pools. The decomposition rate of leaf and wood litter is defined as PFT-specific decomposition rates at 10°C for leaf, wood and root, which has been analysed and proposed by Brovkin et al. (2012) for leaf and wood. The temperature dependence function for the

fast and slow soil carbon and the leaf and root litter pool $g(T_{\text{soil}})$ was already described in eq. (44). For wood litter decomposition it is calculated as follows:

$$k_{\text{wood,p}} = Q_{10}, \frac{(T_{\text{soil}} - 10)}{10.0} \tag{92}$$



SI-Table 5 presents $(1/\tau_{10_{(p)}})$ used for leaves and wood and the $Q_{10}$ parameter for temperature-dependent wood decomposition in the litter pool. The soil moisture function follows Schaphoff et al. (2013):

$$f(\theta_{(l)}) = 0.04021601 - 5.00505434 \cdot \theta_{(l)}^3 \\ + 4.26937932 \cdot \theta_{(l)}^2 + 0.71890122 \cdot \theta_{(l)} \tag{93}$$

$\theta_{(l)}$ is the soil volume fraction of the layer $l$. Parameters are chosen based on the assumption that rates are maximal at field capacity and decline for higher $\theta_{(l)}$ to 0.2. $f(\theta_{(l)})$ is very small (0.04021601) when $\theta_{(l)}$ equals 1 due to oxygen limitation and when $\theta_{(l)}$ is 0.

To account for different decomposition rates in the different soil layers, a vertical soil carbon distribution is now implemented in LPJmL4 following Schaphoff et al. (2013). Jobbagy and Jackson (2000) suggested a cumulative log-log distribution of the fraction of soil organic carbon (Cf$_l$) as a function of depth with:

$$\text{Cf}_{(l)} = 10^{k_{\text{soc}} \cdot \log_{10}(d_{(l)})}, \tag{94}$$

where $d_{(l)}$ is the relative share of the layer $l$ in the entire soil bucket and the parameters $k_{\text{soc}}$ was adjusted for the soil layer depth now used in LPJmL4 (see SI-Table 5). The total amount of soil carbon $C_{s_{\text{total}}}$ is estimated from the mean annual decomposition rate $k_{\text{mean}_{(l)}}$ and the mean litter input into the soil as in (Sitch et al., 2003), but is distributed to all root layers separately (eq. (95)). The envisaged vertical soil distribution $C_{(l)}$:

$$C_{(l)} = \sum_{p=1}^{n_{PFT}} d_{(l)}^{k_{\text{soc}(p)}} \cdot C_{s_{\text{total}}} \tag{95}$$

is estimated after a carbon equilibrium phase of 2310 years. The mean decomposition rate for each PFT $k_{\text{mean}_{\text{PFT}(p)}}$ can be derived from the mean annual decomposition rate $k_{\text{mean}_{(l)}}$ of the spinup years as a layer-weighted value derived from eq. (94):

$$k_{\text{mean}_{\text{PFT}(p)}} = \sum_{l=1}^{n_{\text{soil}}} k_{\text{mean}_{(l)}} \cdot \text{Cf}_{(l,p)} \tag{96}$$

The annual carbon shift rates $C_{\text{shift}_{(l,p)}}$ describe the organic matter input from the different PFTs into the respective layer due to cryoturbation and bioturbation and are designed for global applications:

$$C_{\text{shift}_{(l,p)}} = \frac{\text{Cf}_{(l,p)} \cdot k_{\text{mean}_{(l)}}}{k_{\text{mean}_{\text{PFT}(p)}}}. \tag{97}$$

## 2.6 Water balance

The terrestrial water balance is a pivotal element in LPJmL4 as water and vegetation are linked in multiple ways:



1. the coupling of plant transpiration and carbon uptake from the atmosphere in the process of photosynthesis;

2. the down-regulation of photosynthesis, plant growth and productivity in response to soil water limitation (relative to atmospheric moisture demand), in case actual canopy conductance is below potential canopy conductance (in the demand function that describes transpiration);

3. the effect of changes in vegetation type, distribution, phenology and production on evaporation, transpiration, interception, runoff and soil moisture;

4. the anthropogenic stimulation of crop growth through irrigation with water taken from rivers, dams, lakes and assumed renewable groundwater.

These couplings of water and vegetation dynamics enable simulation of the interacting mutual feedbacks between freshwater cycling in and above the earth surface and terrestrial vegetation dynamics.

### 2.6.1 Soil water balance

Advancing the former two-layer approach (Sitch et al., 2003), LPJmL4 divides the soil column into five hydrological active layers of $0.2$, $0.3$, $0.5$, $1$ and $1$ m thickness (Schaphoff et al., 2013). Water holding capacity (water content at permanent wilting point, at field capacity and at saturation) and hydraulic conductivity are derived for each grid cell using soil texture from the Harmonized World Soil Database (HWSD) version 1 (Nachtergaele et al., 2008) and relationships between texture and

hydraulic properties from Cosby et al. (1984), see also section 2.1.3.

Water content in soil layers is altered by infiltrating rainfall and vertical movement of gravitational water (percolation). Since the accuracy needed for a global model does not justify the computational costs of an exact solution of the governing differential equation, a simplified storage approach is implemented in LPJmL4. Rather than calculating infiltration and percolation of precipitation at once,

total precipitation is divided in portions of $4$ mm that are routed through the soil one after another. This effectively emulates a time discretization, which leads to a higher proportion of runoff being generated for higher amounts precipitation.

**Infiltration**

The infiltration rate of rain and irrigation water into the soil (infil, in mm) depends on current soil

water content of the first layer as follows:

$$\mathrm{infil} = P \cdot \sqrt{1 - \frac{\mathrm{SW}_{(1)} - W_{\mathrm{pwp}_{(1)}}}{W_{\mathrm{sat}_{(1)}} - W_{\mathrm{pwp}_{(1)}}}}, \qquad (98)$$

where $W_{\mathrm{sat}_{(1)}}$ is the soil water content at saturation and $W_{\mathrm{pwp}_{(1)}}$ the soil water content at wilting point, and $\mathrm{SW}_{(1)}$ the total actual soil water content of the first layer in mm. $P$ is the amount of water in the current portion of daily precipitation or applied irrigation water (maximum 4 mm). The

surplus water that does not infiltrate is assumed to generate surface runoff.





**Percolation**

Subsequent percolation through the soil layers is calculated by the storage routine technique (Arnold et al., 1990) as used in regional hydrological models such as SWIM (Krysanova et al., 1998).

$$\mathrm{FW}_{(t+1,\,l)} = \mathrm{FW}_{(t,\,l)} \cdot \exp\left(-\frac{\Delta t}{\mathrm{TT}_{(l)}}\right),$$ (99)

where $\mathrm{FW}_{(t,l)}$ and $\mathrm{FW}_{(t+1,l)}$ are the soil water content between field capacity and saturation at the beginning and the end of the day for all soil layers $l$, respectively. $\Delta t$ is the time interval (here, 24 hours) and $\mathrm{TT}_l$ determines the travel time through the soil layer in hours:

$$\mathrm{TT}_{(l)} = \frac{\mathrm{FW}_{(l)}}{\mathrm{HC}_{(l)}}$$ (100)

$\mathrm{HC}_{(l)}$ is the hydraulic conductivity of the layer in $\mathrm{mm\,hour^{-1}}$:

$$\mathrm{HC}_{(l)} = K_{s(l)} \cdot \left(\frac{\mathrm{SW}_{(l)}}{W_{\mathrm{sat}_{(l)}}}\right)^{\beta_{(l)}},$$ (101)

where $W_{\mathrm{sat}_{(l)}}$ is the soil water content at saturation, $K_{s_{(l)}}$ is the saturated conductivity in $\mathrm{mm\,hour^{-1}}$ and $\mathrm{SW}_{(l)}$ the total soil water content of the layer in mm. Thus, percolation can be calculated by subtracting $\mathrm{FW}_{(t,l)}$ from $\mathrm{FW}_{(t+1,l)}$ for all soil layers:

$$\mathrm{perc}'_{(l)} = \mathrm{FW}_{(t,l)} \cdot \left[1 - \exp\left(\frac{-\Delta t}{\mathrm{TT}_{(l)}}\right)\right]$$ (102)

The percolation rate $\mathrm{perc}_{(l)}$ is limited by soil moisture of the lower layer, similar to the infiltration approach.

$$\mathrm{perc}_{(l)} = \mathrm{perc}'_{(l)} \cdot \sqrt{1 - \frac{\mathrm{SW}_{(l)} - W_{\mathrm{pwp}_{(l)}}}{W_{\mathrm{sat}_{(l)}} - W_{\mathrm{pwp}_{(l)}}}}$$ (103)

Excess water over the saturation levels forms lateral runoff in each layer and contributes to sub-surface runoff. The formation of groundwater, which is the seepage from the bottom soil layer, has

been recently introduced into LPJmL4 (Schaphoff et al., 2013). Both surface and subsurface runoff are simulated to accumulate to river discharge (see section 2.6.3).

**2.6.2 Evapotranspiration**

Similar to Gerten et al. (2004), evapotranspiration (ET) is the sum of vapor flow from the earth surface to the atmosphere. It consists of three major components: evaporation from bare soils, evap-

oration of intercepted rainfall from the canopy, and plant transpiration through leaf stomata. The calculation of these different components in LPJmL4 is based on equilibrium evapotranspiration ($E_{\mathrm{eq}}$) resp. the PET as described in section 2.1.1 and eq. (2) and (6).





**Canopy evaporation**

Canopy evaporation is the evaporation of rainfall that has been intercepted by the canopy, limited either by PET or the amount of intercepted rainfall $I$ (both in mm day$^{-1}$):

$$E_{\text{canopy}} = \min(\text{PET}, I) \tag{104}$$

The amount of intercepted rainfall is given as:

$$I = \sum_{\text{pft}=1}^{n_{\text{PFT}}} I_{\text{pft}} \cdot \text{LAI}_{\text{pft}} \cdot \text{Pr}, \tag{105}$$

where $I_{pft}$ is the interception storage parameter for each PFT (Gerten et al., 2004), $\text{LAI}_{pft}$ the PFT-specific leaf area per unit of grid cell area and Pr is daily precipitation in mm day$^{-1}$.

**Soil evaporation**

Soil evaporation ($E_s$) only occurs from bare soil, where the vegetation cover ($f_v$) is less than 100%. The $f_v$ is the sum of all present PFT's FPC (see eq. (55)) taking daily phenology into account. The evaporation flux depends on available energy for the vaporization of water (see eq. (6)) and the available water in the soil. LPJmL4 assumes that water for evaporation is available from the upper 0.3 m of the soil, implicitly accounting for some capillary rise. Evaporation-available soil water ($w_{\text{evap}}$) is thus all water above wilting point of the upper layer (0.2 m) and one third of the second layer (0.3 m). Actual evaporation is then computed according to eq. (107), with $w$ being the evaporation-available water relative to the water holding capacity in that layer whc$_{\text{evap}}$

$$w = \min(1, w_{\text{evap}}/\text{whc}_{\text{evap}}) \tag{106}$$

thus:

$$E_s = \text{PET} \cdot w^2 \cdot (1 - f_v) \tag{107}$$

This potential evaporation flux is reduced if a portion of the water is frozen or if the energy for the vaporization has already been used to vaporize water that was intercepted by the canopy or for plant transpiration (see section 2.6.2).

**Plant transpiration coupled with photosynthesis**

Plant transpiration ($E_T$) is modelled as the lesser of plant-available soil water supply function ($S$) and atmospheric demand function ($D$), following Federer (1982):

$$E_T = \min(S, D) \cdot f_v \tag{108}$$

$S$ is defined by a PFT-specific maximum water transport capacity ($E_{\max}$) and the relative water content ($w_r$) and phenology (phen$_{\text{PFT}}$), because plants need no water when they have shed all their




leaves:

$$S = E_{\max} \cdot w_r \cdot \text{phen}_{\text{PFT}} \tag{109}$$

The water accessible for plants ($w_r$) is computed from the relative water content at field capacity ($w_l$) and the fraction of roots ($\text{rootdist}_l$) within each soil layer ($l$) as

$$w_r = \sum_{l=1}^{n_{\text{soil}}-1} w_l \cdot \text{rootdist}_l \tag{110}$$

$\text{rootdist}_l$ can be calculated from the proportion of roots from surface to soil depth $z$, $\text{rootdist}_z$, as in Jackson RB et al. (1996):

$$\text{rootdist}_z = \frac{\int_0^z (\beta_{\text{root}})^{z'} dz'}{\int_0^{z_{\text{bottom}}} (\beta_{\text{root}})^{z'} dz'} = \frac{1 - (\beta_{\text{root}})^z}{1 - (\beta_{\text{root}})^{z_{\text{bottom}}}}, \tag{111}$$

where $\beta_{\text{root}}$ represents a numerical index for root distribution (for parameter values see SI-Table 6). $\text{rootdist}_l$ is given by the difference $\text{rootdist}_{z(l)} - \text{rootdist}_{z(l-1)}$. If the soil depth of layer $l$ is greater than the thawing depth then $\text{rootdist}_l$ is set to zero. The non-zero $\text{rootdist}_l$ are rescaled in such a way that their sum is normalized to 1 considering the reallocation of the root distribution under freezing conditions.

Plants in natural vegetation compete for water resources and thus only have access to the fraction of water that corresponds to their foliage projected cover ($\text{fpc}_{\text{PFT}}$)

$$S_{\text{PFT}} = S \cdot \text{fpc}_{\text{PFT}} \tag{112}$$

For agricultural crops, water supply is also dependent on their root biomass $\text{bm}_{\text{root}}$

$$S = E_{\max} \cdot w_r \cdot (1 - \exp(-0.0411 \cdot \text{bm}_{\text{root}})) \tag{113}$$

Atmospheric demand ($D$) is a hyperbolic function of $g_c$ (see section 2.2.1 and eq. (38)), following Monteith (1995), and employs a maximum Priestley-Taylor coefficient $\alpha_m = 1.391$, describing the asymptotic transpiration rate, and a conductance scaling factor $g_m = 3.26$:

$$D = (1 - \text{wet}) \cdot E_{\text{eq}} \cdot \alpha_m / (1 + g_m / g_c), \tag{114}$$

where wet is the fraction of $E_{\text{eq}}$ that was used to vaporize intercepted water from the canopy (see section 2.6.2) and $g_c$ is the potential canopy conductance. If $S$ is not sufficient to fulfill transpiration demand $g_c$ is recalculated for $D = S$ and photosynthesis rate might be adjusted see section 2.2.1.

### 2.6.3 River routing

#### Description of the river routing module

The river routing module computes the lateral exchange of discharge (see section 3.1.2 for input) between grid cells through the river network (Rost et al., 2008). The transport of water in the river





channel is approximated by a cascade of linear reservoirs. River sections are divided into $n$ homogeneous segments of length $L$, each behaving like a linear reservoir. Following the unit hydrograph method (Nash, 1957), the outflow $Q_{\text{out}}(t)$ of a linear reservoir cascade for an instantaneous inflow $Q_{\text{in}}$ is given as:

$$Q_{\text{out}}(t) = Q_{\text{in}} \cdot \frac{1}{K \cdot \Gamma(n)} \left( \frac{t}{K} \right)^{n-1} \cdot \exp(-t/K), \tag{115}$$

where $\Gamma(n)$ is the gamma function that replaces $(n-1)!$ to allow for non-integer values of $n$. $K$ is the storage parameter, defined as the hydraulic retention time of a single linear reservoir segment of length $L$. It can be calculated as the average travel time of water through a single river segment:

$$K = \frac{L}{v}, \tag{116}$$

where $v$ is the average flow velocity.

The river routing in LPJmL4 is calculated at a time step of $\Delta t = 3$ hour. We assume a globally constant flow velocity $v$ of $1\,\text{m}\,\text{s}^{-1}$ and a segment length $L$ of $10\,\text{km}$ to calculate the parameters $n$ and $K$ for each route between grid-cell mid points. At the start of the simulation, for each route the unit hydrograph for a rectangular input impulse of length $\Delta t$ is determined from eq. (115). Because eq. (115) assumes an instantaneous input impulse, we numerically determine the response to a rectangular input impulse by adding up the responses of a series of 100 consecutive instantaneous input impulses. From the obtained unit hydrograph, the sum of outflow during each subsequent time step $\Delta t$ is recorded until 99% of the total input impulse has been released (maximum 24 time steps). During simulation thus determined response function is then used to calculate the convolution integral for the flow packages routed through the network. An efficient parallelisation of the river-routing scheme using global communicators of the MPI message passing library is described in Von Bloh et al. (2010).

### 2.6.4 Irrigation and dams

LPJmL4 explicitly accounts for human influences on the hydrological cycle by accounting for irrigation water abstraction, consumption and return flows, and non-agricultural water consumption from households, industry and livestock (HIL), as well as an implementation of reservoirs and dams.

**Irrigation**

LPJmL4 features a mechanistic representation of the world's most important irrigation systems (surface, sprinkler, drip), which is key to refined global simulations of agricultural water use as constrained by biophysical processes and water tradeoffs along the river network. HIL water use in each grid cell is based on Flörke et al. (2013) (accounting for 201 km$^3$ in the year 2000). We assume HIL water to be withdrawn prior to irrigation water. LPJmL4 comes with the first input dataset that details the global distribution of irrigation types for each cell and crop type (Jägermeyr et al., 2015).





Irrigation water partitioning is dynamically calculated in coupling to the modelled water balance,
and climate, soil, and vegetation properties. The spatial pattern of improved irrigation efficiencies
are presented in Jägermeyr et al. (2015).

Irrigation water demand is withdrawn from available surface water, i.e. river discharge (see section 2.6.3), lakes, and reservoirs (section 2.6.5), and if not sufficient in the respective grid cell, requested from neighboring upstream cells. The amount of daily irrigation water requirements is
915 based on the soil water deficit, resulting crop water demand (net irrigation requirements, NIR), and irrigation-system-specific application requirements (specified below). If soil moisture goes below the CFT-specific irrigation threshold ($it$), the total amount (daily gross irrigation requirements) is requested for abstraction. NIR is defined as the water needs of the top 50 cm soil layer to avoid water limitation to the crop. It is calculated to meet field capacity if the water supply (root-available
soil water) falls below the atmospheric demand (potential evapotranspiration, see section 2.6.2) as:

$$\text{NIR} = W_{\text{fc}} - w_a - w_{\text{ice}}, \quad \text{NIR} \geq 0, \tag{117}$$

where $w_a$ is actually available soil water and $w_{\text{ice}}$ the frozen soil content in mm. Due to inefficiencies in any irrigation system, excess water is required to meet the water demand of the crop. To this end, we calculate system-specific conveyance efficiencies ($E_c$) and application requirements (AR),
which lead to gross irrigation requirements (GIR, in mm):

$$\text{GIR} = \frac{\text{NIR} + \text{AR} - \text{Store}}{E_c}, \tag{118}$$

where Store stands in as a storage buffer (see also SI-Fig. 2 for conceptual description). For pressurized systems (sprinkler and drip), $E_c$ is set to 0.95. For surface irrigation we link $E_c$ to soil saturated hydraulic conductivity ($K_s$, see section 2.6.1), adopting $E_c$ estimates from Brouwer et al.
(1989). Half of conveyance losses are assumed to occur due to evaporation from open water bodies and the remainder is added to the return flow as drainage.

Indicative of application losses, AR represents the excess water needed to uniformly distribute irrigation across the field. AR is calculated as a system-specific scalar of the free water capacity:

$$\text{AR} = (W_{\text{sat}} - W_{\text{fc}}) \cdot d_u - \text{FW}, \quad \text{AR} \geq 0, \tag{119}$$

where $d_u$ is the water distribution uniformity scalar, as a function of the irrigation system and FW represents the available free water (see sections 2.6 and 2.6.2) (see Jägermeyr et al. (2015) for details).

Irrigation scheduling is controlled by Pr and the irrigation threshold (IT) that defines tolerable soil water depletion prior to irrigation (see SI-Table 12). Accessible irrigation water is subtracted
by the precipitation amount. Irrigation water volumes that are not released (if $S > \text{IT}$) are added to Store and are available for the next irrigation event. Withdrawn irrigation volumes are subsequently reduced by conveyance losses.



Irrigation water application is assumed to occur below the canopy for surface and drip systems, and sprinkler systems above-canopy, which leads to interception losses (calculated as described above). Drip system are assumed to apply irrigation water localized to the plant root zone below the surface and thereby reduce soil evaporation by 60% (section 2.6.2). Note that drip systems are parameterized to represent a modest form of deficit irrigation, i.e. to save water and not to maximize yields. For detailed parameterization of the three irrigation systems implemented, see SI-Table 12 and Jägermeyr et al. (2015).

### 2.6.5 Dams, lakes and reservoirs

The operation of large reservoirs affects the seasonal discharge patterns downstream of the dam, as well as the amount of water that is locally available for irrigation.

In LPJmL4, reservoirs are considered starting from the prescribed year they were built (Biemans et al., 2011). The reservoir is filled daily with discharge from upstream locations and with local precipitation. At the beginning of an operational year, which is defined as the first month when mean monthly inflow is lower than mean annual inflow, the actual storage in the reservoir is compared with the maximum storage capacity of the reservoir. The reservoir outflow factor of the following year is adjusted accordingly to compensate for interannual flow fluctuations.

Subsequently, a target release is defined based on the main purpose of the reservoir. Dams built primarily for irrigation are assumed to release their water proportionally to gross irrigation water demand downstream. Dams built primarily for other purposes (hydropower, flood control, etc.) are assumed to be designed for releasing a constant water volume throughout the year. The actual release from a reservoir is simulated to depend on its storage capacity relative to its inflow. If an irrigation purpose is defined for the reservoir, part of the outflow is diverted to irrigated lands downstream. Cells receive water from the reservoirs when the following conditions are met: the cells have a lower altitude than the cell containing the reservoir, and they are situated along the main river downstream or at maximum five cells upstream. Thus, a cell can receive water from multiple reservoirs.

As irrigation demands vary daily, water released from reservoirs can be stored in the conveyance system for up to five days. If the total irrigation water demand to a reservoir cannot be fulfilled, all requesting fields are supplied with the same fraction of their demand (see Biemans et al. (2011) for details).

### 2.7 Land use

Human land use is represented in LPJmL4 by dividing grid cells, which have the same climate and soil-texture input, into separate sub-units, referred to as stands. Stands are driven by the same input data, but changes in soil water and soil carbon are computed separately. When new stands are created, their soils are direct copies of the stand from which they are generated. If stands are merged, soil properties are averaged according to the two stands' size to maintain mass and energy balance.





Natural vegetation (i.e. PFTs), agricultural crops (i.e. CFTs), managed grasslands and bioenergy plantations are represented on separate stands that can partly or fully cover any grid cell. The size of each stand is determined by the extent of land-use, defined by the input data prescribing fractions of each land-use type (crops, managed grassland, bioenergy plantations; all as rainfed and/or irrigated cultivation). All natural vegetation grows on a single natural vegetation stand on which all present PFTs compete for water and light. Agricultural crops are implemented as monocultures where only one single crop is cultivated and where there is no competition for resources with other stands (fields or natural vegetation) within that cell. For each crop and irrigation system (irrigated or rainfed) there can always only be one stand within one grid cell. For irrigated crops only one irrigation system (sprinkler, surface or drip, see section 2.6.4) can be selected.

At the beginning of each simulation year, present total agricultural land (all crops, all managed grasslands) is compared with land requirements for the year according to the input data. If there is too little agricultural land available, the needed fraction is cleared (liberated) from the natural vegetation stand, if there is too much, the excess agricultural land is abandoned and merged into the natural vegetation stand, leaving new space there for establishment of natural vegetation. Unculti­vated cropland (i.e. outside the cropping period) is merged in set-aside stands, separated in irrigated and rainfed to prevent that irrigation water from irrigated stands is transferred to rainfed stands in off-seasons. Set-aside land from irrigated agriculture is not irrigated during fallow periods, but is kept separate because of the soil water content that is enhanced through irrigation during the grow­ing period. Land can be transferred between the two set-aside stands if the ratio between irrigated and rainfed cropland changes.

Depending on the scenario setting, if inter-cropping is assumed, a simple intercrop (grass) can be grown on the set-aside stand during the fallow period. Once a sowing date for a crop is reached (see section 2.7.1), the prescribed fraction of that crop and irrigation system is removed from the set-aside stand by copying the soil properties of the set-aside stand to the newly created stand and reducing the set-aside stand's size accordingly. The crop is then cultivated on that newly created stand and returned to the set-aside upon harvest of the crop.

### 2.7.1 Sowing dates

Sowing dates are simulated based on a set of rules depending on climate and crop specific thresholds as described in Waha et al. (2012). The start of the growing period is assumed to dependend either on the onset of the wet season in tropical and subtropical regions or on the exceeding of a crop-specific temperature threshold for emergence in temperate regions.

In order to simulate a reasonable global distribution of temperate and tropical regions, we describe the intra-annual variability of precipitation and temperature in each location using variation coeffi­cients for temperature ($CV_{temp}$) and precipitation ($CV_{prec}$), calculated from past monthly climate





data. We assume temperature seasonality if $CV_{temp}$ exceeds $0.01$ and precipitation seasonality if $CV_{prec}$ exceeds $0.4$. Hence, four seasonality types can be differentiated (SI-Fig. 3):

1. temperature seasonality

2. precipitation seasonality, and

3. temperature and precipitation seasonality

4. no temperature and no precipitation seasonality

For locations with a combined temperature and precipitation seasonality, we additionally consider
the mean temperature of the coldest month. If it exceeds $10°C$, we assume absence of a cold season, i.e. the risk of frost occurrence is negligible, assuming temperatures are high enough to sow all year-round. Accordingly, precipitation seasonality defines the timing of sowing. If the mean temperature of the coldest month is equal to or below $10°C$, temperature seasonality determines the timing of sowing. In regions with precipitation seasonality only, sowing date is at the onset of the main wet
season. The precipitation-to-potential-evapotranspiration ratio is used to find moist and dry months in a year, as suggested by Thornthwaite (1948). The main wet season is identified by the largest sum of monthly precipitation-to-potential-evapotranspiration ratios of 4 consecutive months, because the length of that period aligns well with the length of the growing period of the majority of the simulated crops. In regions with bimodal rainfall patterns, the wet season starts with the first month of the
longest wet season. Crops are sown at the first wet day in the main wet season of the year i.e. when daily precipitation exceeds $0.1$ mm. The onset of the growing period depends on temperature, if temperature seasonality is detectable. Accordingly, crop emergence is related to temperature, and thus sowing starts when daily average temperature exceeds a certain threshold ($T_{fall}$ resp. $T_{spring}$ SI-Table 8). Locations without any temperature or precipitation seasonality e.g. in the wet tropics
crops are sown on the 1st of January.

### 2.7.2 Management and cropping intensity

Agricultural management is represented as a distinct set of options and a calibration of cropping intensity. Explicit management options include:

1. cultivar choices, see section 2.4

2. sowing dates, see section 2.7.1

3. irrigation shares and type section 2.6.4

4. residue removal section 2.4

5. intercrops section 2.7





Different irrigation systems can be represented as follows. For drip irrigation systems assuming lo-calised sub-surface water application, soil evaporation is reduced, so that only 40% of the applied irrigation water are available for evaporation. Also, for rainwater management (see section 2.6.2), soil evaporation can be reduced, mimicking agricultural management systems like mulching tech-niques or conservation tillage (Jägermeyr et al., 2016). Secondly, to simulate improved rainwater management (see section 2.6.1), the infiltration capacity can be increased, mimicking agricultural management practices such as different tillage systems or organic mulching (Jägermeyr et al., 2016).

Other than that, management options are not treated explicitly in the LPJmL4 model, that is, it assumes no nutrient limitation to crop growth. Current management patterns, which is desirable for, e.g., studies of the carbon or water cycle, can be represented by calibrating national cropping in-tensity to FAO statistics as described in Fader et al. (2010). For this the maximum leaf area index ($LAI_{max}$), the harvest index parameter ($hi_{opt}$), and a scaling factor for scaling leaf-level photosyn-thesis to stand level ($\alpha_a$, Haxeltine and Prentice (1996)) are scaled in combination. $LAI_{max}$ can range between 1 (lowest intensity) and 5 for maize or 7 for all other crops (highest intensity), and $\alpha_a$ ranges from $0.4$ to $1.0$. The Parameter $hi_{opt}$ is crop-specific (see SI-Table 8), which can be reduced by up to 20%, assuming there are more robust in return less productive varieties (Gosme et al., 2010).

### 2.7.3 Managed grassland

On managed grassland stands, only herbaceous PFTs (TrH, TeH, PoH, see Table 1 for definition) can establish conditional to their bio-climatic limits (see SI-Table 2). If more than one herbaceous PFT establishes, these compete for light and water resources, but do not interact with other stands in that grid cell. In contrast to annual C allocation described above (see section 2.3.1), LPJmL4 simulates managed grasslands and herbaceous biomass plantations (see section 2.7.4) with a daily allocation and turnover scheme where it dynamically computes leaf biomass per day as described below. It therefore enables to better represent the current phenological state and suitable times for harvest.

**Daily allocation of managed grasslands**

The allocation scheme is designed to distributes daily biomass increment to leaf and root biomass in a way that best fulfils the predetermined ratio of leaf to root mass, lr, for the whole plant. This allows for short-term deviations from allowed leaf-to-root-mass ratios lr after harvest events, when much of the leaf biomass is removed. After a harvest event, NPP is first allocated to leaves until lr is restored. If more $CO_2$ is assimilated than needed for maintenance respiration (i.e. NPP is positive), assimilated carbon $B_I$ is allocated to the root ($R$) and the leaf carbon pool ($L$) by calculating the respective increments ($L_I, R_I$) (eqs. (120) and (121)).

$$L_I = \min\left( B_I, \max\left( \frac{B_I + R - L/\text{lr}}{1 + 1/\text{lr}}, 0 \right) \right) , \qquad (120)$$
$$R_I = B_I - L_I . \qquad (121)$$





In case of negative NPP (i.e. maintenance and growth respiration are larger than the GPP of that day), both compartments (leaves and roots) are reduced proportionally. lr is scaled with a measure of average growing-season water stress (mean of daily ratios of plant water supply to atmospheric demand, see section 2.6.2) to account for the functional relationship that plants allocate more carbon to roots under dry conditions.

$$\mathrm{lr} = \mathrm{lr}_p \cdot W_{\mathrm{supply}}/W_{\mathrm{demand}} \tag{122}$$

**Grassland harvest routine**

LPJmL4 employs a default harvest scheme that attempts to approximate the actual global grassland production of 2.3 Gt DM (Herrero et al., 2013) while avoiding degradation. A harvest event of grass biomass occurs when leaf biomass increases over the previous month. Prior to the harvest event, grass leaf biomass ($C_{\mathrm{leaf}}$) and the biomass after the last harvest event ($\mathrm{MC}_{\mathrm{leaf}}$) is summed up for all grass species at the managed grassland stand:

$$C_{\mathrm{leaf}} = \sum_{\mathrm{pft}} \mathrm{Bml}_{\mathrm{pft}}, \qquad \mathrm{MC}_{\mathrm{leaf}} = \sum_{\mathrm{pft}} \mathrm{Mml}_{\mathrm{pft}}. \tag{123}$$

On the last day of each month harvest occurs and the harvest index $H_{\mathrm{frac}}$ is determined depending on the leaf biomass $C_{\mathrm{leaf}}$.

$$\text{If day} \quad (C_{\mathrm{leaf}} > \mathrm{MC}_{\mathrm{leaf}}) \qquad H_{\mathrm{frac}} = 1 - \frac{1000}{1000 + C_{\mathrm{leaf}}} \tag{124}$$

Harvested biomass is taken from the leaf biomass of each herbaceous PFT. Depending on the amount of carbon in the leaves the harvested fraction is increasing (SI-Fig. 4) and biomass harvested depends on the present leaf carbon. In the absence of any detailed information about actual grassland management systems, this generic harvest routine does not represent specific management systems but allows for simulating regular harvest events (be it grazing or mowing) during productive periods of the year and the harvest amount is automatically adjusting to productivity.

### 2.7.4 Biomass plantations

Three biomass functional types (BFTs) were implemented in LPJmL4 (one fast-growing C4 grass, a temperate and a tropical tree) to allow for the simulation of dedicated biomass plantations (Beringer et al., 2011). These BFTs are generic representations of some of the most promising types of crops for the production of 2nd generation biofuels, biomaterials, or energy (possibly in combination with carbon capture and storage mechanisms). Their parametrization is partly identical to their natural PFT equivalents tropical C4 perennial grass, temperate broadleaved summergreen tree and tropical broadleaved raingreen tree yet with some important modifications to characterize the enhanced growth characteristics of these managed vegetation types (see SI-Table 10).

Woody energy crops are represented as short rotation coppice systems (SRC). In short intervals young tree stems are cut down to near ground stumps , implemented as regularly cycles (see SI-





Table 11). A grown root system and nutrient storage in roots and stumps enables high yielding varieties of poplar, willow and Eucalyptus used for SRC to regrow forcefully in renewal years. Until plantations need to replanted after 40 years, several harvest cycles are possible. Under an effective pest and fire control on modern biomass plantations mortality and fire occurrence are reduced (to
1115 zero emissions) compared to natural vegetation.

## 3 Modelling protocol

The objective of this publication is to provide a comprehensive description of the LPJmL4 model. Here we also provide some outputs from a standard simulation of the historic period 1901 to 2011, which is also the basis for the actual, more comprehensive model evaluation described in the com-
1120 panion paper (Schaphoff et al., submitted).

### 3.1 Model setup and inputs

For this simulation, all carbon and water pools in the model are initialized to zero and a spinup simulation for 5000 years is conducted in which plants dynamically establish, grow and die following the model dynamics described above. After the soil carbon equilibrium phase of 2310 simulation
1125 years (see section 2.5), equilibrium soil carbon pool sizes are estimated and corrected, depending on organic matter input and mean decomposition rate in each grid cell, and after another 2690 spinup simulation years, all carbon pools have reached a dynamic equilibrium. For the spinup simulation, we cyclically repeat the first 30 years of climate data input and prescribe atmospheric carbon dioxide concentrations at 278 ppm. During the spinup simulation, land use is introduced in the year 1700,
from where it expands annually according to the historic land-use data set (Fader et al. (2010) and section 3.1.2).

### 3.1.1 Climate, river routing and soil inputs

We use monthly climate data inputs on precipitation provided by the Global Precipitation Climatology Centre (GPCC Full Data Reanalysis Version 7.0, (Becker et al., 2013)), daily mean temperature
from Climatic Research Unit (CRU TS version 3.23 University of East Anglia Climatic Research Unit; Harris (2015); Harris et al. (2014)), shortwave downward radiation and net downward longwave radiation are reanalysis data from ERA-Interim (Dee et al., 2011), and the number of wet days per months are derived synthetically as suggested by New et al. (2000), which is used to allocate monthly precipitation to individual days. Precipitation are stochastically disaggregated while
preserving monthly sum, temperature linearly interpolated (Gerten et al., 2004). Besides climate information, the model is forced with invariant information on the soil texture (Nachtergaele et al., 2008) and annual information on land-use from Fader et al. (2010), but now also explicitly describing sugar cane areas (see sectio 3.1.2. For the SPITFIRE module LPJmL4 uses additional input.



Dew point temperature is approximated from daily minimum temperature (Thonicke et al., 2010).
Monthly average wind speeds are based on NCEP re-analysis data, which were regridded to CRU
(NOAA-CIRES Climate Diagnostics Center, Boulder, Colorado, USA, Kalnay et al. (1996)).

For the transport directions we use the global ($0.5° \times 0.5°$). Simulated Topological Network (STN-30) drainage direction map (Vorosmarty and Fekete, 2011). STN-30 organizes the Earth's land area into drainage basins and provides the river network topology under the assumption that each grid cell can drain into one of the eight next-neighbor cells, as well as detailed information on water reservoirs obtained from GRanD database (Lehner et al., 2011), including informations on storage capacity, total area and main purpose. Natural lakes are obtained from Lehner and Döll (2004).

### 3.1.2 Land use input

In principle, LPJmL4 can be driven by any land-use data information. As the default land-use input file, the cropping areas for each of the CFTs are taken from MIRCA2000 (Portmann et al., 2010) which is a combination of crop-specific areas from Monfreda et al. (2008) and areas equipped for irrigation from 1900-2005 from (Siebert et al., 2015). Monfreda et al. (2008) defined 175 crops and this number was reduced to 26 in MIRCA2000, (therefore, e. g., the group pulses consist of 12 individual crops). These land-use patterns that have been derived from maximum monthly growing areas per crop and grid cell have been combined and if these areas add up to more than 1, i.e. when sequential cropping systems are present, total cropland fraction was reduced to not exceed physical land area in each pixel. A more detailed description of the procedure can be found in Fader et al. (2010). After the implementation of sugar cane as a 12th annual crop that is explicitly represented in LPJmL4 (Lapola et al., 2009), the standard land-use input data set was amended by subtracting the sugar cane areas from the "others" band and implementing it as a separate input data band. All 16 input data bands (CFTs 1-12, others, managed grassland, bioenergy grass and bioenergy trees) are included four times in the data set, the first 16 bands representing purely rainfed agricultural areas, the second, third and fourth set representing irrigated areas of these land-use types for surface, sprinkler and drip irrigation respectively.

### 3.2 Standard outputs

The multiple aspects of the terrestrial biosphere and hydrosphere that are implemented in LPJmL4 allow for assessing multiple processes from natural and managed land which span ecological, hydrological and agricultural components. The consistent single modelling framework allows for analyzing interactions among these multiple sectors from local to global scale spanning seasons to centuries.

Being driven by climate and land-use data LPJmL4 can be applied to quantify both climatic and anthropogenic impacts on the terrestrial biosphere. Computed dynamics of biogeochemical and hydrological processes thus arise from vegetation dynamics in natural ecosystems under climate change





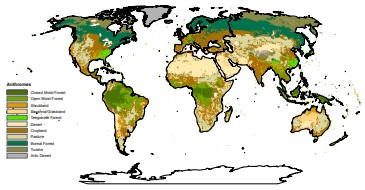

**Figure 2.** Global anthrome classes in the potential natural vegetation and the agricultural areas. Anthrome classes are defined using coverage with (a) types of dominating natural vegetation (e.g. tropical forest), (b) dominant agricultural usage (e.g. cropland) and external drivers (e.g. temperature)..

and elevated atmospheric $CO_2$ concentrations, land-use change as well as climate and management

driven changes in managed ecosystems. Each grid cell can be dominated by managed land (croplands and pastures) but still contain fractions of natural vegetation, and vice versa. We here apply the anthromes concept (cf. Ellis et al. (2010)) to illustrate the global distribution of natural vegetation and managed land as simulated by LPJmL4 (Fig. 2). We use simulated vegetation carbon, potential evapotranspiration, foliar projective cover for each PFT, managed grassland and CFT, and combine

it with climate input data to map natural biomes and anthromes at the global scale (see Boit et al. (2016) for the algorithm description). The composition of natural ecosystems is dynamically computed by LPJmL4, as the different PFTs compete with each other. Bounded by the bioclimatic limits, the modelled global distribution of forests, shrubland and natural grasslands as well as the spatial extent of polar and alpine ecosystems and deserts are in broad agreement with the biomes identified

by Olson et al. (2001). The integrated mapping of biomes and anthromes underlines the extent of anthropogenic impacts on the terrestrial biosphere and how much of the potential vegetation coverage is left. By applying the land-use input (see section 3.1.2); LPJmL4 simulates cropland in 27% and pasture in 16% of the ice-free global land area. Simulating biophysical and biochemical processes in densely populated or urban areas could be considered in future model developments of LPJmL4

given their large spatial extent (Ellis et al., 2010).

Carbon and water fluxes, productivity and harvest of crops and managed grasslands have also been quantified. Results show that soils are the largest carbon pool of the terrestrial biosphere, with highest amounts of more than 60 $kgC\,m^{-2}$ in the boreal zone, most notably in permafrost soils (see Fig. 3a and 3b). Vegetation carbon pools are largest in the tropics with almost 20 $kgC\,m^{-2}$

and in the temperate zone with about 8 $kg\,C\,m^{-2}$. The large vegetation carbon pools are a result of high net primary productivity (NPP) in tropical and subtropical ecosystems, which process about 1000 to 1200 $gC\,m^{-2}\,yr^{-1}$, respectively (see Fig. 4b). Fire carbon emissions are highest in the tropics as a result of high ignition probability and high biomass values simulated (Fig. 4c). Crop



productivity is determined by climatic conditions and management strategies and is currently highest

in the temperate zone of North America and Europe, but also in regions in eastern China, the irrigated

Ganges Valley in India and temperate South America (see Fig. 3c).





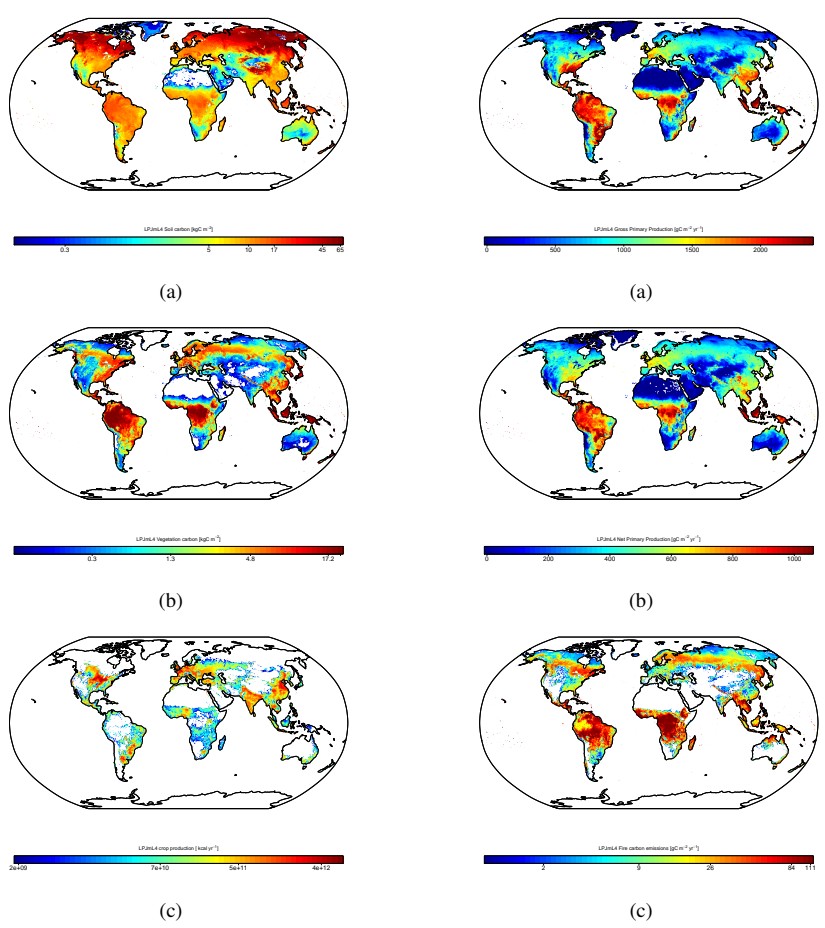

**Figure 3.** Soil (a) and vegetation (b) carbon pool and cumulative crop production (c) computed by LPJmL4 as an average of the time period 1996-2005. Note: Values plotted on a logarithmic scale.

**Figure 4.** Annual GPP (a), NPP (b) and fire carbon emissions (c) computed by LPJmL4 as an average of the time period 1996-2005. Note: Values for fire carbon emissions are plotted on a logarithmic scale.





Over the 20th century, changes in climate, land use and atmospheric $CO_2$ concentrations had distinct effects on the terrestrial carbon stocks and fluxes. Global vegetation carbon declined by 20 PgC after 1940 and is rising again since 2005, whereas carbon stored in soil and litter increased constantly over the simulation period (see Fig. 5). GPP, NPP and heterotrophic respiration also follow this trend and show considerable interannual variability, while fire-related carbon emissions declined in the 1970s and remained relatively stable thereafter. Interception and runoff also show a positive trend, while evaporation from bare soil decreased (see Fig. 5).

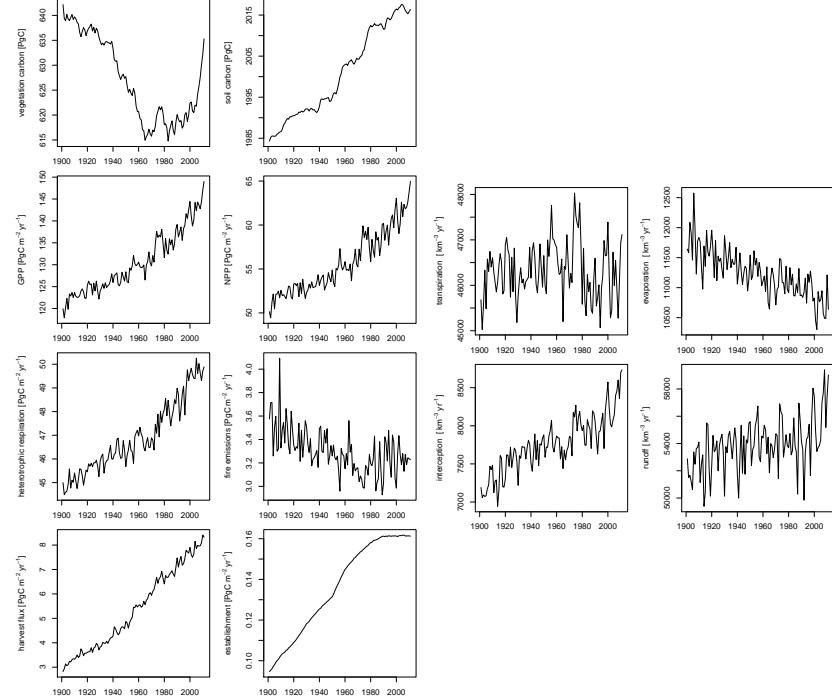

**Figure 5.** Time series of global carbon stocks and fluxes and global water fluxes computed by LPJmL4.

## 4 Discussion

Previous versions of LPJmL4 were used in a large number of applications to evaluate vegetation, water and carbon dynamics under current and future climate and land use change. In total almost a hundred papers were published since 2007 which cover a wide range of model developments and process analyses (SI Table 1) including 18 studies that describe significant model developments. The majority of the studies deal with modelling human land-use, with a focus on different crop types ($N = 54$ studies), managed grasslands ($N = 21$) and agricultural water use ($N = 18$). Most





were conducted at global scale ($N = 58$) but also at regional scales, mainly for Europe, Africa and Amazonia. Many studies ($N = 43$) investigate potential future impacts of climate change, while the remaining studies evaluate effects of current or historic climates. In the following, we will highlight some of the most important previous publications using LPJmL, describing the most prominent fields of model application.


An important field of LPJmL model application, testing and subsequent development is the analysis of historic events. LPJmL simulation results contributed to the analysis of extreme event impacts on the biosphere globally (Zscheischler et al., 2014b, a) and at pan-European scale (Rammig et al., 2015; Rolinski et al., 2015). In combination with remote sensing and eddy flux data, LPJmL has been applied to estimate ecosystem respiration (Jägermeyr et al., 2014) and productivity (e.g., Jung et al., 2008). The increasing trend in atmospheric $CO_2$ amplitude could be explained by increasing productivity in subarctic and boreal forest ecosystems and less so by increases in agricultural land and productivity (Forkel et al., 2016) as simulated building on the improved phenology scheme by Forkel et al. (2014)). The coupling of LPJmL to the climate model SPEEDY allowed for investigating climate-vegetation feedbacks from land-use change (e.g., Strengers et al., 2010; Boisier et al., 2012).



Studies on agricultural water consumption (Rost et al., 2008) and virtual water contents and water footprints for crops (Fader et al., 2010, 2011) have contributed to illuminate the role of agriculture in human water consumption.

The majority of LPJmL applications addresses the impacts of climate and land-use change on various biogeochemical and ecosystem properties of the terrestrial biosphere. To evaluate impacts of climate change on ecosystem processes and the carbon cycle at the global scale, Heyder et al. (2011) performed a risk analysis of terrestrial ecosystems based on an integrated metric that considered joint changes in carbon and water fluxes, carbon stocks and vegetation structure. Applying CMIP3 and CMIP5 climate change scenarios to LPJmL, severe impacts for the terrestrial biosphere were found when global warming levels exceed 3K local temperature in cold and tropical biomes and 4K in the temperate biome (Heyder et al., 2011; Ostberg et al., 2013). The coupling of LPJmL to the integrated assessment model IMAGE allowed to evaluate feedbacks between land-use change, the carbon balance and climate change, so that the dynamics of a potential reversal of the terrestrial carbon balance could be assessed (Müller et al., 2016). Also economic feedbacks of agricultural production were evaluated by coupling LPJmL to the agro-economic model MAgPIE (Lotze-Campen et al., 2008), for example measures of land-use protection for climate change mitigation (Popp et al., 2014).




Regional climate-change applications investigated the role of $CO_2$ fertilization on Amazon rainforest stability (Rammig et al., 2010), and analyzed additional threats arising from tropical deforestation (Gumpenberger et al., 2010; Poulter et al., 2010). Boit et al. (2016) applied the anthromes concept to LPJmL simulation results to differentiate the relative importance of future climate vs. land-use change in Latin America. In that study, land-use change was identified as the main driver




of biome shifts and biome degradation early in the 21st century while climate change impacts would dominate the second half. LPJmL simulations also showed that in the boreal forest and Arctic ecosystems, 100 years of future climate change might destabilize carbon stored in permafrost soils over several centuries due to feedbacks between permafrost and dynamic vegetation (Schaphoff et al., 2013).

Several studies used LPJmL to investigate water limitations of natural ecosystems Gerten et al. (2013) and of food production in particular (e.g., Gerten et al., 2008; Biemans et al., 2011, 2013). Gerten et al. (2011) applied LPJmL to quantify "green" and "blue" water requirements for future food security, finding that water scarcity will increase in many countries, as confirmed by other studies prepared in the context of multi-model intercomparisons such as ISIMIP (e.g., Schewe et al., 2014). In the context of planetary boundaries, Gerten et al. (2013) and Steffen et al. (2015) have proposed sub-global modifications to the planetary boundary for freshwater use. Future climate change effects on irrigation requirements were investigated by Konzmann et al. (2013); Jägermeyr et al. (2015); Fader et al. (2016).

LPJmL simulations have also shown (Asseng et al., 2015; Müller and Robertson, 2014; Rosenzweig et al., 2014) that climate change constitutes a major threat to agricultural productivity, especially in the tropics, but with large uncertainties regarding the benefits from elevated atmospheric $CO_2$ on crop water use and productivity (Müller et al., 2015; Deryng et al., 2016). It was also shown that the potential for major cereal crop production may decline in a future climate (Pugh et al., 2016b).

Building on the development of bioenergy plantations (Beringer et al., 2011), LPJmL was applied to analyse synergies and trade-offs of biomass plantations finding that demands for future bioenergy potentials implicit in climate mitigation and climate engineering portfolios could only be met at substantial environmental costs (Boysen et al., 2016; Heck et al., 2016).

The present publication describing the LPJmL4 model code and the companion paper (Schaphoff et al., submitted) providing a thorough model evaluation are intended to serve as a comprehensive description of the current LPJmL4 model. The model code will be published under an open source license on a Gitlab server. We hope that this will help to promote further development and improvement of LPJmL4 and to foster high-profile research in areas such as multi-sectoral climate change impacts, earth system dynamics, planetary boundaries, and SDGs. Besides the ongoing implementation of the dynamics of major nutrients, such as nitrogen, there are new plant physiological insights that have not yet found their way into LPJmL4 or related models (Pugh et al., 2016a; Rogers et al., 2017). The consistent coverage of natural and managed ecosystems as well as the full carbon and water dynamics linked by vegetation dynamics and land-use management is central to further development of LPJmL4.





## 5 Code and data availability

Pending the publication of the present paper, we will make the model code of LPJmL4 through a
1295 Gitlab repository publicly available. Additionally, we will provide data from the model simulation
used here in a research data repository (see http://dataservices.gfz-potsdam.de/portal/).

*Acknowledgements.* This study was supported by the German Federal Ministry of Education and Research's
(BMBF's) "PalMod 2.3 Methankreislauf, Teilprojekt 2 Modellierung der Methanemissionen von Feucht- und
Permafrostgebieten mit Hilfe von LPJmL", (FKZ 01LP1507C). A.R. and J.H. acknowledge funding from the
1300 Helmholtz Alliance "Remote Sensing and Earth System Dynamics. We thank the Climatic Research Unit for
providing global gridded temperature input, the Global Precipitation Climatology Centre for providing pre-
cipitation input and the coordinators of ERA-Interim for providing shortwave downward radiation and net
downward longwave radiation. Furthermore, we thank the authors of MIRCA2000 for providing land use in-
put. We thank Jannes Breier and Nele Steinmetz for editorial help. Finally, the many people involved in the
1305 development, testing and application of LPJmL4 and its predecessors are gratefully acknowledged.



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





**Table 1.** Abbreviation of PFTs, BFTs and CFTs.

| | |
|---|---|
| Tropical broadleaved evergreen tree | TrBE |
| Tropical broadleaved raingreen tree | TrBR |
| Temperate needleleaved evergreen tree | TeNE |
| Temperate broadleaved evergreen tree | TeBE |
| Temperate broadleaved summergreen tree | TeBS |
| Boreal needleleaved evergreen tree | BoNE |
| Boreal broadleaved summergreen tree | BoBS |
| Boreal needleleaved summergreen tree | BoNS |
| Tropical herbaceous | TrH |
| Temperate herbaceous | TeH |
| Polar herbaceous | PoH |
| Bioenergy tropical tree | BTrT |
| Bioenergy temperate tree | BTeT |
| Bioenergy C4 grass | BGrC4 |
| Temperate cereals | TeCer |
| Rice | Rice |
| Maize | Maize |
| Tropical cereals | TrCer |
| Pulses | Pul |
| Temperate roots | TeRo |
| Tropical roots | TrRo |
| Sunflower | SunFl |
| Soy-bean | Soy |
| Groundnut | GrNu |
| Rapeseed | Rape |
| Sugar-cane | SuCa |