# Peer review of "LPJmL4 – a dynamic global vegetation model with managed land: Part I – Model description"

_Geoscientific Model Development, 2017_

## Short Comment (SC1) · 5 Aug 2017

Sibyll

Thanks for your manuscript.

GMD has the policy that access to the program code needs to be provided when the manuscript is submitted. The access needs to be permanent which can be via a DOI for the particular version been references in the manuscript (e.g. using Zenodo) or via uploading the program code as a supplement. This policy guarantees the reproducibility of the results for the reviewer as well as the reader. For your manuscript DOI is the preferable option as the model version is also referenced by gmd-2017-146.

I would also like to suggest that you refer to the described model as LPJmL4.0 in the

title and the manuscript. This takes in consideration the fact that you may update the model code after submission but allows referencing back to the version used for the manuscript.

Finally I am strongly encourage you to include a statement the license of use as well as in the program code.

Many thanks and all the best

Lutz Gross GMD executive editor

---

## Author Comment (AC1) · 17 Aug 2017

Dear Dr. Gross,

thanks for watching out that GMD code availability standards are met. Indeed, the code availability passage in the paper is a bit misleading. We are in the process of setting up a publicly accessible open-source model code repository at https://gitlab.pik-potsdam.de/ where the code as described here will be the first release version. Due to legal and formal issues, this process is still ongoing. The URL to this repository as well as the DOI to the archived code snapshot will be supplied in the final version of this paper. In the meantime, we will certainly make the model code available to editors and reviewers and the code has already been supplied to the editorial office and Dr. Hargreaves, the topical editor of this paper. With making the code available to reviewers and editors, we meet the minimum requirements of GMD, see https://www.geoscientific-model-development.net/about/code_and_data_policy.html, but are working on a better solution.

Best regards, Sibyll Schaphoff

––––––––––––––––––––––––––––––––––––––

---

## Referee Comment (RC1) · Anonymous Referee #1 · 12 Sep 2017

Dear authors

The manuscript "LPJmL4 - a dynamic global vegetation model with managed land: Part I – Model description" is suitable for Geoscientific Model Development. This model could contribute the broader science including earth system modeling, climate science, atmospheric chemistry and so on. Authors tried to make new version of LPJmL model for global carbon, water, and energy cycling. I agree this model is quite important tool to assess the anthropogenic activities in global C cycling. The manuscript is well written and model is enough described even in the current version. I appreciate all efforts to describe such big model. Honestly speaking, I'm sorry that I cannot follow all of the topics and processes implemented in this model. So, perhaps, I overlooked fatal errors

in mathematical formulations in this text.

From the view to description paper in GMD, I can almost recommend acceptance for the publication. However, some points are needed to improve in the model description before the publication .

**Major comments**

I don't have any strong objection for this data processing and the products. Another concerns are as follow;

**Summary table** Please add tables for the inputs (and outputs) variables of LPJmL4.

**Mathematical expression** If possible, in equations, please use italic font for the parameters and roman font for inputs and predictive variables. Generally, in this manuscript, the mathematical expression is according to this rule. But, some parameters and variables are not (e.g., $T_{soil}$ should be replaced as $\mathrm{T}_{\mathrm{soil}}$. $H$ should be replaced as H. $t_{\mathrm{fire}}$ should be $\mathrm{t}_{\mathrm{fire}}$. Equations for crop model are not entirely followed this rule.).

**Parameter and input variables** In some parameters and variable, there are no units in the text and SI table (e.g., $\Gamma*$, $[O_2]$, $\mathrm{V}_m$, Michaelis-Menten constants, LA, , SA, H, D, $\mathrm{mort}_{\mathrm{heat}}$, $n_{h,ig}$, $\mathrm{TW}_{\mathrm{PFT}}$, phu, $\mathrm{hi}_{\mathrm{opt}}$), even though the author showed some of them in SI table. But, for the readability, I recommend to specify these units in all parameters also in the text, as much as you can.

**Figures** The letters in the figure are too small to read. Please enlarge all letters in Fig.1–5.

**Individual comments**

**Introduction** A short descriptions of LPJmL (i.e., history of LPJmL model) is needed in this section, even though the detail information in 2nd section and discussion.

**L24–26** Could you clarify and add the reference for this sentence? Le Quéré et al. (2015) have just described carbon budget.

**L31** SDGs is more appropriate.

**L34** No citation in the reference list.

**L47** Could you clarify "improve the DGVMs' skills" in the text?

**L153** "Celsius degrees" -> "°C"

**L152–154** Please add the definition of "ni (the proportion of bright sky)" among L152–154. To me, "ni" is confusing with "NI (Nesterov index)".

**L152–198** "1 and 1 m depth" -> "1 and 2 m depth"?

**L233** Are there any reference for sublimation rate of snow.

**L360–361** Are there any reference for the growth respiration parameter $r_{gr}$.

**L448–449** Please add the equation for new sapling rates, here.

**L445–449; Establishment** I can't understand the rate of establishment. Is this "per month (day)" or "per year"? If "per year", which seasons are new saplings introduced in each grid cell.

**L454–460; Background mortality** Same as above. Which timings are plants died in the model? Is this uniform rate during a year?

**L466–469** Are there any reference for the heat damage function?

**L679; Eq 82** Why $\mathrm{lai}_{\mathrm{inc}}$ don't have time step subscription $t$?

**L713–716** Are there any reference for the fast and slow fraction of the residue?

**L731; Eq 93** I guess that a significant figure in coefficients of Eq. 93 is too much.

**L786–790** Please clarify the units in parameters.

**Eq. 117 and 119** There is no definition of $W_{\mathrm{fc}}$ in the text. Perhaps, just after the "field capacity" in L919 is appropriate insert place for $W_{\mathrm{fc}}$.

**L975** "changes in soil water and soil carbon are computed separately". How to deal spin-up period among different stands (especially between crops and natural vegetations).

**L1010** I cannot understand the meaning of "In order to simulate a reasonable global distribution of temperate and tropical regions". For??

**L1014; SI-Fig. 3** Please clarify the climate data used for making this map (CRU TS?)?

**L1013–1019** Is this definition appropriate also in the projection period?

**L1207–1214; Fig 5** These results and figures are not very impressive and not informative to see model performance. At least, it ie needed to focus the topic (e.g., just see fire dynamics).

**L1216–1223** Very interesting information. Could you add citation for some of representative papers in each studies deal ?

**Fig 1** Could you highlight major update processes of LPJmL 4 in this figure?

---

## Referee Comment (RC2) · Anonymous Referee #2 · 17 Oct 2017

The submitted manuscript provides a very comprehensive description of the Dynamic Global Vegetation Model (DGVM) LPJmL covering both, natural and agricultural vegetation. Besides recent development, the manuscript also provides an historic overview of the models core components since its origins, worth reading for scientists working with other LPJ derived models, too. Although all individual processes described here can be looked up in the respective papers, this manuscript provides an overview, combining all of these processes. Since the model source code will be made publicly available, this manuscript will be the reference for that code. After clarifying my comments below, I recommend the paper for publication.

[Figure]

Comments

1. To my knowledge "Figure" should be abbreviated.

2. Abstract: Instead of mentioning the number of publications, I would prefer having a strong statement over what a broad range of research fields LPJmL was applied to, so far, summarizing the Discussion section in one or two sentences.

3. Line 134, 138 (Eq. 3, 5): In Prentice et al. (1993) the variables $\lambda$ and $\gamma$ were taken from tables, where do the equations now come from? Are they common knowledge, not needing a reference anymore?

4. Line 182 (Eq. 17):

   - Maybe rename F$_{bare}$ to FPC$_{bare}$, otherwise it is confusing with F$_{snow}$. I guess FPC$_{bare}$ should be:

   $$FPC_{bare} = 1 - \sum_{PFT=1}^{n_{PFT}} FPC_{PFT} \qquad (1)$$

   since it is not mentioned explicitly.

   - Isn't the index "PFT" missing for $FPC$? I would prefer having the FPC$_{bare}$ part infront of the sum, otherwise one could think it is part of the sum:

   $$\beta = FPC_{bare} \cdot (...) + \sum_{PFT=1}^{n_{PFT}} \beta_{PFT} \cdot FPC_{PFT} \qquad (2)$$

5. Line 194: Reorder the sentences, so that soil layer is explained before its first usage and/or refer to Fig. 1:

[Figure]

[...] in LPJ (Beer et al., 2007). The soil column is divided into five hydrological active layers of 0.2 0.3, 0.5, 1 and 1 m depth ($\Delta z$) (see section 2.6.1). Soil temperatures ($T_{soil}$) for each layer are [...]

I guess the thermal and hydrological layers are identical, without the later mentioned thermal buffer.

6. line 202: Is it also possible to use another soil texture database, since in my experience HWSD is not as "harmonized" as the name implies?

7. Line 423ff: Is the index "$ind$" in these equations identical to "PFT" as in all previous equations, since LPJmL is a "big leaf" model and not a gap model? If so, please use the same indices throughout the manuscript. And in Eq. 52/53 isn't the index "PFT" missing for SLA?

8. Line 456: Where is the "mean PFT longevity", I only see the growth efficiency mortality here.

9. Line 182ff, 423ff, 746, 823, 1090: Be consistent in how you name your indices in the equations if they have the identical meaning, please.

Technical and minor comments

- line 40: change "interferencces" to "interferences"

- Line 207/208 and 215: Replace the the second and third author by "et al."

- Line 292/293 (Eq. 34): Display as fraction without "/" for better readability and to avoid the linebreak in the equation.

- Line 1228: Why is the ordering "b, a" in Zscheischler et al., 2014?

- Line 1234: Remove the second ")".

---

## Author Response (AR1)

We thank Anonymous Referee 1 for supporting the efforts of describing the LPJmL4 model as a whole. We have tried to include all important processes represented by the model. Additionally, we really appreciate the great effort to comment on such a voluminous model description, on which we reply below. Line numbers refer to the marked-up version of the manuscript.

*"The manuscript "LPJmL4 - a dynamic global vegetation model with managed land: Part I – Model description" is suitable for Geoscientific Model Development. This model could contribute the broader science including earth system modeling, climate science, atmospheric chemistry and so on. Authors tried to make new version of LPJmL model for global carbon, water, and energy cycling. I agree this model is quite important tool to assess the anthropogenic activities in global C cycling. The manuscript is well written and model is enough described even in the current version. I appreciate all efforts to describe such big model. Honestly speaking, I'm sorry that I cannot follow all of the topics and processes implemented in this model. So, perhaps, I overlooked fatal errors Discussion paper in mathematical formulations in this text. From the view to description paper in GMD, I can almost recommend acceptance for the publication. However, some points are needed to improve in the model description before the publication ."*

*"Major comments*

*I don't have any strong objection for this data processing and the products. Another concerns are as follow;*

*Summary table Please add tables for the inputs (and outputs) variables of LPJmL4."*

We now provide a table for inputs and outputs in the SI. Nevertheless it is not possible to provide a full list of possible outputs, as that would include all potential variables (hundreds) that the model computes internally. Therefore we decided to provide the list of outputs we will make available via the Online-Database http://pmd.gfz-potsdam.de/portal/.

*"**Mathematical expression** If possible, in equations, please use italic font for the parameters and roman font for inputs and predictive variables. Generally, in this manuscript, the mathematical expression is according to this rule. But, some parameters and variables are not (e.g., $T_{soil}$ should be replaced as $\mathrm{T_{soil}}$ . H should be replaced as H. $t_{fire}$ should be $t_{fire}$ . Equations for crop model are not entirely followed this rule.)."*

This is a question of the journals style. We have reassured with the editorial office of GMD whether our present notation style meets the requirements of GMD and we got the confirmation that this is the case. Nonetheless, we have checked the full text to be consistent, but the final style will be proved during the typesetting process anyway.

*"**Parameter and input variables** In some parameters and variable, there are no units in the text and SI table (e.g., $\Gamma*$, $[O_2$ $]$, $V m$ , Michaelis-Menten constants, $\mathrm{LA}$, , $\mathrm{SA}$,  H, D, $\mathrm{mort}_{heat}$ , $n_{h,ig}$ , $\mathrm{TW_{PFT}}$ ,*

phu, hi $_{opt}$*), even though the author showed some of them in SI table. But, for the readability, I recommend to specify these units in all parameters also in the text, as much as you can."*

Thanks for making us aware of that. We now ensure that the units are given in the text.

*"**Figures** The letters in the figure are too small to read. Please enlarge all letters in Fig.1–5."*

In the revised version we have enlarged the figure legends as much as possible to improve legibility.

*"**Individual comments***

*Introduction A short descriptions of LPJmL (i.e., history of LPJmL model) is needed in this section, even though the detail information in 2nd section and discussion."*

We have extended the introduction as for the history of LPJ (Sitch et al. (2003)), the first model version, and the first version of LPJmL including agricultural land use (Bondeau et al. 2007), see L.: 66-69.

*"**L24–26** Could you clarify and add the reference for this sentence? Le Quéré et al. (2015) have just described carbon budget."*

We have added a respective reference to underpin this statement (L.: 30).

*"**L31** SDGs is more appropriate."*

Thanks, we have changed that.

*"**L34** No citation in the reference list."*

Thanks, reference have been added.

*"**L47** Could you clarify "improve the DGVMs' skills" in the text?"*

Indeed it was not clear what is meant here. We have rephrased this sentence (L.: 52-53).

*"**L153** "Celsius degrees" -> " ∘ C" "*

Done.

*"**L152–154** Please add the definition of "ni (the proportion of bright sky)" among L152– 154. To me, "ni" is confusing with "NI (Nesterov index)"."*

Thanks for this hint, we have properly defined it now (L.: 164-165). We'd like to point out that we have tried to use as few duplicate variable names as possible, but due to the amount

of variables and the reproducibility in the code (which uses the variable names denoted here) in rare cases we stuck to some variable names, even if they were used twice.

*"**L152–198** "1 and 1 m depth" -> "1 and 2 m depth"? "*

No, here the depth of the respective layer is meant, each being 1m. We rephrased this sentence to clarify this misapprehension (L.:212).

*"**L233** Are there any reference for sublimation rate of snow."*

We have added a reference which supports our assumption of the sublimation rate (L.:250).

*"**L360–361** Are there any reference for the growth respiration parameter $r_{gr}$."*

We have added a reference (L.:385).

*"**L448–449** Please add the equation for new sapling rates, here."*

We have added the respective equation, now eq. 58.

*"**L445–449;** Establishment I can't understand the rate of establishment. Is this "per month (day)" or "per year"? If "per year", which seasons are new saplings introduced in each grid cell."*

Even though we have already indicated in the first sentence of the paragraph that establishment occurs each year we have added the unit to be more precise. To avoid high establishment fluxes the model assumes distribution of new saplings over the globe during the year, see L.: 476, 482-483.

*"**L454–460;** Background mortality Same as above. Which timings are plants died in the model? Is this uniform rate during a year?"*

Thanks, we have added the unit here as well (L.: 487).

*"**L466–469** Are there any reference for the heat damage function?"*

Yes , we have added the reference for the heat damage function and a reference for the evidence of heat induced tree mortality (L.: 498, 500).

*"**L679; Eq 82** Why $\text{lai}_{inc}$ don't have time step subscription t?"*

Thanks, you are right, we have changed that accordingly, now eq. 85.

*"**L713–716** Are there any reference for the fast and slow fraction of the residue?"*

Thanks, we have added a reference (L.: 751).

*"**L731; Eq 93** I guess that a significant figure in coefficients of Eq. 93 is too much."*

We have constrained it to less decimal places, now eq. 96.

*"**L786–790** Please clarify the units in parameters."*

Done.

**"Eq. 117 and 119** There is no definition of $W_{fc}$ in the text. Perhaps, just after the "field capacity" in L919 is appropriate insert place for $W_{fc}$ ."*

Thanks, we have inserted the definition there (L.: 956).

*"**L975** "changes in soil water and soil carbon are computed separately". How to deal spin-up period among different stands (especially between crops and natural vegetations)."*

The first spin-up is done for natural vegetation only, as we need the carbon 'history' for all vegetation stands. In this spin-up phase the model simulates only one stand – natural vegetation. A second spin-up phase is done including land use change since 1700 (with simulation years 1610–1699 using the information for 1700), to take into account the effects of the land use history on agricultural land as well, but still without climate change. This is followed by the transient runs from 1901-2011, see revised section: "3.1
 Model setup and inputs".

*"**L1010** I cannot understand the meaning of "In order to simulate a reasonable global distribution of temperate and tropical regions". For??"*

Thanks, we have rephrased this sentence (L.:1047).

*"**L1014**; SI-Fig. 3 Please clarify the climate data used for making this map (CRU TS?)?"*

The seasonality type is calculated by LPJmL4, so it uses the same input as the other results. We changed the caption to make that clear.

*"**L1013–1019** Is this definition appropriate also in the projection period?"*

We do not show projected sowing dates in this paper, but with future climate projections sowing dates as simulated by the model would change with changing temperature and rainfall according to the rules defined here. Climate is not the only driver of planting times and sowing dates but it has been shown that for large areas sowing dates can be simulated very well just considering climate as a driver. Comparing simulated sowing dates to a global crop calendar showed that for 60% of global cropland the deviation is less than 1 month and for 80% it's less than 2 months, except for rapeseed (Waha et al. 2012). We added a sentence for possible adaptation of sowing dates in the future (L.: 1070-1071).

*"**L1207–1214**; Fig 5 These results and figures are not very impressive and not informative to see model performance. At least, it ie needed to focus the topic (e.g., just see fire dynamics)."*

We have moved these figures to the SI (Fig.: S5) as they are indeed not a requirement for the main text. But we want to illustrate that the model represents dynamics of key processes for the last 100 years.

*"**L1216–1223** Very interesting information. Could you add citation for some of representative papers in each studies deal ?"*

In L. 1267-1269, we state that we will highlight some of the most important previous publications for which the references are given in the following paragraph (L. 1270ff). In order to make clear that references for all publications are given in the supplementary Table 1, we have added in L.1262 "(see references in SI Table 1)".

*"**Fig 1** Could you highlight major update processes of LPJmL 4 in this figure?"*

As we want to describe LPJmL4 in its entirety as it is, using the figure to illustrate the complexity of the model. We'd like to avoid highlighting special processes.
We thank Anonymous Referee 2 for the constructive review. We reply to the comments below. Line numbers refer to the marked-up version of the manuscript.

*"The submitted manuscript provides a very comprehensive description of the Dynamic Global Vegetation Model (DGVM) LPJmL covering both, natural and agricultural vegetation. Besides recent development, the manuscript also provides an historic overview of the models core components since its origins, worth reading for scientists working with other LPJ derived models, too. Although all individual processes described here can be looked up in the respective papers, this manuscript provides an overview, combining all of these processes. Since the model source code will be made publicly available, this manuscript will be the reference for that code. After clarifying my comments below, I recommend the paper for publication."*

*"1. To my knowledge "Figure" should be abbreviated."*

We have checked the journal's style guidelines and earlier papers how figures are commonly referred to. We found both variants, but in most cases "Figure", so we have decided to keep it as it is.

*"2. Abstract: Instead of mentioning the number of publications, I would prefer having a strong statement over what a broad range of research fields LPJmL was applied to, so far, summarizing the Discussion section in one or two sentences."*

Thanks, we have added a description about the different possibilities for which LPJmL has been applied and the recent development within the model.

*"3. Line 134, 138 (Eq. 3, 5): In Prentice et al. (1993) the variables λ and γ were taken from tables, where do the equations now come from? Are they common knowledge, not needing a reference anymore?"*

We have added a statement (L.143-144) how these variables were derived.

"4. Line 182 (Eq. 17):

• *Maybe rename* F $_{bare}$ *to* FPC $_{bare}$ *, otherwise it is confusing with* F $_{snow}$ *. I guess* FPC $_{bare}$ *should be:*

$$FPC_{bare} = 1 - \sum_{PFT=1}^{n_{PFT}} FPC_{PFT} \qquad (1)$$

*since it is not mentioned explicitly.*

*• Isn't the index "PFT" missing for* FPC*? I would prefer having the* FPC bare

*part in front of the sum, otherwise one could think it is part of the sum:*

$$\beta = FPC_{bare} \cdot (...) + \sum_{PFT=1}^{n_{PFT}} \beta_{PFT} \cdot FPC_{PFT} \qquad (2)$$

*"*

> We avoid to rename $F_{bare}$ to $FPC_{bare}$, as FPC represents the foliage projective cover, which has not the meaning of bare ground. But we have added the equation you mentioned to show the meaning of $F_{bare}$ . You are totally right with the second point, we have added the index here (eq. 17).

*"5. Line 194: Reorder the sentences, so that soil layer is explained before its first*
*usage and/or refer to Fig. 1:*
*Discussion paper[...] in LPJ (Beer et al., 2007). The soil column is divided into five hydrological active layers of 0.2 0.3, 0.5, 1 and 1 m depth ($\Delta z$) (see section 2.6.1). Soil*
*temperatures (T soil ) for each layer are [...]*
*I guess the thermal and hydrological layers are identical, without the later mentioned thermal buffer."*

> Thanks, we have reordered and rephrased this paragraph for a better understanding and we have added the reference to Figure 1 (L:. 208-215).

*"6. line 202: Is it also possible to use another soil texture database, since in my*
*experience HWSD is not as "harmonized" as the name implies?"*

> Yes, one can use any input. Here we only want to present the functionality of the model and some standard inputs.

*"7. Line 423ff: Is the index "ind" in these equations identical to "PFT" as in all previous*
*equations, since LPJmL is a "big leaf" model and not a gap model? If so,*
*please use the same indices throughout the manuscript. And in Eq. 52/53 isn't*
*the index "PFT" missing for SLA?"*

> No, the index ind means an average individual representative for a specific PFT, which is not equal to PFT. We have added the definition as it was not given in the paper. We have revised the allocation section to make indices consistent (L.: 422-467).

*"8. Line 456: Where is the "mean PFT longevity", I only see the growth efficiency mortality here."*

> We are very grateful for this comment. This paragraph is now rephrased and we have added the equation for growth efficiency to explain the relations fully (L.:488).

*"Line 182ff, 423ff, 746, 823, 1090: Be consistent in how you name your indices in*

*the equations if they have the identical meaning, please."*

> Thanks, we have tried to be consistent in all indices. We went carefully through the documentation again to ensure a consistent representation of all equations.

*"Technical and minor comments*

*• line 40: change "interferencces" to "interferences"*

*• Line 207/208 and 215: Replace the the second and third author by "et al."*

*• Line 292/293 (Eq. 34): Display as fraction without "/" for better readability and to*

*avoid the linebreak in the equation.*

*• Line 1228: Why is the ordering "b, a" in Zscheischler et al., 2014?*

*• Line 1234: Remove the second ")".   "*

> Thank you, we took all suggestions if they comply with the journals style.

[revised manuscript text omitted]

see eq. (46) : for $R_{\text{leaf}}$ and $A_{\text{dt}}$ is

$$A_{\text{dt}} = A_{\text{nd}} - R_d \tag{38}$$

The photosynthesis rate can be related to canopy conductance ($g_c$ in $\text{mm s}^{-1}$) through the $CO_2$ diffusion gradient between the intercellular air spaces and the atmosphere:

$$g_c = \frac{1.6 A_{\text{dt}}}{p_a \cdot (1 - \lambda)} + g_{\min}, \tag{39}$$

where $g_{\min}$ [$\text{mm s}^{-1}$] is a PFT-specific minimum canopy conductance scaled by FPC that occurs due to processes other than photosynthesis, . Combining both methods determining $A_{\text{dt}}$ (eqs. 38, 39) gives:

$$0 = A_{\text{dt}} - A_{\text{dt}} = A_{\text{nd}} + (1 - \text{daylength}/24) \cdot R_{\text{
[revised manuscript text omitted]
_{\sim\sim sapwood\sim\sim sapwood,ind} \cdot k_{la:sa}}{WD \cdot C_{\sim leaf\sim leaf,ind} \cdot SLA}\tag{53}$$

Stem diameter can then be calculated by inverting eq. (50). Leaf area is related to leaf biomass  $C_{leaf,ind}$ by PFT-specific SLA, thus, the individual leaf area index ($LAI_{ind}$) is given by:

$$LAI_{ind} = \frac{\sim C_{leaf} \cdot SLA \sim}{\sim CA \sim} \frac{C_{leaf,ind} \cdot SLA}{CA_{ind}}\tag{54}$$

450 SLA is related to leaf longevity ($\alpha_{leaf}$) in month (see SI-Table S6), which determines whether deciduous or evergreen phenology suits a given climate suggested by Reich et al. (1997). The equation is based on the form suggested by Smith et al. (2014) for needleleaved and broadleaved PFTs as follows:

$$SLA = \frac{2 \times 10^{-4}}{DM_C} \cdot 10^{\beta_0 - \beta_1 \cdot \log(\alpha_{leaf})/\log(10)}\tag{55}$$

455 The parameter $\beta_0$ is adapted for broadleaved ($\beta_0 = 2.2$) and needleleaved trees ($\beta_0 = 2.08$) and for grass ($\beta_0 = 2.25$) and $\beta_1$ is set to $0.4$. Both parameters were derived from data given in Kattge et al. (2011). The dry matter carbon content of leaves $DM_C$ is set to $0.4763$ obtained from Kattge et al. (2011). $LAI_{ind}$ can be converted into foliar projective cover ($FPC_{ind}$, which is the proportion of ground area covered by leaves) using the canopy light-absorption model (Lambert-Beer law, Monsi

460 (1953)):

$$FPC_{ind} = 1 - \exp(-k \cdot LAI_{ind}),\tag{56}$$

where $k$ is the PFT specific light extinction coefficient (see SI-Table S5). The overall FPC of a PFT in a grid cell is obtained by the product $LAI_{ind}$, mean individual $CA_{ind}$, and mean number of

individuals per unit area ($P$), which is determined by the vegetation dynamics (see section 2.3.2).

$$\text{FPC}_{\text{PFT}} = \text{CA}_{\text{ind}} \cdot P \cdot \text{FPC}_{\text{ind}} \tag{57}$$

 $\text{FPC}_{\text{PFT}}$ directly measures the ability of the canopy to intercept radiation (Haxeltine and Prentice, 1996).

**2.3.2 Vegetation dynamics**

**Establishment**

For PFTs within their bioclimatic limits ($T_{\text{c,min}}$, see SI-Table S4), each year, new woody PFT individuals and herbaceous PFTs can establish depending on available space. Woody PFTs have a maximum establishment rate $k_{\text{est}}$ of 0.12 (saplings m$^{-2}$ a$^{-1}$), which is a medium value of tree density for all biomes (Luyssaert et al., 2007). New saplings can establish on bare ground in the grid cell that is not occupied by woody PFTs. Establishment rate of tree individuals is calculated:

$$\text{EST}_{\text{TREE}} = k_{\text{est}} \cdot (1 - \exp(-5 \cdot (1 - \text{FPC}_{\text{TREE}}))) \cdot \frac{(1 - \text{FPC}_{\text{TREE}})}{n_{\text{
[revised manuscript text omitted]

S. Schaphoff et al.

*Correspondence to:* Sibyll.Schaphoff@pik-potsdam.de

**S1    Supplementary informations to the description of the LPJmL4 model**

Fig. S1 gives a schematic overview of the model structure represented in LPJmL4. Fig. S2 to S4 provides further information of implemented processes in LPJmL4. Global time series of some key parameter estimated by LPJmL4 is given in Fig. S5, these time series of carbon stocks and fluxes and
5    water fluxes illustrate the high dynamic of the different parameter between the years. Furthermore, we provide a list of applications which have used the LPJmL model (Table S1). This represents not a complete list of all references with LPJmL applications, but it illustrates the range of fields for topical, spatial and temporal use of the model. Table S2 gives an overview of input variables and their references used by LPJmL4 here. Additionally we give a list of output variables (see Table S3)
10    computed by LPJmL4 and provided via the Online-Database: http://pmd.gfz-potsdam.de/portal/. Complementary to the associated Schaphoff et al. (under Revision) we give a comprehensive list of parameters (Tables S4 to S14) used by the model and are described in Schaphoff et al. (under Revision). Additionally, we provide a list of equations (Table S15), which are described in detail by the associated manuscript.

[Figure]

**Figure S1.** Flowchart describing the order of processes which are represented in the LPJml4 model.

[Figure]

**Figure S2.** Irrigation water flows in LPJmL4 from plant-specific net irrigation requirement to actual field application. Variables represented in grey-shaded boxes depend on system-specific parameters that are presented in Table 2, adopted from Jägermeyr et al. (2015).

[Figure]

no seasonality

temperature seasonality

precipitation seasonality

both seasonalities and temperature of coldest month <= 10°C

both seasonalities and temperature of coldest month > 10°C

**Figure S3.** Seasonality types for sowing date calculation by LPJmL4.

[Figure]

**Figure S4.** Leaf carbon (x-axis) that is remaining after harvest (solid line) and being harvested (between solid and dashed lines).

[Figure]

**Figure S5.** Time series of global carbon stocks and fluxes and global water fluxes computed by LPJmL4.

Table S1: Reference table of application using LPJmL since 2007.

| Paper | Ecosystem processes | | | | | Carbon cycle | | | | | | | Water cycle | | | Agriculture | | | | Temporal domain | | | Spatial domain | | | Type | | |
|---|---|---|---|---|---|---|---|---|---|---|---|---|---|---|---|---|---|---|---|---|---|---|---|---|---|---|---|---|
| | Vegetation dynamics | Permafrost | Fire | Phenology | Albedo | Photosynthesis | Respiration | Fire emissions | Land C sink | Biomass | Soil carbon | Atmospheric composition | Evapotranspiration | Runoff, discharge | Human water use | Crops | Managed grassland | Agricultural trees | Bioenergy | Historic | Present and recent past | Future climate | Global | Regional | Site-level | Development | Evaluation | Application |
| Beer et al. (2007) | x | x | x | x | | | | | | | | | | | | | | | | | x | | | x | | | x | x |
| Gerten et al. (2007) | x | | x | | | x | | | | | | | x | | | | | | | | x | x | x | | | x | | x |
| Müller and Lucht (2007) | | | | | | x | x | x | x | x | x | | x | x | | | | | | | x | | x | | | | x | |
| Müller et al. (2007) | | | | | | | | | x | | | | | | | x | | | | | | x | x | | | | | x |
| Gerten et al. (2008a) | x | | x | | | x | | | | | | | x | | | | | | | | | x | x | | | | | x |
| Gerten et al. (2008b) | | | | | | | | | | | | | | x | | | | | | | x | | | | x | | | x |
| Jung et al. (2008) | | | | | | x | | | | | | | | | | | | | | | x | | | x | | | x | x |
| Lotze-Campen et al. (2008) | | | | | | | | | | | | | | x | x | x | x | | | | | | x | | | | | x |
| Luo et al. (2008) | x | | x | | | x | x | | | x | x | | x | | | | | | | | | x | x | | x | | x | x |
| Rost et al. (2008) | | | | | | | | | | | | | x | x | x | x | x | | | x | x | | x | | | x | x | x |

| Study | Green 1 | Green 2 | Green 3 | Green 4 | Green 5 | Green 6 | Yellow 1 | Yellow 2 | Yellow 3 | Yellow 4 | Yellow 5 | Yellow 6 | Blue 1 | Blue 2 | Orange 1 | Orange 2 | Orange 3 | Orange 4 | Orange 5 | Gray 1 | Gray 2 | Gray 3 | Gray 4 | Gray 5 | Gray 6 | Gray 7 |
|---|---|---|---|---|---|---|---|---|---|---|---|---|---|---|---|---|---|---|---|---|---|---|---|---|---|---|
| Biemans et al. (2009) | | | | | | | | | | | | | x | | | | | | | x | | x | | x | | |
| Lapola et al. (2009) | | | | | | | | | | | | | | | | | | | | | | | | | | |
| Pitman et al. (2009) | x | x | | | | | | | x | | | | x | | x | | x | | | x | x | x | | | | x |
| Poulter et al. (2009) | x | | | | | | x | | | | | | | | | | | | | x | | x | | x | x | x |
| Rost et al. (2009) | | | | | | | | | | | | | x | x | x | x | | | | | x | x | | x | | x |
| Jung et al. (2010) | | | | | | | | | | | | | x | | | | | | | x | | x | | x | x | x |
| Von Bloh et al. (2010) | | | | | | | | | | | | | x | | | | | | | x | | x | x | | | x |
| Fader et al. (2010) | | | | | | | | | | | | | x | x | x | x | | | | x | x | x | x | | x | x |
| Gumpenberger et al. (2010) | x | | | | | | x | | | | | | | | | | | | | x | | x | x | | | x |
| Lotze-Campen et al. (2010) | | | | | | | | | | | | | x | | x | x | | | x | x | | x | | | | x |
| Neumann et al. (2010) | | | | | | | | | | | | | | x | x | | | | | x | | x | | | | x |
| Poulter et al. (2010a) | x | | | | | | x | x | x | | | | | | | | | | | x | | x | x | | | x |
| Poulter et al. (2010b) | x | | | | | | | x | x | x | | | | | | | | | | x | | x | x | | | x |
| Rammig et al. (2010) | x | | | | | | | | | x | | | | | | | | | | x | | x | x | | | x |
| Strengers et al. (2010) | x | x | | | | | x | x | x | x | | | x | | x | x | | | | x | x | x | x | x | x | x |
| Thonicke et al. (2010) | | x | | | | | | x | x | | | | | | | | | | | | | x | | x | x | x |
| Beringer et al. (2011) | | | | | | | | | | | | | | | | | | | x | x | x | x | | x | x | x |
| Biemans et al. (2011) | | | | | | | | | | | | | x | x | x | x | | | | x | x | x | | x | x | x |

| Reference | (green) | | | | | | (yellow) | | | | | | | | | (blue) | | | | (orange) | | | | | (grey) | | (dark) | | (light) |
|---|---|---|---|---|---|---|---|---|---|---|---|---|---|---|---|---|---|---|---|---|---|---|---|---|---|---|---|---|---|
| Fader et al. (2011) | | | | | | | | | | | | | | | | | | | x | | x | x | | | x | | x | | x |
| Franck et al. (2011) | | | | | | | | | | | | | | | | | | | | | x | | | | x | | x | | x |
| Gerten et al. (2011) | | | | | | | | | | | | | | | | | x | x | x | | x | x | | | x | | x | | x |
| Haberl et al. (2011) | | | | | | | x | x | | | | | | | | | | | | | x | | | | x | x | | | x |
| Haddeland et al. (2011) | | | | | | | | | | | | | | | | | x | x | | | | | | | x | | x | x | |
| Heyder et al. (2011) | x | | | | | | x | x | x | x | | | | | | | x | x | | | | | | | x | x | x | | x |
| Neumann et al. (2011) | | | | | | | | | | | | | | | | | x | x | | | | | | | x | | x | | x |
| Popp et al. (2011a) | | | | | | | | | | | | | | | | | | | | x | x | | | | x | x | | | x |
| Popp et al. (2011b) | | | | | | | | | | | | | | | | | | | | x | | | | | x | x | | | x |
| Poulter et al. (2011) | | | | | | | | | | x | | | | | | | | | | | | | | | x | | x | | x |
| Jiang et al. (2012) | x | | | | | | | | | | | | | | | | | | | | | | | | x | x | x | | x |
| Boisier et al. (2012) | x | | | | | | | | | | | | | | | | | | | | x | x | | | x | x | | | x |
| de Noblet-Ducoudré et al. (2012) | x | x | | | | | | | | | | | | | | | | | | | x | x | | | x | x | | | x |
| Dietrich et al. (2012) | | | | | | | | | | | | | | | | | | | | | x | | | | x | | | | x |
| Souty et al. (2012) | | | | | | | | | | | | | | | | | | | | | x | | | | x | | | | x |
| Waha et al. (2012) | | | | | | | | | | | | | | | | | | | | | x | | | | x | | x | x | x |
| Asseng et al. (2013) | | | | | | | | | | | | | | | | | | | | | x | | | | x | | | x | x |
| Biemans et al. (2013) | | | | | | | | | | | | | | | | x | x | x | x | | x | | | | x | x | | x | x |

Dass et al. (2013)

Fader et al. (2013)

Gerten et al. (2013b)

Gerten et al. (2013a)

Konzmann et al. (2013)

Langerwisch et al. (2013)

Ostberg et al. (2013)

Schaphoff et al. (2013)

Schierhorn et al. (2013)

Siderius et al. (2013)

Waha et al. (2013a)

Waha et al. (2013b)

Bassu et al. (2014)

Elliott et al. (2014)

Forkel et al. (2014)

Jägermeyr et al. (2014)

Kummu et al. (2014)

Müller and Robertson (2014)

| Study | | | | | | | | | | | | | | | | | | | | | | | |
|---|---|---|---|---|---|---|---|---|---|---|---|---|---|---|---|---|---|---|---|---|---|---|---|
| Müller et al. (2014) | | | | | | | | | | x | | | x | x | | | x | x | | x | | | x |
| Piontek et al. (2014) | x | x | | | | | | | | x | | | x | x | | | x | x | x | x | | | x |
| Rosenzweig et al. (2014) | | | | | | | | | | | | | | x | | | x | x | x | x | | | x |
| Sakschewski et al. (2014) | | | | | | | | | | | | | | x | x | | x | x | x | x | | | x |
| Zscheischler et al. (2014b) Zscheischler et al. (2014a) | | | | | | | | | x | | | | | | | | | | | x | | x | |
| Zscheischler et al. (2014a) Zscheischler et al. (2014b) | | | | | | | | | x | | | | | | | | | | | x | | x | |
| Asseng et al. (2015) | | | | | | | | | | | | | | x | | | | | x | x | | | x |
| Fader et al. (2015) | | | | | | | | | | | x | | x | x | x | | | x | x | | | | x |
| Forkel et al. (2015) | x | | | x | | | | | | | | | | | | | | | | x | | | x |
| Jägermeyr et al. (2015) | | | x | | | | | | | | | x | x | x | x | | | x | x | | x | | x |
| Kollas et al. (2015) | | | | | | | | | | | | x | | x | | | | | x | | | x | x |
| Martre et al. (2015) | | | | | | | | | | | | | | x | | | | | x | | | x | x |
| Müller et al. (2015) | | | | | | | | | | | | | | x | | | x | x | x | x | | | x |
| Ostberg et al. (2015) | x | | | | | x | x | | | | x | x | x | x | | x | x | x | | x | | | x |
| Pirttioja et al. (2015) | | | | | | | | | | | | | | x | | | x | x | x | x | | | x |
| Weindl et al. (2015) | | | | | | | | | | | | | | x | x | | x | x | x | x | | x | x |
| Cammarano et al. (2016) | | | | | | | | | | | | x | | x | | | x | x | x | x | | x | x |
| Deryng et al. (2016) | | | | | | | | | | | | | | x | | | x | x | x | x | | | x |

**Table S2.** Model specific inputs applied by LPJmL4.

| Input variables | Description | References |
|---|---|---|
| Precipitation | GPCC Full Data Reanalysis Version 7.0 | Becker et al. (2013) |
| Temperature | CRU TS version 3.23 | Harris et al. (2014); University of East Anglia Climatic Research U... |
| Net downward long-wave radiation | ERA-Interim | Dee et al. (2011) |
| Shortwave downward radiation | ERA-Interim | Dee et al. (2011) |
| Number of wet days per months | synthetically derived | New et al. (2000) |
| Wind speed | NCEP re-analysis data | NOAA-CIRES Climate Diagnostics Center, Kalnay et al. (1996) ) |
| Landuse | MIRCA2000+ (see Fader et al. (2010) ) | Portmann et al. (2010); Monfreda et al. (2008); Siebert et al. (2015... |
| Soil texture | Harmonized World Soil Database (HWSD) | FAO/IIASA/ISRIC/ISSCAS/JRC (2012); Nachtergaele et al. (2008... |
| Drainage direction map | Topological Network (STN-30) | Vorosmarty and Fekete (2011) |
| Water reservoirs | GRanD database | Lehner et al. (2011) |
| Lakes | natural lakes | Lehner and Döll (2004) |
| Atmospheric $CO_2$ concentrations | NOAA/ESRL | http://www.esrl.noaa.gov/gmd/ccgg/trends/ |

**Table S3.** Standard outputs computed by LPJmL4.

| | Variable | Units |
|---|---|---|
| Carbon pools | Soil carbon | $gC\,m^{-1}$ |
| | Litter carbon | $gC\,m^{-1}$ |
| | Vegetation carbon | $gC\,m^{-1}$ |
| | Above ground biomass | $gC\,m^{-1}$ |
| Carbon fluxes | Monthly net primary production | $gC\,m^{-1}\,month^{-1}$ |
| | Monthly gross primary production | $gC\,m^{-1}\,month^{-1}$ |
| | Monthly soil respiration | $gC\,m^{-1}\,month^{-1}$ |
| | Annual fire carbon emissions | $gC\,m^{-1}\,a^{-1}$ |
| | Monthly interception | $mm\,month^{-1}$ |

**Table S4.** Model PFT-specific bioclimatic limits similar as in Sitch et al. (2003).

| PFT | $T_{c,\text{min}}$ (°C) | $T_{c,\text{max}}$ (°C) | $T_{mort,\text{min}}$ (°C) | $GDD_{\text{min}}$ (°C) |
|---|---|---|---|---|
| TrBE | 15.5 | - | - | |
| TrBR | 15.5 | - | - | |
| TeNE | -2.0 | 22 | | 900 |
| TeBE | 3.0 | 18.8 | | 1200 |
| TeBS | -17.7 | 15.5 | | 1200 |
| BoNE | -32.5 | -2.0 | 23 | 600 |
| BoBS | - | -2.0 | 23 | 350 |
| BoNS | -46.5 | -5.4 | 23 | 350 |
| TrH | 7.0 | - | 6500 | |
| TeH | -39.0 | 15.5 | - | |
| PoH | - | -2.6 | - | |

**Table S5.** PFT-specific albedo and light extinction values.

| PFT | $\beta_{\text{leaf}}$ | $\beta_{\text{stems}}$ | $\beta_{\text{litter}}$ | $k$ | $\alpha_a$ |
|---|---|---|---|---|---|
| TrBE | 0.14 | 0.10 | 0.10 | 0.5 | 0.4 |
| TrBR | 0.13 | 0.07 | 0.060 0.06 | 0.5 | 0.4 |
| TeNE | 0.137 | 0.04 | 0.01 0.10 | 0.4 | 0.4 |
| TeBE | 0.15 | 0.04 | 0.10 | 0.5 | 0.4 |
| TeBS | 0.15 | 0.04 | 0.10 | 0.6 | 0.4 |
| BoNE | 0.13 | 0.10 | 0.10 | 0.4 0.5 | 0.4 |
| BoBS | 0.18 | 0.10 | 0.10 | 0.5 | 0.4 |
| BoNS | 0.12 | 0.05 | 0.01 | 0.6 | 0.4 |
| TrH | 0.21 | - | 0.10 | 0.4 | 0.4 |
| TeH | 0.20 | - | 0.10 | 0.4 0.5 | 0.4 |
| PoH | 0.21 | - | 0.10 | 0.4 0.5 | 0.4 |
| BTrT | 0.13 | 0.04 | 0.10 | 0.6 | 0.8 |
| BTeT | 0.14 | 0.04 | 0.10 | 0.6 | 0.8 |
| BGrC4 | 0.21 | - | 0.10 | 0.6 | 0.8 |
| All crops | 0.18 | - | 0.06 | 0.5 | 1.0 |

$\beta_{\text{leaf}}$ is leaf albedo, $\beta_{\text{stems}}$ is the albedo of stems, $\beta_{\text{litter}}$ is albedo of litter, $k$ is the light extinction coefficient in Lambert-Beer relationship, $\alpha_a$ is a scaling factor from leaf to ecosystem level (Haxeltine and Prentice, 1996). $\beta_{\text{leaf}}$ as suggested by Strugnell et al. (2001), $\beta_{\text{stems}}$ and $\beta_{\text{litter}}$ parameters are determined by a tuning process described by Forkel et al. (2014).

**Table S6.** Global parameters and constants similar as in Sitch et al. (2003) and Schaphoff et al. (2013).

| | Symbol | Value | Units | Description |
|---|---|---|---|---|
| **Energy balance** | $c_\text{water}$ | $4.2 \times 10^6$ | $\text{J m}^{-3}\,\text{K}^{-1}$ | heat capacity of water |
| | $c_\text{min}$ | $1.9259 \times 10^6$ | $\text{J m}^{-3}\,\text{K}^{-1}$ | heat capacity of mineral soil (De Vries, 1963) |
| | $c_\text{ice}$ | $2.1 \times 10^6$ | $\text{J m}^{-3}\,\text{K}^{-1}$ | heat capacity of ice |
| **Vegetation structure** | $k_\text{allom1}$ | 100 | | Parameter for allometric relation ship Eq. 50 |
| | $k_\text{allom2}$ | 40 | | Parameter for allometric relation ship Eq. 49 |
| | $k_\text{allom3}$ | 0.67 | | Parameter for allometric relation ship Eq. 49 |
| | $k_\text{la:sa}$ | 4000 | | leaf area to sapwood area Eq. 47 |
| | WD | 20000 | $\text{gC m}^{-3}$ | wood density Eq. 51 |
| | $k_\text{rp}$ | 1.6 | | Reineke parameter Eq. 50 |
| **Photosynthesis** | $[O_2]$ | 20900 | Pa | O₂ partial pressure |
| | $K_{O_{25}}$ | 30000 | Pa | Michaelis constant for O₂ at 25°C |
| | $K_{C_{25}}$ | 30 | Pa | Michaelis constant for CO₂ at 25°C |
| | $\tau_{25}$ | 2600 | | $\tau$ at 25°C |
| | $Q_{10_{K_O}}$ | 1.2 | | $Q_{10}$ for temperature-sensitive parameter $K_O$ |
| | $Q_{10_{K_C}}$ | 2.1 | | $Q_{10}$ for temperature-sensitive parameter $K_C$ |
| | $Q_{10_\tau}$ | 0.57 | | $Q_{10}$ for temperature-sensitive parameter $\tau$ |
| | $\alpha_{C3}$ | 0.08 | | intrinsic quantum efficiencies for CO₂ uptake in C3 plants |
| | $\alpha_{C4}$ | 0.053 | | intrinsic quantum efficiencies for CO₂ uptake in C4 plants |
| | $\theta$ | 0.7 | | Co-limitation (shape) parameter |
| **Plant respiration** | $b_{C3}$ | 0.015 | rate per day | leaf respiration as fraction of $V_m$ for C3 plants |
| | $b_{C4}$ | 0.035 | rate per day | leaf respiration as fraction of $V_m$ for C4 plants |
| | $CN_\text{sapwood}$ | 330 | | C:N ratios for above-ground tissue |
| | $CN_\text{root}$ | 30 | | C:N ratios below-ground tissue |
| | $r_\text{gr}$ | 0.25 | | share of growth respiration |
| | $k$ | 0.0548 | rate per day | respiration coefficient Eq. 42 (Sprugel et al., 1995) |
| **Establishment and mortality** | $k_\text{est}$ | 0.12 | saplings m$^{-2}$ | establishment rate |
| | $k_\text{mort1}$ | 0.03 | a$^{-1}$ | asymptotic maximum mortality rate |
| | $k_\text{mort2}$ | 0.2 | | coefficient of growth efficiency for mortality |
| | $\text{tw}_\text{PFT}$ | 400 | °C | Parameter for heat damage function |
| **Soil and litter decomposition** | $\tau_{10_\text{root,litter}}$ | 0.3 | a$^{-1}$ | mean residence time of roots in litter Eq. 91 |
| | $\tau_{10_\text{root,fastSoil}}$ | 0.03 | a$^{-1}$ | mean residence time of roots in fast soil carbon pool Eq. 91 |
| | $\tau_{10_\text{root,slowSoil}}$ | 0.001 | a$^{-1}$ | mean residence time of roots in slow soil carbon pool Eq. 91 |

**Table S7.** PFT-specific parameters of litter turnover rates suggested by Brovkin et al. (2012) and shape factor for vertical distribution of soil organic matter (Schaphoff et al., 2013) .

| PFT | $\tau_{10_{\text{leaf,litter}}}$ $(\text{a}^{-1})$ | $\tau_{10_{\text{wood,litter}}}$ $(\text{a}^{-1})$ | $Q_{10_{\text{wood,litter}}}$ (-) | $k_{\text{soc}}$ (-) |
|---|---|---|---|---|
| TrBE | 0.93 | 0.039 | 2.75 | 0.38009 |
| TrBR | 1.17 | 0.039 | 2.75 | 0.51395 |
| TeNE | 0.70 | 0.041 | 1.97 | 0.32198 |
| TeBE | 0.86 | 0.104 | 1.37 | 0.43740 |
| TeBS | 0.95 | 0.104 | 1.37 | 0.28880 |
| BoNE | 0.76 | 0.041 | 1.97 | 0.28670 |
| BoBS | 0.94 | 0.104 | 1.37 | 0.28670 |
| BoNS | 0.76 | 0.041 | 1.97 | 0.28670 |
| TrH | 0.97 | - | - | 0.46513 |
| TeH | 1.20 | - | - | 0.38184 |
| PoH | 1.20 | - | - | 0.38184 |
| BTrT | 0.93 | 0.039 | 2.75 | 0.38009 |
| BTeT | 0.95 | 0.104 | 1.37 | 0.28880 |
| BGrC4 | 0.97 | - | - | 0.46513 |
| All crops | 0.97 | - | - | 0.40428 |

**Table S8.** PFT-specific parameters.

| PFT | $\beta_{\mathrm{root}}$ | $g_{\mathrm{min}}$ (mm s$^{-1}$) | $\alpha_{\mathrm{leaf}}$ (a) | $\tau_{\mathrm{leaf}}$ (a) | $\tau_{\mathrm{root}}$ (a) | $\tau_{\mathrm{sapwood}}$ (a) | $r_{\mathrm{PFT}}$ gC gN$^{-1}$ day$^{-1}$ | lr$_{\mathrm{max}}$ |
|------|------|------|------|------|------|------|------|------|
| TrBE | 0.962 | 0.5 | 1.60 | 2.0 | 2.0 | 20.0 | 0.2 | 1.0 |
| TrBR | 0.961 | 0.5 | 0.50 | 1.0 | 1.0 | 20.0 | 0.2 | 1.0 |
| TeNE | 0.976 | 0.5 | 4.00 | 4.0 | 4.0 | 20.0 | 1.2 | 1.0 |
| TeBE | 0.964 | 0.5 | 1.60 | 1.0 | 1.0 | 20.0 | 1.2 | 1.0 |
| TeBS | 0.966 | 0.5 | 0.45 | 1.0 | 1.0 | 20.0 | 1.2 | 1.0 |
| BoNE | 0.943 | 0.3 | 4.00 | 4.0 | 4.0 | 20.0 | 1.2 | 1.0 |
| BoBS | 0.943 | 0.5 | 0.50 | 1.0 | 1.0 | 20.0 | 1.2 | 1.0 |
| BoNS | 0.943 | 0.5 | 0.65 | 1.0 | 1.0 | 20.0 | 1.2 | 1.0 |
| TrH | 0.972 | 0.5 | 0.40 | 1.0 | 2.0 | - | 0.2 | 0.60 |
| TeH | 0.943 | 0.5 | 0.35 | 1.0 | 2.0 | - | 1.2 | 0.60 |
| PoH | 0.943 | 0.5 | 0.35 | 1.0 | 2.0 | - | 1.2 | 0.60 |

$\beta_{\mathrm{root}}$ is the root distribution slope parameter for water availability, $g_{\mathrm{min}}$ is the minimum canopy conductance, $\alpha_{\mathrm{leaf}}$ is the leaf longevity, $\tau_{\mathrm{leaf,root,sapwood}}$ is the compartment specific turnover times, $r_{\mathrm{PFT}}$ is the respiration coefficient for maintenance respiration of sapwood and root, lr$_{\mathrm{max}}$ is the maximum leaf-to-root mass ratio

**Table S9.** PFT-specific parameters for the SPITFIRE module.

| PFT | $\alpha_p$ | $\rho_b$ | $m_e$ | $\Phi_w$ | scorch height | crown length | $r_{\mathrm{CK}}$ | $p$ |
|---|---|---|---|---|---|---|---|---|
| TrBE | 0.0000334 | 25 | 0.3 | 0.4 | 0.1487 | 0.3334 | 1.0 | 3.00 |
| TrBR | 0.0000334 | 13 | 0.3 | 0.4 | 0.0610 | 0.1000 | 0.05 | 3.00 |
| TeNE | 0.0000667 | 25 | 0.3 | 0.4 | 0.1000 | 0.3334 | 1.00 | 3.75 |
| TeBE | 0.0000334 | 22 | 0.3 | 0.4 | 0.3710 | 0.3334 | 0.95 | 3.00 |
| TeBS | 0.0000667 | 22 | 0.3 | 0.4 | 0.0940 | 0.3334 | 1.0 | 3.00 |
| BoNE | 0.0000667 | 25 | 0.3 | 0.4 | 0.1100 | 0.3334 | 1.0 | 3.00 |
| BoBS | 0.0000667 | 22 | 0.3 | 0.4 | 0.0940 | 0.3334 | 1.0 | 3.00 |
| BoNS | 0.0000667 | 22 | 0.3 | 0.4 | 0.0940 | 0.3334 | 1.0 | 3.00 |
| TrH | 0.0000667 | 2 | 0.3 | 0.6 | - | - | - | - |
| TeH | 0.0000667 | 4 | 0.3 | 0.6 | - | - | - | - |
| PoH | 0.0000667 | 4 | 0.3 | 0.6 | - | - | - | - |

$\alpha_p$ defines the slope of the probability risk function, $\rho_b$ is the fuel bulk density, $m_e$ is the moisture of extinction,$\Phi_w$ is the windspeed dampening , $r_{\mathrm{CK}}$ is the resistance factor, $p$ is the crown damage parameter

**Table S10.** Parameters for annual CFTs for the computation of variety and sowing day parameters.

| CFT | representative crop | crops represented | $phu_{w_{low}}$ | $phu_{w_{high}}$ | $phu_{s_{low}}$ | $phu_{s_{high}}$ | $T_{base_{low}}$ | $T_{base_{high}}$ | pf | $T_{fall}$ | $T_{spring}$ | $T_{vern}$ |
|---|---|---|---|---|---|---|---|---|---|---|---|---|
| temperate cereals | wheat | wheat, rye, barley | 1700 | 2876.9 | 1000 | 2648.4 | 0.0 | 0.0 | 200 | 12 | 5 | 12 |
| rice | rice | paddy rice | NA | NA | 1600 | 1800 | 10 | 10 | 167 | NA | 18 | NA |
| maize | maize | maize for food | NA | NA | 1600 | 1600 | 5 | 15 | 167 | NA | 14 | NA |
| tropical cereals | millet | millet, sorghum | NA | NA | 1500 | 1500 | 10 | 10 | 167 | NA | 12 | NA |
| pulses | field pea | pulses | NA | NA | 2000 | 2000 | 1.0 | 1.0 | 167 | NA | 10 | NA |
| temperate roots | sugar beet | sugar beet | NA | NA | 2700 | 2700 | 3.0 | 3.0 | 167 | NA | 8 | NA |
| tropical roots | cassava | cassava | NA | NA | 2000 | 2000 | 15 | 15 | 167 | NA | 22 | NA |
| sunflower | sunflower | sunflower | NA | NA | 1000 | 1600 | 6.0 | 6.0 | 167 | NA | 13 | NA |
| soybean | soybean | soybean | NA | NA | 1000 | 1000 | 10 | 10 | 167 | NA | 13 | NA |
| groundnuts | groundnuts | groundnuts | NA | NA | 1500 | 1500 | 14 | 14 | 167 | NA | 15 | NA |
| rapeseed | rapeseed | rapeseed | 2100 | 3279.7 | 1000 | 2648.4 | 0.0 | 0.0 | 200 | 17 | 5 | 12 |
| sugarcane | sugarcane | sugarcane | NA | NA | 2000 | 4000 | 11 | 15 | 167 | NA | 14 | NA |

**Table S11.** Parameters for annual CFTs for the computation of LAI development and biomass allocation.

| CFT | $\beta_{root}$ | $fphu_c$ | $flaimax_c$ | $fphu_k$ | $flaimax_k$ | $fphu_{sen}$ | ssn | $flaimax_h$ | $\alpha_{leaf}$ | $hi_{opt}$ |
|---|---|---|---|---|---|---|---|---|---|---|
| temperate cereals | 0.9690 | 0.05 | 0.05 | 0.45 | 0.95 | 0.7 | 2.0 | 0.0 | 0.5 | 0.5 |
| rice | 0.9690 | 0.1 | 0.05 | 0.5 | 0.95 | 0.8 | 2.0 | 0.0 | 0.5 | 0.5 |
| maize | 0.9690 | 0.1 | 0.05 | 0.5 | 0.95 | 0.75 | 2.0 | 0.0 | 0.5 | 0.5 |
| tropical cereals | 0.9690 | 0.15 | 0.01 | 0.5 | 0.95 | 0.85 | 2.0 | 0.0 | 0.5 | 0.25 |
| pulses | 0.9690 | 0.15 | 0.01 | 0.5 | 0.95 | 0.90 | 2.0 | 0.0 | 0.5 | 0.45 |
| temperate roots | 0.9690 | 0.15 | 0.05 | 0.5 | 0.95 | 0.75 | 0.5 | 0.75 | 0.5 | 3.5 |
| tropical roots | 0.9690 | 0.15 | 0.05 | 0.5 | 0.95 | 0.75 | 0.5 | 0.75 | 0.5 | 2.0 |
| sunflower | 0.9690 | 0.15 | 0.01 | 0.5 | 0.95 | 0.7 | 2.0 | 0.0 | 0.5 | 0.4 |
| soybean | 0.9690 | 0.15 | 0.05 | 0.5 | 0.95 | 0.7 | 0.5 | 0.0 | 0.5 | 0.4 |
| groundnuts | 0.9690 | 0.15 | 0.01 | 0.5 | 0.95 | 0.75 | 0.5 | 0.0 | 0.5 | 0.4 |
| rapeseed | 0.9690 | 0.05 | 0.01 | 0.5 | 0.95 | 0.85 | 2.0 | 0.0 | 0.5 | 0.3 |
| sugarcane | 0.9690 | 0.01 | 0.01 | 0.4 | 0.95 | 0.95 | 2.0 | 0.5 | 0.5 | 0.8 |

**Table S12.** Model parameters describing biomass plantation management.

| BFT | Corresponding biomass crop | Harvest interval | Plant density (ha$^{-1}$) |
|---|---|---|---|
| BTrT | Poplar, Willow | 8 years | 8000 |
| BTeT | Eucalyptus | 8 years | 5000 |
| BGrC4 | Miscanthus, Switchgrass | (Multiple) annual harvest | n.a. |

**Table S13.** Overview of BFT parameter values and constants in model equations.

| Parameter | Description | BTrT | BTeT | BGrC4 |
|---|---|---|---|---|
| $g_{\min}$ | Minimum canopy conductance | 0.2 | 0.2 | 0.5 |
| $\text{LAI}_{\text{sapl}}$ | Leaf area index of saplings (-) | 1.6 | 1.6 | 0.001 |
| $\alpha_a$ | fraction of PAR assimilated at ecosystem level, relative to leaf level (-) | 0.8 | 0.8 | 0.8 |
| $T_{\text{lim,CO2}}$ | lower and upper temperature limit for $CO_2$ (°C) | 24,55 | -4.0, 38.0 | 4, 55 |
| $T_{\text{lim,opt,photo}}$ | lower and upper limit of temperature optimum for photosynthesis (°C) | 25, 38 | 15, 30 | 15, 45 |
| $T_{\text{lim,cold,month}}$ | lower and upper coldest monthly mean temperature (°C) | 7, - | -30, 8 | -40, - |
| $\tau_{\text{leaf,root,sapwood}}$ | Turnover leaf, root, sapwood | 2, 2, 10 | 1, 2, 10 | 1,2,- |
| $\text{CA}_{\max}$ | Tree maximum crown area (m$^2$) | 2 | 1.5 | - |
| $C_{\text{sapwood,sapling}}$ | sapling carbon (gC m$^{-2}$) | 2.2 | 2.5 | - |
| $k_{\text{allom1}}$ | Allometry parameter 1 | 110 | 110 | - |
| $k_{\text{allom2}}$ | Allometry parameter 2 | 35 | 35 | - |
| $k_{\text{allom3}}$ | Allometry parameter 3 | 0.75 | 0.75 | - |
| $k_{\text{est}}$ | Saplings per m$^2$ | 0.5 | 0.8 | - |

**Table S14.** Parametrization of irrigation systems in LPJmL4.

| Irrigation system | Distribution uniformity scalar | Conveyance efficiency[1] | Soil evaporation | Inter-ception | Runoff | Irrigation threshold[2] | Minimal irrig. amount |
|---|---|---|---|---|---|---|---|
| Surface | 1.15 | open canal: sand 0.7, loam 0.75, clay 0.8 | unrestricted | no | surface, lateral, percolation | C4: 0.7 C3 (prec <900): 0.8 C3 (prec≥ 900): 0.9 Rice: 1.0 | 1 mm |
| Sprinkler | 0.55 | pipe: 0.95 | unrestricted | yes | lateral, percolation | | 1 mm |
| Drip | 0.05 | pipe: 0.95 | soil evap. of irr. water reduced by 60% | no | none, only indirect precip. leaching | | none |

[1] open canal conveyance efficiency depends on soil hydraulic conductivity ($K_s$): $K_s > 20$: sand, $10 \leq K_s \leq 20$: loam, $K_s < 10$: clay; 50% of conveyance losses are assumed to evaporate, for loam and clay (higher $K_s$) and open canal conveyance the fraction is 60% and 75%, resp. [3] depending on crop type, see Jägermeyr et al. (2015) for details.

Table S15: Equation table describing the different processes represented in the LPJmL4 model.

| Parameter/Variable | abbreviation | unit | Equation |
|---|---|---|---|
| | | Energy balance | |
| Photosynthetic active radiation | $PAR$ | $\mathrm{mol\ m^{-2}\ day^{-1}}$ | $\mathrm{PAR} = 0.5 \cdot c_q \cdot R_{s_{\mathrm{day}}}$ |
| conversion factor from J to mol for solar radiation at 550 nm | $c_q$ | | $c_q = 4.6 \cdot 10^{-6}$ |
| daily incoming solar irradiance | $R_{s_{\mathrm{day}}}$ | $\mathrm{J\ m^{-2}\ day^{-1}}$ | $R_{s_{\mathrm{day}}} = (c + d \cdot \mathrm{ni}) \cdot Q_0 \cdot (\sin(\mathrm{lat}) \cdot \sin(\delta) \cdot h_{1/2} + \cos(\mathrm{lat}) \cdot \cos(\delta) \cdot h_{1/2})$ |
| potential evapotranspiration | PET | $\mathrm{mm\ day^{-1}}$ | $\mathrm{PET} = \mathrm{pt} \cdot E_{\mathrm{eq}}$ |
| equilibrium evapotranspiration | $E_{\mathrm{eq}}$ | $\mathrm{mm\ day^{-1}}$ | $E_{\mathrm{eq}} = \dfrac{s}{s+\gamma} \cdot \dfrac{R_{n_{\mathrm{day}}}}{\lambda}$ |
| daily surface net radiation | $R_{n_{\mathrm{day}}}$ | $\mathrm{J\ m^{-2}\ day^{-1}}$ | |
| latent heat of vaporization | $\lambda$ | $\mathrm{J\ kg^{-1}}$ | $\lambda = 2.495 \times 10^6 + 2380 \cdot T_{\mathrm{air}}$ |
| slope of the saturation vapour pressure curve | $s$ | $\mathrm{Pa\ K^{-1}}$ | $s = 2.502 \times 10^6 \cdot \exp[17.269 \cdot T_{\mathrm{air}}/(237.3 + T_{\mathrm{air}})]]/(237.3 + T_{\mathrm{air}})^2$ |
| psychrometric constant | $\gamma$ | $\mathrm{Pa\ K^{-1}}$ | $\gamma = 65.05 + 0.064 \cdot T_{\mathrm{air}}$ |
| Priestley-Taylor coefficient | pt | $\mathrm{W\ m^{-2}}$ | |
| net surface radiation | $R_n$ | $\mathrm{W\ m^{-2}}$ | |
| incoming solar irradiance (downward) at the surface | $R_s$ | $\mathrm{W\ m^{-2}}$ | $R_s = (c + d \cdot \mathrm{ni}) \cdot Q_0 \cdot \cos(z)$ or as input |

| Parameter/Variable | abbreviation | unit | Equation |
|---|---|---|---|
| outgoing (upward positive) net long-wave radiation flux at the surface | $R_l$ | W m$^{-2}$ | $R_l = (b + (1-b)\cdot \text{ni})\cdot(A - T_{\text{air}})$ or as input |
| albedo | $\beta$ | | $\beta = \sum_{\text{PFT}=1}^{n_{\text{PFT}}}\beta_{\text{PFT}}\cdot\text{FPC} + F_{\text{bare}}\cdot(F_{\text{snow}}\cdot\beta_{\text{snow}} + (1 - F_{\text{snow}})\cdot\beta_{\text{soil}})\beta = \sum$ |
| albedo bare soil | $\beta_{\text{soil}}$ | | |
| albedo snow | $\beta_{\text{snow}}$ | | |
| plant compartments specific albedo | $\beta_{\text{PFT}}$ | | |
| coverage of bare soil | $F_{\text{bare}}$ | | |
| coverage of snow | $F_{\text{snow}}$ | | |
| empirical constant | $b$ | | see Prentice et al. (1993) |
| empirical constant | $A$ | | see Prentice et al. (1993) |
| mean daily air temperature | $T_{\text{air}}$ | °C | |
| net outgoing daytime long-wave flux | $R_{l_{\text{day}}}$ | J m$^{-2}$ day$^{-1}$ | $R_{l_{\text{day}}} = R_l \cdot \text{daylength} \cdot 3600$ |
| angular distance between the sun's rays and the local vertical | $z$ | | |
| proportion of bright sky | ni | | $\text{ni} = 1 - \text{cloudiness}$ |
| empirical constant | $c$ | | see Prentice et al. (1993) |
| empirical constant | $d$ | | see Prentice et al. (1993) |
| solar constant | $Q_0$ | W m$^{-2}$ | $Q_0 = Q_{00}\cdot(1 + 2\cdot 0.01675\cdot\cos(2\cdot\pi\cdot i/365))$ |
| solar zenith angle | $z$ | | |
|  latitude | lat | radians | $\cos(z) = \sin(\text{lat})\cdot\sin(\delta) + \cos(\text{lat})\cdot\cos(\delta)\cdot\cos(h)$ |
| hour angle | $h$ | | |

| Parameter/Variable | abbreviation | unit | Equation |
|---|---|---|---|
| solar declination | $\delta$ | radians | $\delta = -23.4 \cdot \pi/180 \cdot \cos(2\cdot\pi\cdot(i+10)/365)$ |
| half-day length | $h_{1/2}$ | angular units | $h_{1/2} = \arccos(-(\sin(\text{lat})\cdot\sin(\delta))/(\cos(\text{lat})\cdot\cos(\delta)))$ |
| duration of sunshine of a single day | daylength | hours | $\text{daylength} = 24 \cdot \frac{h_{1/2}}{\pi}$ |
|  Soil temperatures | $T_{\text{soil}}$ | °C | $\frac{\partial T_{\text{soil}}}{\partial t} = \alpha \cdot \frac{\partial^2 T_{soil}}{\partial \bar z^2}$ |
| thermal diffusivity | $\alpha = \lambda/c$ | $\text{m}^2\,\text{s}^{-1}$ | |
| thermal conductivity | $\lambda$ | $\text{W}\,\text{m}^{-1}\,\text{K}^{-1}$ | |
| soil layer | $l$ | | |
| time step | $t$ | | |
| stability criterion | $r$ | | $r = \frac{\alpha \Delta t}{(\Delta z)^2}$ |
| Heat capacity | $c$ | $\text{J}\,\text{K}^{-1}\,\text{m}^{-3}$ | $c = c_{\min} \cdot m_{\min} + c_{\text{water}} \cdot m_{\text{water}} + c_{\text{ice}} \cdot m_{\text{ice}}$ |
| soil minerals | $c_{\min}$ | | |
| soil water content | $c_{\text{water}}$ | | |
| soil ice content | $c_{\text{ice}}$ | | |
| corresponding shares of $c_{\min}, c_{\text{water}}, c_{\text{ice}}$ | $m$ | $\text{m}^3$ | |

Plant physiology

| Parameter/Variable | abbreviation | unit | Equation |
|---|---|---|---|
| absorbed photosynthetically active radiation | APAR | mol m$^{-2}$ day$^{-1}$ |
 $APAR_{PFT} = PAR \cdot FAPAR_{PFT} \cdot \alpha_{apFT}$
 $FAPAR_{PFT} = FPC_{PFT} \cdot \big((phen_{PFT} - F_{SnowGC}) \cdot (1 - \beta_{leaf,PFT}) - ((1-ph\ldots$ |
|  fractional absorbed photosynthetically active radiation | $FAPAR_{PFT}$ | | |
| scaling factor to scale leaf-level photosynthesis in LPJmL4 to biome level | $\alpha_{apFT}$ | | |
| daily phenological status | $phen_{PFT}$ | | |
| fraction of snow in the green canopy | $F_{SnowGC}$ | | |
| foliage projective cover of the respective PFT | $FPC_{PFT}$ | |
 $FPC_{PFT} = CA_{ind} \cdot P \cdot FPC_{ind}$ |
| masking of the ground by stems and branches without leaves | $c_{fstem}$ | | |
| gross photosynthesis rate | $A_{gd}$ | gC m$^{-2}$ day$^{-1}$ | $A_{gd} = \left(J_E + J_C - \sqrt{(J_E + J_C)^2 - 4 \cdot \theta \cdot J_E \cdot J_C}\right)/(2 \cdot \theta) \cdot \text{daylength}$ |
| light-limited photosynthesis rate | $J_E$ | mol C m$^{-2}$ hour$^{-1}$ | $J_E = C_1 \cdot \dfrac{APAR}{\text{daylength}}$ |
| for C3-Photosynthesis | | |
 $C_1 = \alpha_{C3} \cdot T_{stuess} \cdot \left(\dfrac{p_i - \Gamma_*}{p_i + 2 \Gamma_*}\right)$ |
| for C4-Photosynthesis | | | $C_1 = \alpha_{C4} \cdot T_{stress} \cdot \left(\dfrac{\lambda}{\lambda_{maxC4}}\right)$ |
| internal partial pressure of CO$_2$ | $p_i$ | Pa | |
| ambient pressure | $p_a$ | Pa | |
| relation of internal and ambient pressure | $\lambda$ | | $p_i = \lambda \cdot p_a$ |

| Parameter/Variable | abbreviation | unit | Equation |
|---|---|---|---|
| PFT-specific temperature inhibition function | $T_{\text{stress}}$ | | |
| intrinsic quantum efficiencies for CO$_2$ uptake in C3 plants | $\alpha_{C3}$ | | |
| intrinsic quantum efficiencies for CO$_2$ uptake in C4 plants | $\alpha_{C4}$ | | |
| CO$_2$ compensation point | $\Gamma_*$ | | $\Gamma_* = \frac{[O_2]}{2\cdot\tau}$ |
| specificity factor | $\tau$ | | $\tau = \frac{V_c\cdot K_C}{V_m\cdot K_O}$ |
| Michaelis-Menten constant | $K_C$ | | |
| Michaelis-Menten constant | $K_O$ | | |
| partial pressure of O$_2$ | $O_2$ | | |
| Rubisco-limited photosynthesis rate | $J_C$ | mol C m$^{-2}$ hour$^{-1}$ | $J_C = C_2 \cdot V_m$ |
| maximum Rubisco capacity | $V_m$ | | $V_m = \frac{1}{b}\cdot\frac{C_1}{C_2}\left((2\cdot\theta-1)\cdot s - (2\cdot\theta\cdot s - C_2)\cdot\sigma\right)\cdot \text{APAR}$

 $\sigma = \sqrt{1-\frac{C_2-2}{C_2-\theta s}}$
 $s = 24/\text{daylength}\cdot b$ |
| | $s$ | | |
| | $C_2$ | | $C_2 = \frac{p_i - \Gamma_*}{p_i + K_C\left(1+\frac{[O_2]}{K_O}\right)}$ |
| leaf respiration | $R_{\text{leaf}}$ | gC day$^{-1}$ | $R_{\text{leaf}} = V_m \cdot b$ |
| daily net photosynthesis | $A_{\text{nd}}$ | gC day$^{-1}$ | |
| dark respiration | $R_d$ | | $R_d = (1-\text{daylength}/24)\cdot R_{\text{leaf}}$ |
| daily net daytime photosynthesis | $A_{\text{dt}}$ | | $A_{\text{dt}} = A_{\text{nd}} + R_d$ |
|  canopy conductance | $g_c$ | mm s$^{-1}$ | $g_c = g_{\text{min}} + \frac{1.6 A_{\text{dt}}}{p_a(1-\lambda)}$ |

| Parameter/Variable | abbreviation | unit | Equation |
|---|---|---|---|
| PFT-specific minimum canopy conductance | $g_{min}$ | $\mathrm{mm\,s^{-1}}$ | |
| ambient partial pressure of $CO_2$ | $p_a$ | bar | |
| parameter describing the ratio of the intercellular to the ambient $CO_2$ concentration | $\lambda$ | | |
| daily phenology status | $phen_{PFT}$ | | $phen_{PFT} = f_{cold} \cdot f_{light} \cdot f_{water} \cdot f_{heat}$ |
|  limited by cold temperatures |  $f_{cold}$ | | $f_{cold}, f_{heat}$ |
|  relation to light |  $f_{light}$ | | $f_{light}$ |
| relation to water availability |  $f_{water}$ | | $f_{water}$ |
| limited by heat stress | $f_{heat}$ | | |
| inflection point of the respective logistic function | $b_x$ | | |
| slope of the respective logistic function | $sl_x$ | | |
| change rate parameter | $\tau_x$ | | |
| CN ratio of above-ground tissue | $CN_{sapwood}$ | | |
| CN ratio of below-ground tissue | $CN_{root}$ | | |
| Temperature | $T(T_{air}, T_{soil})$ | $^{\circ}C$ | |
| daily  phenology | $phen_{PFT}$ | | |
|  above-ground | $R_{sapwood}$ | $\mathrm{gC\,m^{-2}\,day^{-1}}$ |  |
| autotrophic respiration  | | |  |
| aboveground tissue | | | |

Plant functional types (PFT) table — tracked-change edits shown

| Parameter/Variable | abbreviation | unit | Equation |
|---|---|---|---|
| autotrophic respiration  belowground tissue | $R_{\mathrm{root}}$ | gC $\mathrm{m}^{-2}$ $\mathrm{day}^{-1}$ |  $R_{\mathrm{root}} = P \cdot r_{\mathrm{PFT}} \cdot k \cdot \frac{C_{\mathrm{root,ind}}}{CN_{\mathrm{root}}} \cdot g(T_{\mathrm{soil}}) \cdot \mathrm{phen}_{\mathrm{PFT}}$ |
| respiration rate | $r_{\mathrm{PFT}}$ | gC $\mathrm{gN}^{-1}$ $\mathrm{day}^{-1}$ | |
| temperature function | $g(T)$ | | $g(T) = \exp\left[308.56 \cdot \left(\frac{1}{56.02} - \frac{1}{(T+46.02)}\right)\right]$ |
| leaf respiration | $R_{\mathrm{leaf}}$ | | $R_{\mathrm{leaf}} = V_m \cdot b$ |
| static parameter | $b$ | | |
|  daily net primary production | NPP | gC $\mathrm{m}^{-2}$ $\mathrm{day}^{-1}$ | $\mathrm{NPP} = 0.75 \cdot (\mathrm{GPP} - R_{\mathrm{leaf}} - R_{\mathrm{sapwood}} - R_{\mathrm{root}})$ |

Plant functional types (PFT)

| Parameter/Variable | abbreviation | unit |
|---|---|---|
|  leaf mass |  $C_{\mathrm{leaf,ind}}$ | gC$\cdot \mathrm{ind}^{-1}$ |
| fine root mass |  $C_{\mathrm{root,ind}}$ | gC$\cdot \mathrm{ind}^{-1}$ |
| sapwood mass |  $C_{\mathrm{sapwood,ind}}$ | gC$\cdot \mathrm{ind}^{-1}$ |
| heartwood mass |  $C_{\mathrm{heartwood,ind}}$ | gC$\cdot \mathrm{ind}^{-1}$ |

| Parameter/Variable | abbreviation | unit | Equation |
|---|---|---|---|
| average individual leaf area | $\text{LA}_{\text{ind}}$ | $\text{m}^2 \cdot \text{ind}^{-1}$ | $\text{LA} = k_{\text{la:sa}} \cdot \text{SA} \quad \text{LA}_{\text{ind}} = k_{\text{la:sa}} \cdot \text{SA}_{\text{ind}}$ |
| ratio of leaf to sapwood area | $k_{\text{la:sa}}$ | | |
| sapwood cross-sectional area | $\text{SA}_{\text{ind}}$ | | |
| grass leaf biomass | $C_{\text{leaf}}$ | $\text{gC m}^{-2}$ | $C_{\text{leaf}} = \text{lr}_{\text{max}} \cdot \omega \cdot C_{\text{roots}}$ |
| leaf-to-root mass ratio | $\text{lr}$ | | $\text{lr} = \text{lr}_p \cdot W_{\text{supply}}/W_{\text{demand}}$ |
| maximum leaf-to-root mass ratio | $\text{lr}_{\text{max}}$ | | |
| tree height | $H$ | m | $H = k_{\text{allom2}} \cdot D^{k_{\text{allom3}}}$ |
| stem diameter | $D$ | m | |
| crown area | $\text{CA}_{\text{ind}}$ | $\text{m}^{2.2} \cdot \text{ind}^{-1}$ | $\text{CA} = k_{\text{allom1}} \cdot D^{k_{\text{rp}}} \quad \text{CA}_{\text{ind}} = k_{\text{allom1}} \cdot D^{k_{\text{rp}}}$ |
| constant wood density | $\text{WD}$ | $\text{gC m}^{-2}$ | $H = \dfrac{C_{\text{sapwood}} \cdot k_{\text{la:sa}}}{\text{WD} \cdot C_{\text{leaf}} \cdot \text{SLA}}$ |
| individual leaf area index | $\text{LAI}_{\text{ind}}$ | | $H = \dfrac{C_{\text{sapwood,ind}} \cdot k_{\text{la:sa}}}{\text{WD} \cdot C_{\text{leaf,ind}} \cdot \text{SLA}}$ $\quad \text{LAI}_{\text{ind}} = \dfrac{C_{\text{leaf}} \cdot \text{SLA}}{\text{CA}} \quad \text{LAI}_{\text{ind}} = \dfrac{C_{\text{leaf,ind}} \cdot \text{SLA}}{\text{CA}_{\text{ind}}}$ |
| leaf longevity | $\alpha_{\text{leaf}}$ | months | |
| | $\beta_0$ | | |
| dry matter carbon content of leaves | $\text{DM}_C$ | | |
| foliar projective cover | $\text{FPC}_{\text{ind}}$ | | $\text{FPC}_{\text{ind}} = 1 - \exp(-k \cdot \text{LAI}_{\text{ind}})$ |
| mean number of individuals per unit area | $P$ | $\text{ind m}^{-2}$ | |
| establishment rate | $k_{\text{est}}$ | $\text{saplings m}^{-2}\,\text{a}^{-1}$ | |
| background mortality rate | $\text{mort}_{\text{greff}}$ | $\text{ind m}^{-2}\,\text{a}^{-1}$ | $\text{mort}_{\text{greff}} = \dfrac{k_{\text{mort1}}}{1+k_{\text{mort2}} \cdot \text{greff}} \quad \text{mort}_{\text{greff}} = P \cdot \dfrac{k_{\text{mort1}}}{1+k_{\text{mort2}} \cdot \text{greff}}$ |
| yearly growth efficiency | $\text{greff}$ | | |
| asymptotic maximum mortality rate | $k_{\text{mort1}}$ | | |

| Parameter/Variable | abbreviation | unit | Equation |
|---|---|---|---|
| parameter governing the slope of the relationship between mortality and growth efficiency | $k_{mort2}$ | | |
| heat stress | $mort_{heat}$ | °C ind m$^{-2}$ a$^{-1}$ | $mort_{heat} = P \cdot \frac{gdd_{tw}}{tw_{PFT}}$ |
| parameter value of the heat damage function | $tw_{PFT}$ | | |
| temperatures above threshold (accumulated) | $gdd_{tw}$ | | |
| Nesterov index | $NI(N_d)$ | °C | $NI(N_d) = \sum_{if\, Pr(d)\leq 3mm}^{N_d} T_{max}(d) \cdot (T_{max}(d) - T_{dew}(d))$ |
| daily maximum temperature | $T_{max}$ | °C | |
| dew-point temperature | $T_{dew}$ | °C | |
| positive temperature day | $d$ | | |
| probability of fire spread | $P_{spread}$ | | $P_{spread} = \begin{cases} 1 - \frac{\omega_0}{m_e}, & \omega_0 \leq m_e \\ 0, & \omega_0 > m_e \end{cases}$ |
| litter moisture | $\omega_0$ | | |
| moisture of extinction | $m_e$ | | |
| fire danger index | FDI | | $FDI = \max\left\{0, 1 - \frac{1}{m_e} \cdot \exp\left(-NI \cdot \sum_{p=1}^{n} \frac{\alpha_p}{n}\right)\right\}$ |
| slope of the probability risk function | $\alpha_p$ | | |
| Human-caused ignitions | $n_{h,ig}$ | ignitions | $n_{h,ig} = P_D \cdot k(P_D) \cdot a(N_D)/100$ |
| population density | $P_D$ | ind km$^{-2}$ | $k(P_D) = 30.0 \cdot \exp(-0.5 \cdot \sqrt{P_D})$ |
| propensity of people to produce ignition events | $a(N_D)$ | individual$^{-1}$ d$^{-1}$ | $a(N_D) = \frac{N_{h,obs}}{t_{obs} \cdot LFS \cdot P_D}$ |

| Parameter/Variable | abbreviation | unit | Equation |
|---|---|---|---|
| average number of human-caused fires | $N_{h,\mathrm{obs}}$ | | |
| observation years | $t_{\mathrm{obs}}$ | | |
| grid cell size area | $A$ | m² | $A_b = \min(E(n_{\mathrm{ig}}) \cdot \mathrm{FDI} \cdot A_f, A)$ |
| mean fire area | $a_f$ | ha | $\overline{a_f} = \dfrac{\frac{\pi}{4 \cdot L_B} \cdot D_T^2}{10000}$ |
| independent estimates of the numbers of lightning | $n_{l,\mathrm{ig}}$ | | |
| human-caused ignition events | $n_{h,\mathrm{ig}}$ | | |
| forward rate of spread | $\mathrm{ROS}_{f,\mathrm{surface}}$ | m min⁻¹ | $\mathrm{ROS}_{f,\mathrm{surface}} = \dfrac{I_R \cdot \zeta \cdot (1+\Phi_w)}{\rho_b \cdot \epsilon \cdot Q_{\mathrm{ig}}}$ |
| reaction intensity | $I_R$ | kJ m⁻² min⁻¹ | |
| propagating flux ratio | $\zeta$ | | |
| multiplier that accounts for the effect of wind | $\Phi_w$ | | |
| fuel bulk density | $\rho_b$ | kg m⁻³ | |
| effective heating number | $\epsilon$ | | |
| heat of pre-ignition | $Q_{\mathrm{ig}}$ | kJ kg⁻¹ | |
| fire duration | $t_{\mathrm{fire}}$ | min | $t_{\mathrm{fire}} = \dfrac{241}{1+240 \cdot \exp(-11.06 \cdot \mathrm{FDI})}$ |
| length to breadth ratio of elliptical fire | $L_B$ | | |
| length of major axis | $D_T$ | m | $D_T = \mathrm{ROS}_{f,\mathrm{surface}} \cdot t_{\mathrm{fire}} + \mathrm{ROS}_{b,\mathrm{surface}} \cdot t_{\mathrm{fire}}$ |
| surface as the backward rate of spread | $\mathrm{ROS}_b$ | | |
| crown damage | CK | 0-1 | $P_m(\mathrm{CK}) = r_{\mathrm{CK}} \cdot \mathrm{CK}^p$ |
| resistance factor | $r_{\mathrm{CK}}$ | | |

**Crop functional types (CFT)**

| Parameter/Variable | abbreviation | unit | Equation |
|---|---|---|---|
| phenological heat unit | phu | | $\mathrm{phu} = -0.1081 \cdot (\mathrm{sdate} - \mathrm{keyday})^2 + 3.1633 \cdot (\mathrm{sdate} - \mathrm{keyday}) + \mathrm{phu}_{w_{\mathrm{high}}}$ |
| harvest indices |  hi$_{\mathrm{opt}}$ | | |
| heat units | hu | | |
| heat units accumulated |  hu$_{\mathrm{sum}}$ | | $\mathrm{hu}_{\mathrm{sum}} = \sum_{t'=\mathrm{sdate}}^{t} \mathrm{hu}_{t'} \cdot v_{\mathrm{rf}} \cdot p_{\mathrm{rf}}$ |
| phenological development stage | fphu | | $\mathrm{fphu} = \mathrm{hu}_{\mathrm{sum}}/\mathrm{phu}$ |
| reduction factor for vernalization | $v_{\mathrm{rf}}$ | | $v_{\mathrm{rf}} = (\mathrm{vdsum} - 10.0)/(\mathrm{pvd} - 10.0)$ |
| reduction factor for photoperiod | $p_{\mathrm{rf}}$ | | $p_{\mathrm{rf}} = (1 - p_{\mathrm{sens}}) \cdot \min(1, \max(0, (\mathrm{daylength} - p_b)/(p_s - p_b))) + p_{\mathrm{sens}}$ |
| day of solstice | keyday | | |
| minimum base temperature for the accumulation of heat unit | $T_{\mathrm{base}_{\mathrm{low}}}$ | | |
| 20-year moving average annual temperature |  atemp$_{20}$ | | |
| CFT-specific scaling factor | pf$_{\mathrm{CFT}}$ | | |
| Vernalization requirements | pvd | | $\mathrm{pvd} = \mathrm{vern}_{\mathrm{date20}} - \mathrm{sdate} - \mathrm{ppvd}_{\mathrm{CFT}}, \quad 0 \leq \mathrm{pvd} \leq 60$ |
| CFT-specific vernalization factor | ppvd$_{\mathrm{CFT}}$ | | |
| julian day of the year of sowing | sdate | | |

| Parameter/Variable | abbreviation | unit | Equation |
|---|---|---|---|
| multi-annual average of the first day of the year when temperatures rise above a CFT-specific vernalization threshold | $vern_{date20}$ | | |
| effective number of vernalizing days | vdsum | | |
| parametrized sensitivity to photoperiod | $p_{sens}$ | | |
| duration of daylight (sunrise to sunset) | daylength | hours | |
| base photoperiod | $p_b$ | hours | |
| aturation photoperiod | $p_s$ | hours | |
| maximum leaf area index | $LAI_{max}$ $LAI_{max}$ | | |
| fraction of total biomass that is allocated to the roots | $f_{root}$ | | $f_{root} = \dfrac{0.4 - (0.3 \cdot fphu) \cdot wdf}{wdf + \exp(6.13 - 0.0883 \cdot wdf)}$ |
| ratio between accumulated daily transpiration and accumulated daily water demand | wdf | | |
| onset of senescence | ssn | | |
| turning points in the phenological development | $fphu_c$, $fphu_k$ | | |
| corresponding fraction of the maximum green LAI | $flai_{max_c}$, $flai_{max_k}$ | | $flai_{max} = \dfrac{fphu}{fphu + c\left(\frac{c}{k}\right)^{\frac{fphu_c - fphu}{fphu_k - fphu_c}}}$ |
| onset of senescence as point in the phenological development | $fphu_{sen}$ | | |

| Parameter/Variable | abbreviation | unit | Equation |
|---|---|---|---|
| daily increment |  $\underline{lai_{inc,t}}$ | |
 $\underline{lai_{inc,t} = (flai_{max_t} - flai_{max_{t-1}}) \cdot lai_{max}}$ |
| maximum green LAI | $flai_{max}$ | | |
| LAI | LAI | | $LAI_t = \sum_{t'=\text{sdate}}^{t} lai_{inc,t'} \cdot \omega$ |
| specific leaf area | SLA | | $SLA = \dfrac{2 \times 10^{-4}}{DM_C} \cdot 10^{(\beta_0 - \beta_1 \cdot \log(\alpha_{leaf}))/\log(10)}$ |
| harvest index | HI | | $HI = \begin{cases} fhi_{opt} \cdot hi_{opt}, & \text{if } hi_{opt} \geq 1 \\ fhi_{opt} \cdot (hi_{opt} - 1.0) + 1.0, & \text{otherwise} \end{cases}$
 $fhi_{opt} = 100 \cdot fphu/(100 \cdot fphu + \exp(11.1 - 10.0 \cdot fphu))$ |
| | $fhi_{opt}$ | | |
| storage organ | $C_{so}$ | $\underline{gC\ m^{-2}}$ | $C_{so} = HI \cdot (C_{leaf} + C_{so} + C_{pool})$ |
| Excess biomass | $C_{pool}$ | $\underline{gC\ m^{-2}}$ | |

Soil and litter carbon pools

| Parameter/Variable | abbreviation | unit | Equation |
|---|---|---|---|
| heterotrophic respiration | $R_h$ | $gC\ m^{-2}\ day^{-1}$ | $R_h = R_{h,\text{litter}} + R_{h,\text{fastSoil}} + R_{h,\text{slowSoil}}$ |
| carbon pool size of soil or litter  $\underline{C_L}$ | | $gC\ \underline{m^{-2}\ layer^{-1}}$ |
 $\underline{C_{0(l)} \cdot (1 - \exp(-k_{(l,t)}))}$ $\underline{\dfrac{dC_{(l)}}{dt} = -k_{(l)} \cdot C_{(l)}}$ |
|  per layer | | | |
| decomposition rates for litter | $k$ | $\underline{a^{-1}\ layer^{-1}}$ | $k_{(l,p)} = \dfrac{1}{\tau_{10(p)}} \cdot g(T_{soil}) \cdot f(\theta)$ |
| mean residence time | $\tau_{10}$ |  $\underline{a}$ | |
| soil volume fraction of the layer | $\theta$ | | |

| Parameter/Variable | abbreviation | unit | Equation |
|---|---|---|---|
| fraction of soil organic carbon per layer | $Cf_L$ | gC | $Cf_{(l)} = 10^{k_{soc} \cdot \log_{10}(d_{(l)})}$ |
| relative share of the layer $l$ | $d_{(l)}$ | | |
| soil layer depth | $k_{soc}$ | mm | |
| total amount of soil carbon | $C_{s_{total}}$ | gC | $C_{(l)} = \sum_{PFT=1}^{n_{PFT}} d_{(l)}^{k_{soc_{PFT}}} \cdot C_{s_{total}}$ |
| mean annual decomposition rate | $k_{mean}$ | gC day$^{-1}$ | $k_{mean_{PFT}(p)} = \sum_{l=1}^{n_{soil}} \left( k_{mean(l)} \cdot Cf_{(l,p)} \right)$ |
| mean decomposition rate for each PFT | $k_{mean_{PFT}}$ | | $k_{mean_{PFT}} = \sum_{l=1}^{n_{soil}} \left( k_{mean(l)} \cdot Cf_{(l,PFT)} \right)$ |
| annual carbon shift rates | $C_{shift}$ | a$^{-1}$ | $C_{shift(l,p)} = \frac{Cf_{(l,p)} \cdot k_{mean(l)} \cdot k_{mean_{PFT}(p)}}{\ldots}$ $\quad C_{shift(l,PFT)} = \frac{Cf_{(l,PFT)} \cdot k_{mean(l)} \cdot k_{mean_{PFT}}}{\ldots}$ |
| infiltration rate of rain water into the soil | infl | mm | $infl = P \cdot \sqrt{1 - \frac{SW_{(0)} - WPW_{(0)}}{W_{sat_{(0)}} - WPW_{(0)}}}$ |

Water balance

| | | | |
|---|---|---|---|
| soil water content at saturation | $W_{sat}$ | mm | |
| soil water content at wilting point | WPW $W_{pwp}$ | mm | |
| total actual soil water content | SW | mm | |
| portion of daily precipitation | $P$ | mm | maximum 4 mm |
| soil water content between saturation and field capacity | FW | mm | |
| soil layer | $l$ | | |
| travel time through the soil layer | TT | hours | $TT_{(l)} = \frac{FW_{(l)}}{HC_{(l)}}$ |

| Parameter/Variable | abbreviation | unit | Equation |
|---|---|---|---|
| hydraulic conductivity | HC | $\mathrm{mm\,h^{-1}}$ | $\mathrm{HC}_{(l)} = K_{s(l)} \cdot \left( \frac{\mathrm{SW}_{(l)}}{W_{\mathrm{sat}(l)}} \right)^{\beta_{(l)}}$ |
| saturated conductivity | $K_s$ | $\mathrm{mm\,h^{-1}}$ | |
| percolation | perc | $\mathrm{mm\,day^{-1}}$ | $\mathrm{perc}_{(l)} = \mathrm{FW}_{(t,l)} \cdot \left[ 1 - \exp\left( \frac{-\Delta t}{\mathrm{TT}_{(l)}} \right) \right]$ |
| Interception | $I$ | $\mathrm{mm\,day^{-1}}$ | |
| PFT-specific interception storage parameter | $I_{pft}I_{\mathrm{PFT}}$ | | |
| PFT-specific leaf area per unit of grid cell area | $\mathrm{LAI}_{pft}\mathrm{LAI}_{\mathrm{PFT}}$ | | |
| daily precipitation | Pr | $\mathrm{mm\,day^{-1}}$ | |
| Soil evaporation | $E_s$ | $\mathrm{mm\,day^{-1}}$ | |
| vegetation cover | $f_v$ | % | |
| evaporation-available soil water | $w_{\mathrm{evap}}$ | | |
| plant transpiration | $E_T$ | | $E_T = \min(S, D) \cdot f_v$ |
| daily water stress | $\omega$ | | |
| Soil water supply | $S$ | | $S = E_{\max} \cdot w_r \cdot \mathrm{phen_{PFT}}$ |
| PFT-specific maximum water transport capacity | $E_{\max}$ | | |
| water accessible for plants | $w_r$ | | $w_r = \sum_{l=1}^{n_{\mathrm{soil}}-1} w_l \cdot \mathrm{rootdist}_l$ |
| relative water content at field capacity | $w$ | | |
| fraction of roots from surface to $z$ | rootdist | | $\mathrm{rootdist} = 1 - \beta_{\mathrm{root}}^z$ |
| soil depth | $z$ | mm | |
| root distribution parameter | $\beta_{\mathrm{root}}$ | | |

| Parameter/Variable | abbreviation | unit | Equation |
|---|---|---|---|
| fraction of water that corresponds to their foliage projected cover |  |  |  $S_{\mathrm{PFT}} = S \cdot \mathrm{FPC}_{\mathrm{PFT}}$ |
| root biomass | bm$_{\mathrm{root}}$ | | |
| Atmospheric demand | $D$ | | $D = (1.0 - \mathrm{wet}) \cdot E_{\mathrm{eq}} \cdot \alpha_m / (1 + g_m/g_c)$ |
| maximum Priestley–Taylor coefficient | $\alpha_m$ | | |
| conductance scaling factor | $g_m$ | | |
| fraction of $E_{\mathrm{eq}}$ that was used to vaporize intercepted water from the canopy | wet | | |
| potential canopy conductance | $g_c$ | | |
| homogeneous segments of length | $L$ | | |
| outflow of a linear reservoir cascade | $Q_{\mathrm{out}}$ | | $Q_{\mathrm{out}}(t) = Q_{\mathrm{in}} \cdot \dfrac{1}{K \cdot \Gamma(n)} \left(\dfrac{t}{K}\right)^{n-1} \cdot \exp(-t/K)$ |
| instantaneous inflow | $Q_{\mathrm{in}}$ | | |
| gamma function | $\Gamma(n)$ | | |
| storage parameter | $K$ | | |
| linear reservoir segment of length | $L$ | km | |
| flow velocity | $v$ | m s$^{-1}$ | $K = \dfrac{L}{v}$ |
| CFT-specific irrigation threshold | it | | |
| amount of water required in the upper 50 cm soil | NIR | | $\mathrm{NIR} = W_{\mathrm{fc}} - w_a - w_{\mathrm{ice}}, \quad \mathrm{NIR} \geq 0$ |
| available soil water | $w_a$ | | |
| frozen soil water | $w_{\mathrm{ice}}$ | mm | |

| Parameter/Variable | abbreviation | unit | Equation |
|---|---|---|---|
| conveyance efficiency | $E_c$ | | |
| application requirements | AR | | $AR = W_{\text{sat}} - W_{fc} - W_{\text{pwp}}) \cdot d_u - w_{\text{fw}}, \quad AR \geq 0$ |
| gross irrigation requirements | GIR | | $GIR = \frac{\text{NIR} + AR - \text{Store}}{E_c}$ |
| storage buffer | Store | | |
| soil saturated hydraulic conductivity | $K_s$ | | |
| water distribution uniformity scalar | $d_u$ | | |
| available free water | $w_{\text{fw}}$ | | |
| annual variation coefficients for precipitation | $CV_{\text{prec}}$ | | |
| annual variation coefficients for temperature | $CV_{\text{temp}}$ | | |
| biomass after the last harvest event | $MC_{\text{leaf}}$ | | |
| harvest index | $H_{\text{frac}}$ | | |